# Second-Order Bilevel Optimization with Accelerated Convergence Rates

**Sheng Yang** [*1]  **Chengchang Liu** [*2]  **Lesi Chen** [3]  **John C.S. Lui** [2]

## Abstract

This paper studies second-order methods for nonconvex-strongly-convex bilevel optimization. We propose a novel fully second-order bilevel approximation method (FSBA) that achieves an iteration complexity of $\tilde{\mathcal{O}}(\epsilon^{-1.5})$ for finding the $(\epsilon, \mathcal{O}(\sqrt{\epsilon}))$ second-order stationary point of the hyper-objective function. Our results demonstrate that second-order methods can achieve an accelerated convergence rate than first-order methods in bilevel optimization. To address the heavy computational cost associated with the second-order oracle, we introduce a lazy variant of FSBA, called LFSBA, which reuses second-order information across several iterations. We prove that LFSBA exhibits better computational complexity than FSBA by a factor of $\sqrt{d}$, where $d$ is the dimension of the problem. We also apply a similar idea to nonconvex strongly-concave minimax optimization and propose the lazy minimax cubic-regularized Newton (LMCN) method with better computational complexity compared to existing second-order methods.

## 1. Introduction

In this paper, we consider the following bilevel optimization problem:

$$\min_{\boldsymbol{x}\in\mathbb{R}^{d_x}} \varphi(\boldsymbol{x}):= f\left(\boldsymbol{x},\boldsymbol{y}^*(\boldsymbol{x})\right),$$
$$\text{where} \quad \boldsymbol{y}^*(\boldsymbol{x}):=\arg\min_{\boldsymbol{y}\in\mathbb{R}^{d_y}} g(\boldsymbol{x},\boldsymbol{y}). \tag{1}$$

We assume that the lower-level function $g$ is strongly convex with respect to $\boldsymbol{y}$. This formulation is widely applied in

[1]Department of Statistics, University of California, Riverside [2]Department of Computer Science & Engineering, The Chinese University of Hong Kong [3]IIIS, Tsinghua University. Sheng Yang and Chengchang Liu contribute equally to this work. Correspondence to: Chengchang Liu <7liuchengchang@gmail.com>.

*Proceedings of the 43$^{rd}$ International Conference on Machine Learning*, Seoul, South Korea. PMLR 306, 2026. Copyright 2026 by the author(s).

various machine learning applications, including but not limited to hyperparameter tuning (Franceschi et al., 2018; Pedregosa, 2016), neural architecture search (Liu et al., 2018; Zhang et al., 2021; Zoph & Le, 2016), meta-learning (Ji et al., 2022; Rajeswaran et al., 2019), reinforcement learning (Hong et al., 2023; Konda & Tsitsiklis, 1999), and adversarial training (Brückner & Scheffer, 2011; Goodfellow et al., 2020; Robey et al., 2023; Zhang et al., 2022).

The strong convexity of lower-level function $g$ with respect to $\boldsymbol{y}$ and proper smooth assumptions on $f$ and $g$ ensure the differentiability of $\varphi(\mathbf{x})$, whose gradient can be expressed as:

$$\nabla\varphi(\boldsymbol{x}) = \nabla_x f(\boldsymbol{x},\boldsymbol{y}^*(\boldsymbol{x}))$$
$$- \nabla_{xy}^2 g(\boldsymbol{x},\boldsymbol{y}^*(\boldsymbol{x}))\left(\nabla_{yy}^2 g(\boldsymbol{x},\boldsymbol{y}^*(\boldsymbol{x}))\right)^{-1}\nabla_y g(\boldsymbol{x},\boldsymbol{y}^*(\boldsymbol{x})).$$

Previous second-order methods, such as approximate implicit differentiation (AID) (Ghadimi & Wang, 2018; Ji et al., 2021; Liao et al., 2018; Lorraine et al., 2020) and iterative differentiation (ITD) (Arbel & Mairal, 2022a; Bolte et al., 2021; Domke, 2012; Franceschi et al., 2017; 2018), utilize Hessian-vector products (Ji et al., 2021; Li et al., 2022) to estimate the hypergradient $\nabla\varphi(\boldsymbol{x})$. They perform inexact gradient descent or (perturbed) accelerate gradient descent (Yang et al., 2023; Wang et al., 2024a) to minimize $\varphi(\cdot)$. The iteration complexities of these methods are consistent with those of first-order methods for non-convex minimization problems: $\mathcal{O}(\epsilon^{-2})$ for the gradient descent type algorithm when $\nabla\varphi(\cdot)$ is Lipschitz continuous, and $\mathcal{O}(\epsilon^{-1.75})$ for the accelerated gradient descent methods if $\nabla^2\varphi(\cdot)$ is Lipschitz continuous.

If we let the Lagrange function as $\mathcal{L}_\lambda(\boldsymbol{x},\boldsymbol{y}):= f(\boldsymbol{x},\boldsymbol{y}) + \lambda\left(g(\boldsymbol{x},\boldsymbol{y}) - g\left(\boldsymbol{x},\boldsymbol{y}^*(\boldsymbol{x})\right)\right)$, Kwon et al. (2023b) shows that (1) can be effectively solved by the following formulation:

$$\min_{\boldsymbol{x}\in\mathbb{R}^{d_x}} \mathcal{L}_\lambda^*(\boldsymbol{x}):= \mathcal{L}_\lambda\left(\boldsymbol{x},\boldsymbol{y}_\lambda^*(\boldsymbol{x})\right),$$
$$\text{where} \quad \boldsymbol{y}_\lambda^*(\boldsymbol{x}):=\arg\min_{\boldsymbol{y}\in\mathbb{R}^{d_y}} \mathcal{L}_\lambda(\boldsymbol{x},\boldsymbol{y}). \tag{2}$$

The gradient $\nabla\mathcal{L}_\lambda^*(\mathbf{x})$ can be expressed as:

$$\nabla\mathcal{L}_\lambda^*(\boldsymbol{x}) = \nabla_x f(\boldsymbol{x},\boldsymbol{y}_\lambda^*(\boldsymbol{x}))$$
$$+ \lambda\left(\nabla_x g(\boldsymbol{x},\boldsymbol{y}_\lambda^*(\boldsymbol{x})) - \nabla_x g(\boldsymbol{x},\boldsymbol{y}^*(\boldsymbol{x}))\right). \tag{3}$$

which can be approximated by using only first-order oracles of $f$ and $g$. Kwon et al. (2023b) proposed F$^2$BA, which per-

*Table 1.* Comparison of computational complexities for finding an $\epsilon$-stationary point of the hyper-objective $\varphi(\boldsymbol{x}) := f(\boldsymbol{x}, \boldsymbol{y}^*(\boldsymbol{x}))$ under Assumption 2.1. **Finding SOSP** indicates whether the algorithm can find an approximate second-order stationary point.

| Oracle | Method | Iteration Complexity | Hessian Computations | Find SOSP? |
|---|---|---|---|---|
| 1st $(f, g)$ | PZOBO (Sow et al., 2022b) | $\tilde{\mathcal{O}}(d_x^2 \kappa^6 \epsilon^{-4})$ [(a)] | - | ✗ |
| | BOME (Liu et al., 2022) | $\tilde{\mathcal{O}}(\text{poly}(\kappa)\epsilon^{-6})$ [(b)] | - | ✗ |
| | F$^2$BA (Kwon et al., 2023b) | $\tilde{\mathcal{O}}(\kappa^7 \epsilon^{-3})$ [(c)] | - | ✗ |
| | F$^2$BA (Chen et al., 2025b) | $\tilde{\mathcal{O}}(\kappa^4 \epsilon^{-2})$ | - | ✓ |
| | RAF$^2$BA (Chen et al., 2025b; Yang et al., 2023) | $\tilde{\mathcal{O}}(\kappa^{3.75} \epsilon^{-1.75})$ | - | ✓ |
| 1st $(f)$ + 2nd $(g)$ | AID (Ghadimi & Wang, 2018) | $\mathcal{O}(\kappa^5 \epsilon^{-2.5})$ | every iteration [†] | ✗ |
| | AID-BiO (Ji et al., 2021) | $\mathcal{O}(\kappa^4 \epsilon^{-2})$ | every iteration [†] | ✗ |
| | ITD-BiO (Ji et al., 2021) | $\tilde{\mathcal{O}}(\kappa^4 \epsilon^{-2})$ | -[*] | ✗ |
| | iNEON (Huang et al., 2025) | $\tilde{\mathcal{O}}(\text{poly}(\kappa)\epsilon^{-2})$ | every iteration [†] | ✓ |
| | RAHGD (Yang et al., 2023) | $\tilde{\mathcal{O}}(\kappa^{3.25} \epsilon^{-1.75})$ | every iteration [†] | ✓ |
| | IAPUN (Wang et al., 2024a) | $\tilde{\mathcal{O}}(\kappa^{4.75} \epsilon^{-1.75})$ | every iteration | ✓ |
| 2nd $(f, g)$ | FSBA (Alg. 2) | $\tilde{\mathcal{O}}(\kappa^3 \epsilon^{-1.5})$ | every iteration | ✓ |
| | LFSBA (Alg. 4) | $\tilde{\mathcal{O}}(\kappa^3 m^{0.5} \epsilon^{-1.5})$ | once every $m$ iterations | ✓ |

**Note:** (a) Assumes $\|\nabla^2 g(\boldsymbol{x}, \boldsymbol{y}) - \nabla^2 g(\boldsymbol{x}', \boldsymbol{y}')\|_F^2 \le \rho_g^2(\|\boldsymbol{x} - \boldsymbol{x}'\|^2 + \|\boldsymbol{y} - \boldsymbol{y}'\|^2)$, stronger than Assumption 3.1c. (b) Additionally assumes both $|f(\boldsymbol{x}, \boldsymbol{y})|$ and $|g(\boldsymbol{x}, \boldsymbol{y})|$ are upper bounded. (c) Additionally assumes $\nabla^2 f$ is Lipschitz and gradients are bounded.
[†] Each iteration requires computing $(\nabla_{yy}^2 g(\boldsymbol{x}, \boldsymbol{y}))^{-1} \boldsymbol{v}$, typically via Conjugate Gradient (CG).
[*] The algorithm does not explicitly construct the Hessian, though the analytic form involves 2nd-order derivatives.

forms the inexact gradient descent on $\mathcal{L}_\lambda^*(\cdot)$. They showed that F$^2$BA can find the $\epsilon$-stationary point of $\varphi(\cdot)$ within $\mathcal{O}(\epsilon^{-3})$ calls to the gradient of $f$ and $g$. Later, Chen et al. (2025b) improved the iteration complexities of F$^2$BA to $\tilde{\mathcal{O}}(\epsilon^{-2})$ and showed that this complexity could be further enhanced to $\tilde{\mathcal{O}}(\epsilon^{-1.75})$ if $\nabla^2 \varphi(\cdot)$ is Lipschitz continuous. It is important to note that these rates (nearly) match those of the second-order methods for bilevel optimization mentioned earlier, while requiring only first-order oracles. Given these results, we find the advantages of accessing second-order oracles in bilevel optimization appear to be rather limited. Thus, it is a natural question to ask:

*"Can we develop a second-order method with improved iteration complexities that demonstrates the benefits of utilizing second-order oracles in bilevel optimization?"*

To achieve better iteration complexity, a straightforward approach is to employ an inexact Newton-type algorithm that utilizes not only the hypergradient $\nabla\varphi(\boldsymbol{x})$, but also the hyperHessian $\nabla^2\varphi(\boldsymbol{x})$. However, accessing $\nabla^2\varphi(\boldsymbol{x})$ requires the third-order derivatives of $g$, which is too expensive or even not feasible. In this work, we adopt the idea of fully first-order bilevel approximation methods to tackle the bilevel optimization problem (1) via its approximation (2). We design a fully second-order bilevel approximation (FSBA) method that estimates both $\nabla\mathcal{L}_\lambda^*(\boldsymbol{x})$ and $\nabla^2\mathcal{L}_\lambda^*(\boldsymbol{x})$, performing inexact cubic-regularized Newton iterations to find the stationary point of $\mathcal{L}_\lambda^*(\cdot)$. We prove that our proposed FSBA method requires only $\tilde{\mathcal{O}}(\epsilon^{-1.5})$ oracles to the gradient and Hessian of $f$ and $g$, which surpass

the performance of the state-of-the-art second-order methods for bilevel optimization. Since nonconvex-strongly-concave bilevel optimization problems subsume nonconvex optimization problems, the oracle complexity of FSBA is near-optimal, aligning with the lower bound $\Omega(\epsilon^{-1.5})$ by Carmon et al. (2020).

Considering the heavy computational cost of second-order oracles, existing second-order methods often use iterative techniques to compute the product of the Hessian inverse and the gradient, or they apply approximation algorithms to reduce the computational burden (Ghadimi & Wang, 2018; Yang et al., 2025). More recently, quasi-Newton methods have been developed to efficiently solve the lower-level problem (Fang et al., 2025). Therefore, it is also crucial to reduce the computational complexity of the FSBA method.

To address this, we leverage the concept of lazy Hessians (Doikov et al., 2023; Doikov & Grapiglia, 2023) and propose a lazy fully second-order bilevel approximation method (LFSBA). Instead of computing the approximate Hessian at each iteration, as is common in the existing literature, LFSBA estimates $\nabla^2\mathcal{L}_\lambda^*(\cdot)$ only at specific snapshot points and reuses this approximate Hessian for the next $m$ iterations. We demonstrate that LFSBA achieves an iteration complexity of $\tilde{\mathcal{O}}(\sqrt{m}\epsilon^{-1.5})$, providing a better computational complexity than FSBA by properly tuning $m$. We compare FSBA and LFSBA with existing first- and second-order methods in Table 1.

**Paper organization.** In Section 2, we introduce the notation and assumptions. In Section 3, we propose the FSBA method and study its convergence behavior. In Section 4, we propose the lazy variant of FSBA method (LFSBA) and show that it has a better computational complexity than FSBA. We conduct experiments to validate our theoretical results in Section 5 and summarize this paper in Section 6. All proofs are deferred to the Appendix.

## 2. Preliminaries

We first introduce some basic notation. For a twice differentiable function $f(\boldsymbol{x}, \boldsymbol{y})$, its partial gradients with respect to $\boldsymbol{x}$ and $\boldsymbol{y}$ are denoted by $\nabla_x f(\boldsymbol{x}, \boldsymbol{y})$ and $\nabla_y f(\boldsymbol{x}, \boldsymbol{y})$, respectively. The Hessian matrix of $f$ at point $(\boldsymbol{x}, \boldsymbol{y})$ is partitioned as $\nabla^2 f(\boldsymbol{x}, \boldsymbol{y}) = \begin{bmatrix} \nabla^2_{xx} f(\boldsymbol{x}, \boldsymbol{y}), \nabla^2_{xy} f(\boldsymbol{x}, \boldsymbol{y}) \\ \nabla^2_{yx} f(\boldsymbol{x}, \boldsymbol{y}), \nabla^2_{yy} f(\boldsymbol{x}, \boldsymbol{y}) \end{bmatrix}$. We use $\|\cdot\|$ to denote the spectral norm of matrices and the Euclidean norm of vectors, respectively. Given a real symmetric matrix $\mathbf{A}$, we let $\lambda_{\min}(\mathbf{A})$ denote its smallest eigenvalue. Furthermore, we denote the following quantity, for any $\boldsymbol{x} \in \mathbb{R}^{d_x}$, $\xi(\boldsymbol{x}) := \left[ -\lambda_{\min}\left( \nabla^2 f(\boldsymbol{x}) \right) \right]_+$, where $[t]_+ := \max\{t, 0\}$ denotes the positive part. Finally, we use $\mathcal{O}(\cdot)$ to hide only absolute constants that are independent of any problem parameters, $\tilde{\mathcal{O}}(\cdot)$ to additionally hide polylogarithmic factors, and $\Omega(\cdot)$ to denote an asymptotic lower bound up to a constant factor.

We now introduce the following assumptions on the upper-level function $f$ and the lower-level function $g$.

**Assumption 2.1.** Suppose that the upper-level function $f$, the lower-level function $g$ and the hyper-objective $\varphi$ satisfy the following conditions:

(a) $g(\boldsymbol{x}, \boldsymbol{y})$ is three times differentiable and $\mu$-strongly convex with respect to $\boldsymbol{y}$ for any fixed $\boldsymbol{x}$;

(b) $f(\boldsymbol{x}, \boldsymbol{y})$ is twice differentiable and $C$-Lipschitz with respect to $\boldsymbol{x}$ and $\boldsymbol{y}$;

(c) $g(\boldsymbol{x}, \boldsymbol{y})$ and $f(\boldsymbol{x}, \boldsymbol{y})$ are $\ell$-gradient Lipschitz with respect to $\boldsymbol{x}$ and $\boldsymbol{y}$;

(d) $f(\boldsymbol{x}, \boldsymbol{y})$ and $g(\boldsymbol{x}, \boldsymbol{y})$ are $\rho$-Hessian Lipschitz with respect to $\boldsymbol{x}$ and $\boldsymbol{y}$;

(e) $g(\boldsymbol{x}, \boldsymbol{y})$ is $\nu$-third-order derivative Lipschitz with respect to $\boldsymbol{x}$ and $\boldsymbol{y}$;

(f) $\varphi(\boldsymbol{x})$ is lower bounded, i.e. $\varphi^* := \min_{\boldsymbol{x} \in \mathbb{R}^{d_x}} \varphi(\boldsymbol{x}) > -\infty$.

These assumptions are common and standard for the nonconvex strongly-convex bilevel optimization (Yang et al., 2023; Chen et al., 2025b; Kwon et al., 2023b; Ghadimi & Wang, 2018). We define the condition number as follows.

**Definition 2.2.** We define the largest smoothness constant $\bar{\ell} := \max\{C, \ell, \nu, \rho\}$ and the condition number $\kappa := \bar{\ell}/\mu$.

Assumption 2.1 means that $\boldsymbol{y}^*(\mathbf{x})$ is Lipschitz continuous, as presented in the following proposition.

**Proposition 2.3** (Lemma 2.2, Ghadimi & Wang (2018)). *Under Assumption 2.1, $\boldsymbol{y}^*(\cdot)$ is $\kappa$-Lipschitz continuous.*

Furthermore, when $\lambda$ is large enough, we also have that $\mathcal{L}^*_\lambda(\cdot)$ is a good proxy of $\varphi(\cdot)$ and that $\nabla \mathcal{L}^*_\lambda(\cdot)$, $\nabla^2 \mathcal{L}^*_\lambda(\cdot)$ are Lipschitz continuous.

**Proposition 2.4** (Lemma 4.1, Lemma 5.1 in Chen et al. (2025b), and Lemma 3.2 in Chen et al. (2025a); Lemma 3.1 in Kwon et al. (2023b)). *Under Assumption 2.1 and let $\lambda \geq 2\ell/\mu$, we have:*

(a) $|\mathcal{L}^*_\lambda(\boldsymbol{x}) - \varphi(\boldsymbol{x})| = \mathcal{O}(\bar{\ell}\kappa^2/\lambda)$, $\|\nabla \mathcal{L}^*_\lambda(\boldsymbol{x}) - \nabla \varphi(\boldsymbol{x})\| = \mathcal{O}(\bar{\ell}\kappa^3/\lambda)$, *and* $\|\nabla^2 \mathcal{L}^*_\lambda(\boldsymbol{x}) - \nabla^2 \varphi(\boldsymbol{x})\| = \mathcal{O}(\bar{\ell}\kappa^5/\lambda)$ *hold for all* $\mathbf{x} \in \mathbb{R}^{d_x}$.

(b) $\nabla \mathcal{L}^*_\lambda(\boldsymbol{x})$ *is $L$-Lipschitz continuous, where* $L := \mathcal{O}(\bar{\ell}\kappa^3)$.

(c) $\nabla^2 \mathcal{L}^*_\lambda(\boldsymbol{x})$ *is $\bar{\rho}$-Lipschitz continuous, where* $\bar{\rho} := \mathcal{O}(\bar{\ell}\kappa^5)$.

In addition, $\mathcal{L}_\lambda(\boldsymbol{x}, \cdot)$ is strongly convex in $\boldsymbol{y}$ when $\lambda$ is large enough.

**Proposition 2.5** (Lemma 3.2, Kwon et al. (2023b)). *Under Assumption 2.1, if $\lambda \geq 2\ell/\mu$, then $\mathcal{L}_\lambda(\boldsymbol{x}, \cdot)$ is $(\lambda\mu/2)$-strongly convex, $(1 + \lambda)\ell$-smooth. The condition number of $\mathcal{L}_\lambda(\boldsymbol{x}, \cdot)$ is $3\kappa$.*

Finally, we give the formal definition of the $\epsilon$-first-order stationary points and $(\epsilon, \tau)$-second-order stationary points as follows.

**Definition 2.6.** We call $\hat{\boldsymbol{x}}$ an $\epsilon$-first-order stationary point (FOSP) of $\varphi(\boldsymbol{x})$ if $\|\nabla \varphi(\hat{\boldsymbol{x}})\| \leq \epsilon$.

**Definition 2.7.** We call $\hat{\boldsymbol{x}}$ an $(\epsilon, \tau)$-second-order stationary point (SOSP) of $\varphi(\boldsymbol{x})$ if $\|\nabla \varphi(\hat{\boldsymbol{x}})\| \leq \epsilon$ and $\lambda_{\min}\left( \nabla^2 \varphi(\hat{\boldsymbol{x}}) \right) \geq -\tau$.

## 3. Fully Second-Order Bilevel Approximation Method

In this section, we introduce our fully second-order bilevel approximation method (FSBA) and present its convergence analysis. We also introduce an inexact variant of second-order method for practical consideration.

### 3.1. The FSBA Method

Proposition 2.4 shows that $\mathcal{L}^*_\lambda(\cdot)$ defined in (2) is a good approximation of $\varphi(\cdot)$, and that $(\epsilon, \mathcal{O}(\sqrt{\epsilon}))$-SOSP $\mathcal{L}^*_\lambda(\mathbf{x})$ is also an $(\mathcal{O}(\epsilon), \mathcal{O}(\sqrt{\epsilon}))$-SOSP of $\varphi(\cdot)$. Hence, the main idea

of our FSBA method is to apply second-order methods to solve the proxy function $\mathcal{L}_\lambda^*(\cdot)$ instead of $\varphi(\cdot)$.

The Hessian of $\mathcal{L}_\lambda^*(\boldsymbol{x})$ can be expressed by

$$\nabla^2 \mathcal{L}_\lambda^*(\boldsymbol{x}) = \nabla_{xx}^2 \mathcal{L}_\lambda(\boldsymbol{x}, \boldsymbol{y}_\lambda^*(\boldsymbol{x}))$$
$$\nabla_{xy}^2 \mathcal{L}_\lambda(\boldsymbol{x}, \boldsymbol{y}_\lambda^*(\boldsymbol{x})) \left[\nabla_{yy}^2 \mathcal{L}_\lambda(\boldsymbol{x}, \boldsymbol{y}_\lambda^*(\boldsymbol{x}))\right]^{-1} \nabla_{yx}^2 \mathcal{L}_\lambda(\boldsymbol{x}, \boldsymbol{y}_\lambda^*(\boldsymbol{x}))$$

where the Hessian block of $\mathcal{L}_\lambda(\mathbf{x}, \mathbf{y})$ can be computed individually by only second-order information of $f$ and $g$. Ideally, using $\nabla \mathcal{L}_\lambda^*(\boldsymbol{x})$ and $\nabla^2 \mathcal{L}_\lambda^*(\boldsymbol{x})$ to construct the cubic-regularized Newton (CRN) method on $\mathcal{L}^*(\cdot)$ with the following update direction

$$\boldsymbol{s}^* = \arg\min_{\boldsymbol{s} \in \mathbb{R}^{d_x}} \left\{\boldsymbol{s}^\top \nabla \mathcal{L}_\lambda^*(\boldsymbol{x}) + \frac{\boldsymbol{s}^\top \nabla^2 \mathcal{L}_\lambda^*(\boldsymbol{x})\boldsymbol{s}}{2} + \frac{M\|\boldsymbol{s}\|^3}{6}\right\} \quad (4)$$

can find an $(\epsilon, \mathcal{O}(\sqrt{\epsilon}))$-SOSP of $\mathcal{L}_\lambda^*(\cdot)$ within $\mathcal{O}(\epsilon^{-1.5})$ iterations. However, $\nabla \mathcal{L}_\lambda^*(\boldsymbol{x})$ and $\nabla^2 \mathcal{L}_\lambda^*(\boldsymbol{x})$ cannot be computed exactly since they contain $\boldsymbol{y}_\lambda^*(\boldsymbol{x})$ and $\boldsymbol{y}^*(\boldsymbol{x})$. The following lemma shows that $\nabla \mathcal{L}_\lambda^*(\boldsymbol{x})$ and $\nabla^2 \mathcal{L}_\lambda^*(\boldsymbol{x})$ can be estimated by introducing additional variables $\boldsymbol{y}$ and $\boldsymbol{w}$.

**Lemma 3.1.** *Under Assumption 2.1 and let*

$$\boldsymbol{g}(\boldsymbol{x}; \boldsymbol{y}, \boldsymbol{w}) := \nabla_x f(\boldsymbol{x}, \boldsymbol{y}) + \lambda(\nabla_x g(\boldsymbol{x}, \boldsymbol{y}) - \nabla_x g(\boldsymbol{x}, \boldsymbol{w})) \quad (5)$$
$$\mathbf{H}(\boldsymbol{x}; \boldsymbol{y}, \boldsymbol{w}) :=$$
$$\nabla_{xx}^2 f(\boldsymbol{x}, \boldsymbol{y}) + \lambda(\nabla_{xx}^2 g(\boldsymbol{x}, \boldsymbol{y}) - \nabla_{xx}^2 g(\boldsymbol{x}, \boldsymbol{w}))$$
$$- \nabla_{xy}^2 \mathcal{L}_\lambda(\boldsymbol{x}, \boldsymbol{y}) \left[\nabla_{yy}^2 \mathcal{L}_\lambda(\boldsymbol{x}, \boldsymbol{y})\right]^{-1} \nabla_{yx}^2 \mathcal{L}_\lambda(\boldsymbol{x}, \boldsymbol{y}) \qquad (6)$$
$$+ \lambda \nabla_{xy}^2 g(\boldsymbol{x}, \boldsymbol{w}) \left[\nabla_{yy}^2 g(\boldsymbol{x}, \boldsymbol{w})\right]^{-1} \nabla_{yx}^2 g(\boldsymbol{x}, \boldsymbol{w}),$$

*then we have*

$$\|\nabla \mathcal{L}_\lambda^*(\boldsymbol{x}) - \mathbf{g}(\boldsymbol{x}; \boldsymbol{y}, \boldsymbol{w})\| \leq 2\lambda\ell \|\boldsymbol{y} - \boldsymbol{y}_\lambda^*(\boldsymbol{x})\| + \lambda\ell\|\boldsymbol{w} - \boldsymbol{y}^*(\boldsymbol{x})\|,$$
$$\|\nabla^2 \mathcal{L}_\lambda^*(\boldsymbol{x}) - \mathbf{H}(\boldsymbol{x}; \boldsymbol{y}, \boldsymbol{w})\| \leq C_1\|\boldsymbol{w} - \boldsymbol{y}^*(\boldsymbol{x})\| + C_2\|\boldsymbol{y} - \boldsymbol{y}_\lambda^*(\boldsymbol{x})\|,$$

*where $C_1 := \mathcal{O}(\lambda\bar{\ell} + \bar{\ell}\kappa^2)$ and $C_2 := \mathcal{O}(\lambda\bar{\ell}\kappa^2)$.*

Since $\mathcal{L}_\lambda(\boldsymbol{x}, \cdot)$ and $g(\boldsymbol{x}, \cdot)$ are strongly convex according to Proposition 2.5 and Assumption 2.1 (a), it will be easy to find $\boldsymbol{y} \approx \boldsymbol{y}_\lambda^*(\boldsymbol{x})$ and $\boldsymbol{w} \approx \boldsymbol{y}^*(\boldsymbol{x})$ by applying the proper first- or second-order method for strongly convex optimization, i.e., the accelerated gradient descent method (AGD, Algorithm 1), on $\mathcal{L}_\lambda(\boldsymbol{x}, \cdot)$ and $g(\boldsymbol{x}, \cdot)$, respectively. This leads to $\boldsymbol{g}(\boldsymbol{x}; \boldsymbol{y}, \boldsymbol{w}) \approx \nabla \mathcal{L}_\lambda^*(\boldsymbol{x})$ and $\mathbf{H}(\boldsymbol{x}; \boldsymbol{y}, \boldsymbol{w}) \approx \nabla^2 \mathcal{L}_\lambda^*(\boldsymbol{x})$ by Lemma 3.1. Replacing $\nabla \mathcal{L}_\lambda^*(\boldsymbol{x})$ and $\nabla^2 \mathcal{L}_\lambda^*(\boldsymbol{x})$ by $\boldsymbol{g}(\boldsymbol{x}; \boldsymbol{y}, \boldsymbol{w})$ and $\mathbf{H}(\boldsymbol{x}; \boldsymbol{y}, \boldsymbol{w})$ in the CRN update (4) leads to our FSBA method, as presented in Algorithm 2.

### 3.2. Convergence Analysis of FSBA

In this section, we provide the convergence analysis of FSBA. The following lemma shows that once $\mathbf{g}(\boldsymbol{x}_t; \boldsymbol{y}_t, \boldsymbol{w}_t)$ and $\mathbf{H}(\boldsymbol{x}_t; \boldsymbol{y}_t, \boldsymbol{w}_t)$ are close enough to $\nabla \mathcal{L}_\lambda^*(\boldsymbol{x}_t)$ and $\nabla^2 \mathcal{L}_\lambda^*(\boldsymbol{x}_t)$, FSBA enjoys a similar convergence rate as applying the exact CRN method (4).

---

**Algorithm 1** AGD $(h(\cdot), \boldsymbol{z}_0, K, \eta, \theta)$

1: **Input:** $\tilde{\boldsymbol{z}}_0 = \boldsymbol{z}_0$
2: **for** $k = 0$ to $K - 1$ **do**
3: $\quad \boldsymbol{z}_{k+1} = \tilde{\boldsymbol{z}}_k - \eta \nabla h(\tilde{\boldsymbol{z}}_k)$
4: $\quad \tilde{\boldsymbol{z}}_{k+1} = \boldsymbol{z}_{t+1} + \theta(\boldsymbol{z}_{k+1} - \boldsymbol{z}_k)$
5: **end for**
6: **Output:** $\boldsymbol{z}_K$

---

**Algorithm 2** Fully Second-Order Bilevel Approximation method (FSBA)

1: **Input:** $\boldsymbol{x}_0 \in \mathbb{R}^{d_x}$, $\boldsymbol{y}_{-1} = \mathbf{0}$, $\boldsymbol{w}_{-1} = \mathbf{0}$, $T$, $\ell_1$, $\ell_2$, $\kappa_1$, $\kappa_2$, $\epsilon$, $M$, $\{K_t^1\}_{t=0}^T$, $\{K_t^2\}_{t=0}^T$
2: **for** $t = 0, 1, \cdots, T - 1$ **do**
3: $\quad \boldsymbol{w}_t = \text{AGD}\left(g(\boldsymbol{x}_t, \cdot), \boldsymbol{w}_{t-1}, K_t^1, \frac{1}{\ell_1}, \frac{\sqrt{\kappa_1} - 1}{\sqrt{\kappa_1} + 1}\right)$
4: $\quad \boldsymbol{y}_t = \text{AGD}\left(\mathcal{L}_\lambda(\boldsymbol{x}_t, \cdot), \boldsymbol{y}_{t-1}, K_t^2, \frac{1}{\ell_2}, \frac{\sqrt{\kappa_2} - 1}{\sqrt{\kappa_2} + 1}\right)$
5: $\quad$ Compute $\mathbf{g}_t = \mathbf{g}(\boldsymbol{x}_t; \boldsymbol{y}_t, \boldsymbol{w}_t)$ according to (5).
6: $\quad$ Compute $\mathbf{H}_t = \mathbf{H}(\boldsymbol{x}_t; \boldsymbol{y}_t, \boldsymbol{w}_t)$ according to (6).
7: $\quad \boldsymbol{s}_t^* = \arg\min_{\boldsymbol{s} \in \mathbb{R}^{d_x}} \left\{\mathbf{g}_t^\top \boldsymbol{s} + \frac{1}{2}\boldsymbol{s}^\top \mathbf{H}_t \boldsymbol{s} + \frac{M}{6}\|\boldsymbol{s}\|^3\right\}$
8: $\quad$ **If** $\|\boldsymbol{s}_t^*\| \leq \frac{1}{2}\sqrt{\epsilon/M}$ **then break**
9: $\quad \boldsymbol{x}_{t+1} = \boldsymbol{x}_t + \boldsymbol{s}_t^*$
10: **end for**
11: **Output:** $\hat{\boldsymbol{x}} = \boldsymbol{x}_{t+1}$

---

**Lemma 3.2** (Theorem 1, Luo et al. (2022)). *Under Assumption 2.1, if we run Algorithm 2 with $M \geq \bar{\rho}$ and $T = \lceil 192(\mathcal{L}_\lambda^*(\boldsymbol{x}_0) - \min_{\boldsymbol{x}} \mathcal{L}_\lambda^*(\boldsymbol{x}))\rceil \sqrt{M}\epsilon^{-3/2}$, and suppose the following condition*

$$\|\nabla \mathcal{L}_\lambda^*(\boldsymbol{x}_t) - \mathbf{g}(\boldsymbol{x}_t; \boldsymbol{y}_t, \boldsymbol{w}_t)\| \leq C_g\epsilon,$$
$$\|\nabla^2 \mathcal{L}_\lambda^*(\boldsymbol{x}_t) - \mathbf{H}(\boldsymbol{x}_t; \boldsymbol{y}_t, \boldsymbol{w}_t)\| \leq C_H\sqrt{M}\epsilon, \qquad (7)$$

*hold with $C_g := 1/192$ and $C_H := 1/48$, then $\hat{\boldsymbol{x}}$ is an $(\epsilon, \sqrt{M\epsilon})$-SOSP of $\mathcal{L}_\lambda^*(\cdot)$.*

In the following lemma, we show that the above condition can be achieved by properly choosing the iteration numbers $K_t^1$ and $K_t^2$ in the AGD subroutine.

**Lemma 3.3.** *Under Assumption 2.1, let $\Delta = \varphi(\boldsymbol{x}_0) - \varphi^*$, $\tilde{\epsilon} = \min\left\{\frac{C_g\epsilon}{4\lambda\ell}, \frac{C_H\sqrt{M\epsilon}}{2C_2}\right\}$, $R = \max\{\|\boldsymbol{y}^*(\boldsymbol{x}_0)\|, \|\boldsymbol{y}_\lambda^*(\boldsymbol{x}_0)\|\}$, if we run Algorithm 2 with $M \geq \bar{\rho}$, $\kappa_1 = \kappa$, $\ell_1 = \ell$, $\kappa_2 = 3\kappa$, $\ell_2 = (1 + \lambda)\ell$, $\lambda = \max\left\{\bar{\ell}\kappa^2/\Delta, \bar{\ell}\kappa^3/\epsilon, \bar{\ell}\kappa^5/\sqrt{M\epsilon}\right\}$, and*

$$K_t^1 = K_t^2 = \begin{cases} \left\lceil 2\sqrt{\kappa_2}\log\left(\frac{\sqrt{\kappa_2}+1}{\tilde{\epsilon}}R\right)\right\rceil & t = 0 \\ \left\lceil 2\sqrt{\kappa_2}\log\left(\frac{\sqrt{\kappa_2}+1}{\tilde{\epsilon}}(\tilde{\epsilon} + 4\kappa\|\boldsymbol{s}_{t-1}^*\|)\right)\right\rceil & t \geq 1 \end{cases},$$

*then the condition (7) in Lemma 3.2 is satisfied.*

Combining Lemma 3.2 and 3.3, we know that FSBA can find $(\epsilon, \sqrt{M\epsilon})$-SOSP of $\mathcal{L}_\lambda^*(\cdot)$ with iteration complexities

of $\tilde{\mathcal{O}}(\epsilon^{-1.5})$. In the following theorem, we formally present the first- and second-order oracle complexities of FSBA to find the SOSP of $\varphi(\cdot)$.

**Theorem 3.4.** *Under Assumption 2.1, run Algorithm 2 with the same setting as Lemma 3.3, let $M = \Omega(\bar{\rho})$ and $T = \Theta((\varphi(\boldsymbol{x}_0) - \varphi^*)\sqrt{M}\epsilon^{-3/2})$, then $\hat{\boldsymbol{x}}$ is an $((\mathcal{O}(\epsilon), \mathcal{O}(\kappa^{2.5}\bar{\ell}^{0.5}\epsilon^{0.5}))$-SOSP of $\varphi(\cdot)$. In addition, the complexities of the first- and second-order oracle can be bounded by $\tilde{\mathcal{O}}(\kappa^3\bar{\ell}^{0.5}\epsilon^{-1.5})$ and $\mathcal{O}(\kappa^{2.5}\bar{\ell}^{0.5}\epsilon^{-1.5})$, respectively.*

### 3.3. An Inexact Version of FSBA

Both the computation of $\mathbf{H}(\boldsymbol{x}_t; \boldsymbol{y}_t, \boldsymbol{w}_t)$ (line 6) and solving the cubic-regularized subproblem (line 7) in Algorithm 2 require the explicit construction of Hessian and the inverse of regularized Hessian, which may limit the application of FSBA in large-scale problems when the problem dimension is extremely large.

In this section, we propose an inexact variant of FSBA. Instead of accessing $\mathbf{H}(\boldsymbol{x}_t; \boldsymbol{y}_t, \boldsymbol{w}_t)$ directly according to (6), we compute $\mathbf{C}(\boldsymbol{x}_t; \boldsymbol{y}_t, \boldsymbol{w}_t)$

$$
\begin{aligned}
&\mathbf{C}(\boldsymbol{x}_t; \boldsymbol{y}_t, \boldsymbol{w}_t) \\
&:= \nabla^2_{xx}f(\boldsymbol{x}_t, \boldsymbol{y}_t) + \lambda\big(\nabla^2_{xx}g(\boldsymbol{x}_t, \boldsymbol{y}_t) - \nabla^2_{xx}g(\boldsymbol{x}_t, \boldsymbol{w}_t)\big) \\
&\quad + \lambda\nabla^2_{xy}g(\boldsymbol{x}_t, \boldsymbol{w}_t)\,\mathbf{C}_{1,t}\nabla^2_{yx}g(\boldsymbol{x}_t, \boldsymbol{w}_t) \\
&\quad - \nabla^2_{xy}\mathcal{L}_\lambda(\boldsymbol{x}_t, \boldsymbol{y}_t)\,\mathbf{C}_{2,t}\nabla^2_{yx}\mathcal{L}_\lambda(\boldsymbol{x}_t, \boldsymbol{y}_t),
\end{aligned}
\tag{8}
$$

which replaces $\nabla^2_{yy}g(\boldsymbol{x}_t, \boldsymbol{w}_t)^{-1}$ and $\nabla^2_{yy}\mathcal{L}_\lambda(\boldsymbol{x}_t, \boldsymbol{y}_t)^{-1}$ by their Chebyshev Polynomials approximations $\mathbf{C}_{1,t}$ and $\mathbf{C}_{2,t}$. In addition, we do not solve the cubic subproblem by regularized Newton step, but instead, using a gradient-type method to approximately solve

$$
\min_{\boldsymbol{s}\in\mathbb{R}^{d_x}} m(\boldsymbol{s}) = \boldsymbol{s}^\top\mathbf{g}_t + \frac{1}{2}\boldsymbol{s}^\top\mathbf{C}_t\boldsymbol{s} + \frac{M}{6}\|\boldsymbol{s}\|^3,
$$

whose gradient $\nabla m(\boldsymbol{s}) = \mathbf{g}_t + \mathbf{C}_t\boldsymbol{s} + \frac{M}{2}\|\boldsymbol{s}\|\boldsymbol{s}$ can be computed with only gradients and Hessian-vector products of $f$ and $g$. We present the inexact fully second-order bilevel approximation method (ISFBA) in Algorithm 3. The detailed implementation of constructing $\mathbf{C}_{1,t}$, $\mathbf{C}_{2,t}$ in $\mathbf{C}_t$ (line 6) and the sub-problem solvers (line 7 and line 10) are presented in Appendix F.

## 4. Lazy Fully Second-Order Bilevel Approximation Algorithm with Better Computational Complexity

In the previous section, we propose FSBA with $\tilde{\mathcal{O}}(\epsilon^{-1.5})$ oracle complexity for non-convex strongly convex bilevel optimization, which is faster than existing first-order methods. However, the second-order oracle always leads to a

---

**Algorithm 3** Inexact Fully Second-Order Bilevel Approximation method (IFSBA)

1: **Input:** $\boldsymbol{x}_0 \in \mathbb{R}^{d_x}, \boldsymbol{y}_{-1} = \mathbf{0}, \ell_1, \ell_2, \kappa_1, \kappa_2, \epsilon, M,$
   $\boldsymbol{w}_{-1} = \mathbf{0}, T, \{K_t^1\}_{t=0}^T, \{K_t^2\}_{t=0}^T,$
2: **for** $t = 0, 1, \cdots, T-1$ **do**
3: $\quad \boldsymbol{w}_t = \text{AGD}\left(g(\boldsymbol{x}_t, \cdot), \boldsymbol{w}_{t-1}, K_t^1, \frac{1}{\ell_1}, \frac{\sqrt{\kappa_1}-1}{\sqrt{\kappa_1}+1}\right)$
4: $\quad \boldsymbol{y}_t = \text{AGD}\left(\mathcal{L}_\lambda(\boldsymbol{x}_t, \cdot), \boldsymbol{y}_{t-1}, K_t^2, \frac{1}{\ell_2}, \frac{\sqrt{\kappa_2}-1}{\sqrt{\kappa_2}+1}\right)$
5: $\quad$ Compute $\mathbf{g}_t = \mathbf{g}(\boldsymbol{x}_t; \boldsymbol{y}_t, \boldsymbol{w}_t)$ according to (5)
6: $\quad$ Compute $\mathbf{C}_t = \mathbf{C}(\boldsymbol{x}_t; \boldsymbol{y}_t, \boldsymbol{w}_t)$ according to (8).
7: $\quad (\boldsymbol{s}_t, \Delta_t) = \text{Cubic-Solver}(\mathbf{g}_t, \mathbf{C}_t, \sigma, \mathcal{K}(\epsilon, \delta'))$
8: $\quad \boldsymbol{x}_{t+1} = \boldsymbol{x}_t + \boldsymbol{s}_t$
9: $\quad$ **If** $\Delta_t > -\frac{\epsilon^3}{128M}$ **then**
10: $\quad\quad \hat{\boldsymbol{s}} = \text{Final-Cubic-Solver}(\mathbf{g}_t, \mathbf{C}_t, \epsilon)$
11: $\quad\quad \boldsymbol{x}_{t+1} = \boldsymbol{x}_t + \hat{\boldsymbol{s}}$
12: $\quad\quad$ **break**
13: $\quad$ **end If**
14: **end for**
15: **Output:** $\hat{\boldsymbol{x}} = \boldsymbol{x}_{t+1}$

---

heavier computational complexity than the first-order oracle. We make the following assumption to differentiate the computational complexity of first- and second-order oracles by following Doikov et al. (2023).

**Assumption 4.1.** We count the computational complexity of first-order oracle of $f$ and $g$, i.e., $\nabla_x f(\boldsymbol{x}, \boldsymbol{y}), \nabla_y f(\boldsymbol{x}, \boldsymbol{y}),$ $\nabla_x g(\boldsymbol{x}, \boldsymbol{y}), \nabla_y g(\boldsymbol{x}, \boldsymbol{y})$ and HVPs computed via automatic differentiation, as $N$. We count the computational complexity of the second-order oracle of $f$ and $g$, i.e., $\nabla^2_{xx}f(\boldsymbol{x}, \boldsymbol{y}),$ $\nabla^2_{xy}f(\boldsymbol{x}, \boldsymbol{y}),$ $\nabla^2_{xx}g(\boldsymbol{x}, \boldsymbol{y}),$ $\nabla^2_{yy}g(\boldsymbol{x}, \boldsymbol{y}),$ $\nabla^2_{xy}g(\boldsymbol{x}, \boldsymbol{y}),$ as $dN$, where $d := \max\{d_x, d_y\}$ denotes the problem dimension.

The computational complexity of FSBA can be bounded by

$$
\begin{aligned}
&\text{Cost(FSBA)} \\
&= N \cdot \#\text{1st-order oracle} + Nd \cdot \#\text{2nd-order oracle} \quad (9) \\
&= \tilde{\mathcal{O}}\big(N(\kappa^{0.5} + d)\kappa^{2.5}\bar{\ell}^{0.5}\epsilon^{-1.5}\big).
\end{aligned}
$$

By Theorem F.5, IFSBA attains an $(\epsilon, \sqrt{\epsilon})$-SOSP with the following cost:

$$
\begin{aligned}
&\text{Cost(IFSBA)} \\
&= N \cdot \#\text{gradient oracle} + N \cdot \#\text{HVP oracle} \quad (10) \\
&= \tilde{\mathcal{O}}\big(N(\kappa^3\bar{\ell}^{0.5}\epsilon^{-1.5} + \kappa^{3.5}\bar{\ell}\epsilon^{-2})\big).
\end{aligned}
$$

### 4.1. The Lazy FSBA Method and its Convergence Analysis

In this section, our aim is to reduce the computational complexity of FSBA. At each iteration of FSBA, it takes

$\tilde{\mathcal{O}}(\kappa^{0.5}N)$ computational complexity to obtain $\boldsymbol{w}_t$ and $\boldsymbol{y}_t$ by AGD, and $\mathcal{O}(dN)$ computational complexity to update $\boldsymbol{x}_t$ by the inexact CRN step. When $d \gg \kappa^{0.5}$ such that the computational complexity of a second-order oracle is large, it is expensive to call the second-order oracle for every iteration. Motivated by the lazy Hessian mechanism (Doikov et al., 2023; Doikov & Grapiglia, 2023; Chen et al., 2024a; Liu et al., 2025; Chen et al., 2025a), we propose the lazy fully second-order bilevel approximation method (LFSBA), which computes the approximate Hessian only at the snapshot point and reuses it for the next $m$ iterations. We formally present LFSBA method in Algorithm 4.

---

**Algorithm 4** Lazy Fully Second-order Bilevel Approximation method (LFSBA)

1: **Input:** $\boldsymbol{x}_0 \in \mathbb{R}^{d_x}$, $\boldsymbol{y}_{-1} = \boldsymbol{0}, \boldsymbol{w}_{-1} = \boldsymbol{0}$, $\ell_1, \ell_2, \kappa_1, \kappa_2$, $m, \epsilon, M, T, \{K_t^1\}_{t=0}^T, \{K_t^2\}_{t=0}^T$.

2: **for** $t = 0, 1, \cdots, T-1$ **do**

3: $\quad \boldsymbol{w}_t = \text{AGD}\left(g\left(\boldsymbol{x}_t, \cdot\right), \boldsymbol{w}_{t-1}, K_t^1, \frac{1}{\ell_1}, \frac{\sqrt{\kappa_1}-1}{\sqrt{\kappa_1}+1}\right)$

4: $\quad \boldsymbol{y}_t = \text{AGD}\left(\mathcal{L}_\lambda\left(\boldsymbol{x}_t, \cdot\right), \boldsymbol{y}_{t-1}, K_t^2, \frac{1}{\ell_2}, \frac{\sqrt{\kappa_2}-1}{\sqrt{\kappa_2}+1}\right)$

5: $\quad$ Compute $\mathbf{g}_t = \mathbf{g}(\boldsymbol{x}_t; \boldsymbol{y}_t, \boldsymbol{w}_t)$ according to (5)

6: $\quad$ **if** $t\%m = 0$

7: $\quad\quad$ Set $\tilde{\boldsymbol{x}} = \boldsymbol{x}_t$

8: $\quad\quad$ Compute $\tilde{\mathbf{H}} = \mathbf{H}(\boldsymbol{x}_t; \boldsymbol{y}_t, \boldsymbol{w}_t)$ according to (6)

9: $\quad s_t^* = \arg\min_{\boldsymbol{s} \in \mathbb{R}^{d_x}} \left\{ \mathbf{g}_t^\top \boldsymbol{s} + \frac{1}{2}\boldsymbol{s}^\top \tilde{\mathbf{H}} \boldsymbol{s} + \frac{M}{6}\|\boldsymbol{s}\|^3 \right\}$

10: $\quad \boldsymbol{x}_{t+1} = \boldsymbol{x}_t + \boldsymbol{s}_t^*$

$\quad\quad$ **If** $\epsilon \geq \frac{1}{M}\left(\frac{288}{287}\right)^2\left(\frac{M+2\bar{\rho}}{\sqrt{2}}\|\boldsymbol{s}_t^*\| + \bar{\rho}\|\tilde{\boldsymbol{x}} - \boldsymbol{x}_t\|\right)^2$

11: $\quad$ **then break**

12: **end for**

13: **Ouput:** $\hat{\boldsymbol{x}} = \boldsymbol{x}_{t+1}$

---

Now, we study the convergence analysis of LFSBA, which updates according to the following direction

$$\boldsymbol{s}_t^* = \arg\min_{\boldsymbol{s} \in \mathbb{R}^{d_x}} \boldsymbol{s}^\top \mathbf{g}_t + \frac{1}{2}\boldsymbol{s}^\top \mathbf{H}_{\pi(t)} \boldsymbol{s} + \frac{M}{6}\|\boldsymbol{s}\|^3,$$

where we denote $\mathbf{g}_t := \mathbf{g}(\boldsymbol{x}_t; \boldsymbol{y}_t, \boldsymbol{w}_t)$, $\mathbf{H}_t := \mathbf{H}(\boldsymbol{x}_t; \boldsymbol{y}_t, \boldsymbol{w}_t)$, and $\pi(t) := t - t \mod m$. The following Lemma shows that once $\mathbf{g}_t$ and $\mathbf{H}_{\pi(t)}$ are good approximations of $\nabla \mathcal{L}_\lambda^*(\boldsymbol{x}_t)$ and $\nabla^2 \mathcal{L}_\lambda^*(\boldsymbol{x}_{\pi(t)})$, then LFSBA enjoys a similar descent property as the lazy cubic-regularized Newton method (Doikov et al., 2023).

**Lemma 4.2.** *Under Assumption 2.1, let $M \geq \bar{\rho}$ and suppose the following conditions*

$$\begin{aligned} \|\nabla \mathcal{L}_\lambda^*(\boldsymbol{x}_t) - \mathbf{g}_t\| &\leq \bar{C}_g \epsilon, \\ \left\|\nabla^2 \mathcal{L}_\lambda^*(\boldsymbol{x}_{\pi(t)}) - \mathbf{H}_{\pi(t)}\right\| &\leq \bar{C}_H \sqrt{M}\epsilon. \end{aligned} \tag{11}$$

*hold with $\bar{C}_g := 1/576, \bar{C}_H := 1/288$ in Algorithm 4, denoting $\gamma(\boldsymbol{x}) := \max\left\{\frac{1}{987M^2}\xi(\boldsymbol{x})^3, \frac{1}{120\sqrt{3M}}\|\nabla \mathcal{L}_\lambda^*(\boldsymbol{x})\|^{3/2}\right\}$,*

*then it holds that*

$$\begin{aligned} \mathcal{L}_\lambda^*(\boldsymbol{x}_t) &- \mathcal{L}_\lambda^*(\boldsymbol{x}_{t+1}) \geq \\ &\gamma(\boldsymbol{x}_{t+1}) + \frac{M}{72}\|\boldsymbol{x}_{t+1} - \boldsymbol{x}_t\|^3 - \frac{13\bar{\rho}^3}{M^2}\|\boldsymbol{x}_{\pi(t)} - \boldsymbol{x}_t\|^3. \end{aligned} \tag{12}$$

The following theorem indicates that by properly choosing the iteration steps in AGD subroutine and the regularization parameter, LSFBA converges at a similar convergence rate as the lazy cubic-regularized-Newton method.

**Theorem 4.3.** *Under Assumption 2.1, let $\Delta = \varphi(\boldsymbol{x}_0) - \varphi^*$, $\tilde{\epsilon} = \min\{\frac{\bar{C}_g \epsilon}{4\lambda \ell}, \frac{\bar{C}_H \sqrt{M\epsilon}}{2C_2}\}$, and $R = \max\{\|\boldsymbol{y}^*(\boldsymbol{x}_0)\|, \|\boldsymbol{y}_\lambda^*(\boldsymbol{x}_0)\|\}$, if we run Algorithm 4 with $M = \Omega(m\bar{\rho})$, $T = \Theta(\Delta\sqrt{M}\epsilon^{-3/2})$, $\lambda = \max\left\{\bar{\ell}\kappa^2/\Delta, \bar{\ell}\kappa^3/\epsilon, \bar{\ell}\kappa^5/\sqrt{M\epsilon}\right\}$, $\kappa_1 = \kappa, \ell_1 = \ell, \kappa_2 = 3\kappa, \ell_2 = (1+\lambda)\ell$, and*

$$K_t^1 = K_t^2 = \begin{cases} \left\lceil 2\sqrt{\kappa_2} \log\left(\frac{\sqrt{\kappa_2}+1}{\tilde{\epsilon}}R\right)\right\rceil & t = 0 \\ \left\lceil 2\sqrt{\kappa_2} \log\left(\frac{\sqrt{\kappa_2}+1}{\tilde{\epsilon}}\left(\tilde{\epsilon} + 4\kappa\|\boldsymbol{s}_{t-1}^*\|\right)\right)\right\rceil & t \geq 1 \end{cases},$$

*then the output $\hat{\boldsymbol{x}}$ is an $(\mathcal{O}(\epsilon), \mathcal{O}(\kappa^{2.5}\bar{\ell}^{0.5}m^{0.5}\epsilon^{0.5}))$-SOSP of $\varphi(\boldsymbol{x})$. Furthermore, the first-order and second-order oracle complexities of Algorithm 4 can be bounded by $\tilde{\mathcal{O}}(\kappa^3 \bar{\ell}^{0.5} m^{0.5} \epsilon^{-1.5})$ and $\mathcal{O}(1 + \kappa^{2.5}\bar{\ell}^{0.5}m^{-0.5}\epsilon^{-1.5})$, respectively.*

**Discussion on the computational complexity.** Theorem 4.3 indicates that the iteration complexity $\tilde{\mathcal{O}}(m^{0.5}\epsilon^{-1.5})$ of LFSBA is worse than $\tilde{\mathcal{O}}(\epsilon^{-1.5})$ of FSBA, which is due to the reuse of Hessian. However, considering the difference in computational complexity between first- and second-order oracles in Assumption 4.1, LFSBA achieves a better computational complexity by tuning $m$ for a trade-off of per-iteration computation cost and iteration complexity. We state the computational complexity of LFSBA as follows

$$\begin{aligned} &\text{Cost(LFSBA)} \\ &= N \cdot \#\text{1st-order oracle} + Nd \cdot \text{2nd-order oracle} \\ &= \tilde{\mathcal{O}}\left(N\kappa^3 m^{0.5}\bar{\ell}^{0.5}\epsilon^{-1.5} + Nd\kappa^{2.5}m^{-0.5}\bar{\ell}^{0.5}\epsilon^{-1.5}\right) \\ &= \tilde{\mathcal{O}}\left(N(\kappa^{0.5} + \kappa^{0.25}d^{0.5})\kappa^{2.5}\bar{\ell}^{0.5}\epsilon^{-1.5}\right), \end{aligned}$$

where the last inequality is by setting the frequency of update Hessian as $m = \Theta\left(1 + \frac{d}{\sqrt{\kappa}}\right)$.

*Remark* 4.4. The computational complexity of LFSBA improves FSBA (9) by a factor of $d^{0.5}/\kappa^{0.25}$, significantly reducing the computational cost when the dimension is large.

*Remark* 4.5. Once $\tilde{\mathbf{H}} = \mathbf{H}(\boldsymbol{x}_{\pi(t)}; \boldsymbol{y}_{\pi(t)}, \boldsymbol{w}_{\pi(t)})$ is computed, the cubic regularized-Newton update of line 9 in Algorithm 4 can be performed efficiently within $\tilde{\mathcal{O}}(d^2)$ by performing the eigenvalue decomposition on $\tilde{\mathbf{H}}$ (Doikov et al., 2023).

**Algorithm 5** Lazy Minimax Cubic Newton method (LMCN)

1: **Input:** $\boldsymbol{x}_0 \in \mathbb{R}^{d_x}, \boldsymbol{y}_{-1} = \mathbf{0}, T, \{K_t\}_{t=0}^T, \kappa_1, \ell_1, \epsilon, m, M$
2: **for** $t = 0, 1 \cdots, T-1$ **do**
3:     $\boldsymbol{y}_t = \text{AGD}\left(-f(\boldsymbol{x}_t, \cdot), \boldsymbol{y}_{t-1}, K_t, \frac{1}{\ell_1}, \frac{\sqrt{\kappa_1}-1}{\sqrt{\kappa_1}+1}\right)$
4:     Compute $\boldsymbol{g}_t = \boldsymbol{g}(\boldsymbol{x}_t; \boldsymbol{y}_t)$ according to (14).
5:     **if** $t\%m = 0$
6:        Set $\tilde{\boldsymbol{x}} = \boldsymbol{x}_t$
7:        Compute $\tilde{\mathbf{H}} = \mathbf{H}(\boldsymbol{x}_t; \boldsymbol{y}_t)$ according to (15)
8:     $\boldsymbol{s}_t^* = \arg\min_{\boldsymbol{s} \in \mathbb{R}^{d_x}} \left\{ \boldsymbol{g}_t^\top \boldsymbol{s} + \frac{1}{2}\boldsymbol{s}^\top \tilde{\mathbf{H}}\boldsymbol{s} + \frac{M}{6}\|\boldsymbol{s}\|^3 \right\}$
9:     $\boldsymbol{x}_{t+1} = \boldsymbol{x}_t + \boldsymbol{s}_t^*$
10:    **If** $\epsilon \geq \frac{1}{M}\left(\frac{288}{287}\right)^2\left(\frac{M+2\bar{\rho}}{\sqrt{2}}\|\boldsymbol{s}_t^*\| + \bar{\rho}\|\tilde{\boldsymbol{x}} - \boldsymbol{x}_t\|\right)^2$
11:    **then break**
12: **end for**
13: **Output:** $\hat{\boldsymbol{x}} = \boldsymbol{x}_{t+1}$

## 4.2. Improved Results for Nonconvex Strongly-Concave Minimax Problems

We adopts the idea of LFSBA to the solve following nonconvex strongly-concave minimax problem

$$\min_{\boldsymbol{x} \in \mathbb{R}^{d_x}} \max_{\boldsymbol{y} \in \mathbb{R}^{d_y}} f(\boldsymbol{x}, \boldsymbol{y}). \tag{13}$$

Let $\varphi(\cdot) := \arg\max_{\boldsymbol{y} \in \mathbb{R}^{d_x}} f(\cdot, \boldsymbol{y})$, the above minimax problem can be regarded as a special bilevel optimization problem (1) with the lower function $g(\boldsymbol{x}, \boldsymbol{y}) = -f(\boldsymbol{x}, \boldsymbol{y})$. We suppose $f(\cdot, \cdot)$ and $\varphi(\cdot)$ satisfy the following assumption.

**Assumption 4.6.** $f(\cdot, \cdot)$ and $\varphi(\cdot)$ satisfy the following conditions: (a) $f(\boldsymbol{x}, \boldsymbol{y})$ is twice differentiable and $\mu$-strongly concave with respect to $\boldsymbol{y}$ for any fixed $\boldsymbol{x}$; (b) $\nabla f(\cdot, \cdot)$ is $\ell$-Lipschitz continuous and and $\nabla^2 f(\cdot, \cdot)$ is $\rho$-Lipschitz continuous; (c) $\varphi^* := \min_{\boldsymbol{x} \in \mathbb{R}^{d_x}} \varphi(\boldsymbol{x}) > -\infty$.

Since $\nabla\varphi(\cdot)$ and $\nabla^2\varphi(\cdot)$ are Lipschitz continuous (Luo et al., 2022), they can be well approximated by introducing an additional variable $\boldsymbol{y} \approx \boldsymbol{y}^*(\boldsymbol{x})$ such that

$$\boldsymbol{g}(\boldsymbol{x}; \boldsymbol{y}) := \nabla_x f(\boldsymbol{x}, \boldsymbol{y}) \tag{14}$$

$$\mathbf{H}(\boldsymbol{x}; \boldsymbol{y}) := \\ \nabla_{xx}^2 f(\boldsymbol{x}, \boldsymbol{y}) - \nabla_{xy}^2 f(\boldsymbol{x}, \boldsymbol{y})[\nabla_{yy}^2 f(\boldsymbol{x}, \boldsymbol{y})]^{-1}\nabla_{yx}^2 f(\boldsymbol{x}, \boldsymbol{y}). \tag{15}$$

Then, applying a similar "lazy" strategy as introduced in LFSBA based on $\boldsymbol{g}(\boldsymbol{x}; \boldsymbol{y})$ and $\mathbf{H}(\boldsymbol{x}; \boldsymbol{y})$ leads to our lazy minimax cubic-regularized Newton method (LMCN), presented in Algorithm 5. The LMCN generalizes the MCN method (Luo et al., 2022) and has better computational complexity than the MCN method in finding the SOSP of $\varphi(\boldsymbol{x})$.

**Theorem 4.7.** *Under Assumptions 4.1 and 4.6, LMCN (Algorithm 5) can find an $(\epsilon, \kappa^{1.25}\sqrt{d\rho\epsilon})$-SOSP of $\varphi(\cdot)$, where $\kappa = \ell/\mu$ within computational complexity*

$$\text{Cost(LMCN)} = \tilde{\mathcal{O}}(N(\kappa^{0.5} + \kappa^{0.25}d^{0.5})\kappa^{1.5}\rho^{0.5}\epsilon^{-1.5}). \tag{16}$$

*We let $\tilde{\epsilon} = \min\{\epsilon/(576\ell), \sqrt{M}\epsilon/(288\rho)\}$, $\bar{\rho} = 4\sqrt{2}\kappa^3\rho$, and set $\kappa_1 = \kappa$, $\ell_1 = \ell$, $m = d/\sqrt{\kappa}+1$, $M = \Theta(m\bar{\rho})$,*

$$K_t = \begin{cases} \left\lceil 2\sqrt{\kappa}\log\left(\frac{\sqrt{\kappa+1}}{\tilde{\epsilon}}\|\boldsymbol{y}^*(\boldsymbol{x}_0)\|\right)\right\rceil & t = 0 \\ \left\lceil 2\sqrt{\kappa}\log\left(\frac{\sqrt{\kappa+1}}{\tilde{\epsilon}}(\tilde{\epsilon} + \kappa\|\boldsymbol{x}_t - \boldsymbol{x}_{t-1}\|)\right)\right\rceil & t \geq 1 \end{cases},$$

*and $T = \Theta\left((\varphi(\boldsymbol{x}_0) - \varphi^*)\sqrt{M}\epsilon^{-3/2}\right)$ in Algorithm 5 to achieve (16).*

**Remark 4.8.** The computational complexity of the LMCN is better than $\tilde{\mathcal{O}}(N(\kappa^{0.5}+d)\kappa^{1.5}\rho^{0.5}\epsilon^{-1.5})$ of the MCN (Luo et al., 2022) due to the AM-GM inequality.

## 5. Numerical Experiments

### 5.1. Data Hypercleaning

We conduct experiments to validate the efficiency of the proposed methods on the *data hyper-cleaning* task (Franceschi et al., 2018; Shaban et al., 2019; Zhou et al., 2022), which can be formulated as a bilevel optimization problem (1) with the following upper and lower-level objectives:

$$f(\boldsymbol{x}, \boldsymbol{y}) := \frac{1}{|\mathcal{D}^{\text{val}}|}\sum_{(\mathbf{a}_i, b_i) \in \mathcal{D}^{\text{val}}} \ell(\langle\mathbf{a}_i, \boldsymbol{y}\rangle, b_i),$$

$$g(\boldsymbol{x}, \boldsymbol{y}) := \frac{1}{|\mathcal{D}^{\text{tr}}|}\sum_{(\mathbf{a}_i, b_i) \in \mathcal{D}^{\text{tr}}} \sigma(x_i)\ell(\langle\mathbf{a}_i, \boldsymbol{y}\rangle, b_i) + c\|\boldsymbol{y}\|^2.$$

In the above, $\mathcal{D}^{\text{tr}}$ denotes the noisy training set and $\mathcal{D}^{\text{val}}$ denotes the validation set. $(\mathbf{a}_i, b_i)$ denotes the $i$-th sample in the dataset, where $\mathbf{a}_i$ represents the feature and $b_i$ represents its corresponding label. We denote $\sigma(\cdot)$ as a clipping function that maps a scalar to the interval $[0, 1]$ and $\ell(\cdot, \cdot)$ is the loss of cross entropy. We set $c = 10^{-3}$.

We compare the FSBA method (Algorithm 2) and its lazy variant (Algorithm 4) with baseline methods, including ITD (Ji et al., 2021), AID with conjugate gradient (Maclaurin et al., 2015), and near optimal fully first-order methods F²BA (Chen et al., 2025b) on "breast-cancer" and "australian" datasets (Chang & Lin, 2011). We report the results on $\mathcal{D}_{tr}$ with different noise rates $p = 25\%$ and $p = 50\%$ (the ratio of training samples with disrupted labels) in Figure 1, which demonstrates that our LFSBA and FSBA methods converge faster than the baselines. We defer the hyperparameter tuning details for this experiment to the appendix G.3.

### 5.2. Hyperparameter Tuning

We validate the proposed methods on *hyperparameter tuning* task, which aims to find the optimal hyperparameter

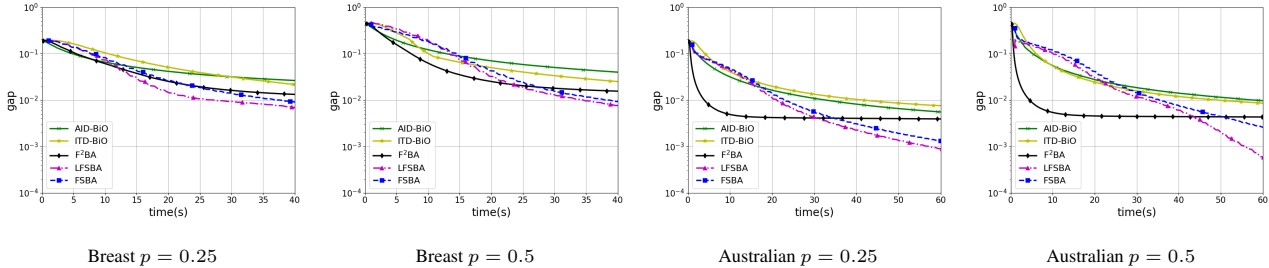

Figure 1. Comparison of various bilevel algorithms for data hyperclearning on "breast-cancer" and "australian" datasets.

that minimizes the loss on the validation dataset. The *hyperparameter tuning* task can be reformulated as a bilevel optimization problem with the upper and lower-level objectives:

$$f(\boldsymbol{x}, \boldsymbol{y}) := \frac{1}{|\mathcal{D}_{\text{val}}|} \sum_{(\boldsymbol{a}_i, \boldsymbol{b}_i) \in \mathcal{D}_{\text{val}}} L(\boldsymbol{y}^*(\boldsymbol{x}); \boldsymbol{a}_i, \boldsymbol{b}_i),$$

$$g(\boldsymbol{x}, \boldsymbol{y}) := \frac{1}{|\mathcal{D}_{\text{tr}}|} \sum_{(\boldsymbol{a}_i, \boldsymbol{b}_i) \in \mathcal{D}_{\text{tr}}} L(\boldsymbol{w}; \boldsymbol{a}_i, \boldsymbol{b}_i) + \frac{1}{2cp} \sum_{j=1}^{c} \sum_{k=1}^{p} \exp(x_k) y_{jk}^2,$$

where $\boldsymbol{x} = [x_1, \cdots, x_k, \cdots, x_p]^\top \in \mathbb{R}^p$, $\boldsymbol{y} \in \mathbb{R}^{c \times p}$, $\mathcal{D}_{\text{tr}} = \{(\boldsymbol{a}_i, \boldsymbol{b}_i)\}$ is the training dataset, $\mathcal{D}_{\text{val}} = \{(\boldsymbol{x}_i, \boldsymbol{y}_i)\}$ is the validation dataset, $L(\cdot; \cdot, \cdot)$ is the cross-entropy loss.

We compare the performance of the inexact variant of FSBA (IFSBA, Algorithm 3) with the baseline algorithms over a logistic regression problem on 20 News group dataset (Grazzi et al., 2020) ($c = 20, p = 130170$). We divide the datasets into three parts: 5657 for training, 5657 for validation, and 7532 for testing.

We use the same hyperparameter tuning protocol as in the data cleaning experiments. The results are presented in Figure 2 and we observe that IFSBA converges faster than the baselines.

### 5.3. Few-Shot Meta-Learning

We then conduct experiments to validate the efficiency of the proposed methods on *few-shot meta-learning* task (Finn et al., 2017; Raghu et al., 2019; Ji et al., 2021; Fang et al., 2025). We consider $m$ few-shot tasks $\{\mathcal{T}_i\}_{i=1}^m$ sampled from a task distribution $\mathcal{P}_{\mathcal{T}}$, where each task has a support set $\mathcal{S}_i$ and a query set $\mathcal{D}_i$. We use a four-layer CNN with shared parameters $\boldsymbol{x}$ as the feature extractor, and use $\boldsymbol{y}_i$ as the last-layer linear classifier for task $\mathcal{T}_i$. The meta-learning objective can be written as

$$\min_{\boldsymbol{x}} \quad \frac{1}{m} \sum_{i=1}^m \mathcal{L}_{\mathcal{D}_i}(\boldsymbol{x}, \boldsymbol{y}_i^*(\boldsymbol{x}))$$

$$\text{s.t.} \quad \boldsymbol{y}_i^*(\boldsymbol{x}) \in \arg\min_{\boldsymbol{y}_i} \mathcal{L}_{\mathcal{S}_i}(\boldsymbol{x}, \boldsymbol{y}_i).$$

Here, $\mathcal{L}_{\mathcal{D}_i}(\boldsymbol{x}, \boldsymbol{y}_i) = \frac{1}{|\mathcal{D}_i|} \sum_{\xi \in \mathcal{D}_i} \ell(\boldsymbol{x}, \boldsymbol{y}_i; \xi)$ is the query loss, and $\mathcal{L}_{\mathcal{S}_i}(\boldsymbol{x}, \boldsymbol{y}_i) = \frac{1}{|\mathcal{S}_i|} \sum_{\xi \in \mathcal{S}_i} (\ell(\boldsymbol{x}, \boldsymbol{y}_i; \xi) + \mathcal{R}(\boldsymbol{y}_i))$ is the support loss. In our experiments, $\ell$ is the cross-entropy loss and $\mathcal{R}$ is an $\ell_2$ regularizer.

Since prior work has shown that PZOBO (Sow et al., 2022b) outperforms standard baselines such as MAML (Finn et al., 2017) and ANIL (Raghu et al., 2019), we follow the same evaluation protocol and compare only against the stronger baselines PZOBO and qNBO (Fang et al., 2025). Under this setting, we evaluate F²BA and IFSBA (Algorithm 3), in 5-way 5-shot experiments on miniImageNet (Vinyals et al., 2016) and FC100 (Oreshkin et al., 2018). Results are averaged over five runs. All algorithms start from the same initialization with 20% test accuracy, and the first data point is omitted for clarity.

For PZOBO and qNBO, we follow the hyperparameter settings used in their respective original implementations (Sow et al., 2022b; Fang et al., 2025). For F²BA and IFSBA, we tune the inner- and outer-loop learning rates over $\{10^{-3}, 10^{-2}, 10^{-1}, 1, 10, 10^2, 10^3\}$, the number of GD or AGD iterations over $\{5, 10, 30, 50\}$, and the penalty multiplier $\lambda$ over $\{1, 10, 10^2, 10^3\}$. For IFSBA, we tune $M$ from $\{1, 10^1, 10^2, 10^3\}$, the number of Cubic-Solver iterations and the order of Matrix Chebyshev Polynomials from $\{1, 5, 10, 100\}$. The results are presented in Figure 3, where IFSBA achieves higher test accuracy than the baselines within the same running time.

## 6. Conclusion

In this paper, we have proposed several fully second-order methods for nonconvex strongly-convex bilevel optimization. The FSBA method takes $\tilde{\mathcal{O}}(\epsilon^{-1.5})$ second-order oracle complexity to find the $(\epsilon, \mathcal{O}(\sqrt{\epsilon}))$ SOSP of the hyper objective $\varphi(\cdot)$, and it is faster than the existing first- and second-order methods, showing the advantage of using second-order oracles in bilevel optimization. The LFSBA method applies the lazy Hessian strategy and reduces the computational complexity of FSBA.

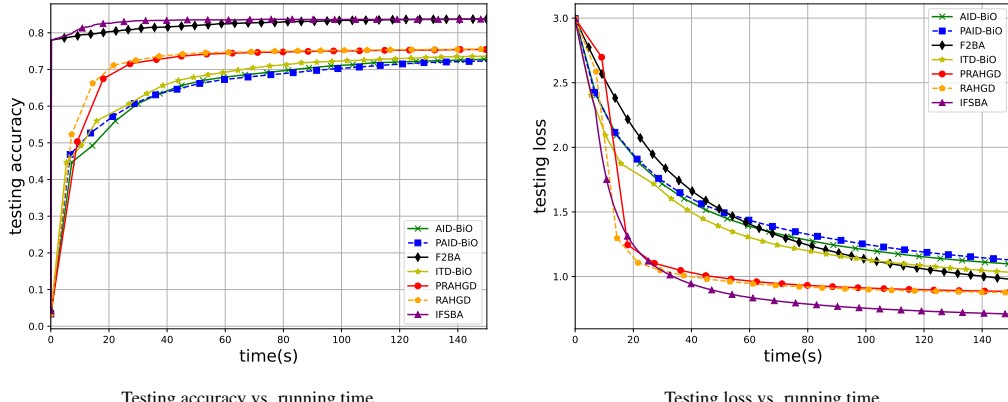

Figure 2. Comparison of different bilevel algorithms for hyperparameter tuning on the 20 Newsgroups dataset.

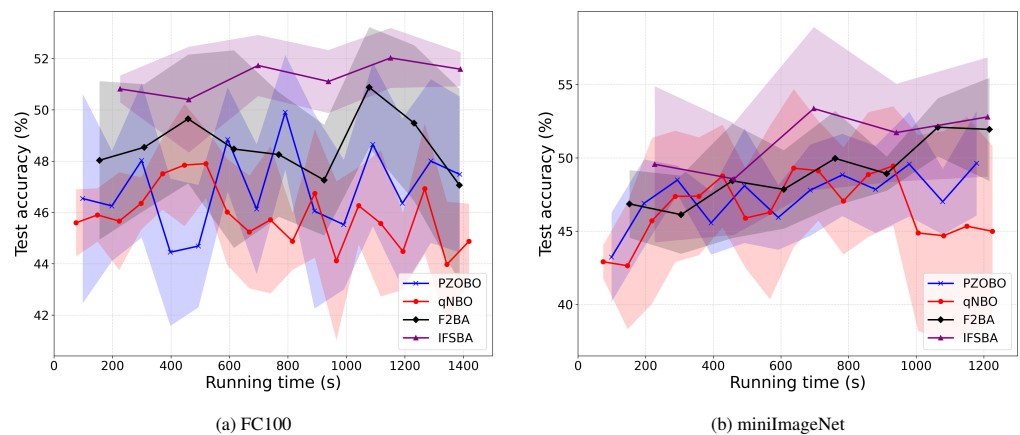

Figure 3. Comparison of different bilevel algorithms for few-shot meta-learning on FC100 and miniImageNet.

## Acknowledgment

The authors thank Luo Luo for valuable discussion and anonymous reviewers for their helpful suggestions. Chengchang Liu is supported by the National Natural Science Foundation of China (624B2125). John C.S. Lui is supported in part by the GRF-14207721 and SRFS2122-4S02.

## Impact Statement

This paper presents work whose goal is to advance the field of machine learning. There are many potential societal consequences of our work, none of which we feel must be specifically highlighted here.

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

# A. Future Directions

We list some future directions of this paper in this section.

- We use AGD to solve the lower-level problems to obtain $\boldsymbol{y}_t \approx \boldsymbol{y}_\lambda^*(\boldsymbol{x}_t)$ and $\boldsymbol{w}_t \approx \boldsymbol{y}(\boldsymbol{x}_t)$ within $\tilde{\mathcal{O}}(\sqrt{\kappa})$ iterations. It is possible to accelerate the lower solvers by using second-order methods (Carmon et al., 2022; Kovalev & Gasnikov, 2022; Kornowski & Shamir, 2020) to improve the complexity dependency on $\kappa$.

- We consider the fully second-order methods for nonconvex strongly-convex bilevel optimization. It will be interesting to develop second-order methods for bilevel optimization without lower strong convexity (Chen et al., 2024b; Liu et al., 2020; 2021a;b; Shen & Chen, 2023; Sow et al., 2022a; Xiao et al., 2023a;b; Kwon et al., 2023a; Yao et al., 2024; Lu, 2023; Arbel & Mairal, 2022b) and demonstrate the superiority of second-order methods over the first-order methods under this setting.

- We consider the deterministic setting such that one can access the exact gradient and Hessian oracle of $f$ and $g$. It is also important to design stochastic (Kwon et al., 2023b; Huo et al., 2023; Khanduri et al., 2021; Chu et al., 2025; Wang et al., 2024b; Yang et al., 2021; Dong et al., 2025) and distributed (Wang et al., 2024b; Lian et al., 2017; Scaman et al., 2017; Mishchenko et al., 2022; Wang et al., 2022; Ye & Abbe, 2018; Yuan et al., 2022) variants of FSBA and LFSBA to further improve the practical performance.

# B. Discussion on the Related Works

Luo et al. (2022) proposed a novel second-order method to solve NC-SC minimax optimization by applying inexact CRN on $\varphi(\cdot)$ and AGD to find $\mathbf{y}_t \approx \mathbf{y}^*(\mathbf{x}_t) = \arg\max_{\mathbf{y}} f(\mathbf{x}_t, \mathbf{y})$, which finds the $(\epsilon, \mathcal{O}(\epsilon^{0.5}))$-SOSP of the $\varphi(\mathbf{x}) = \max_{\mathbf{y}} f(\mathbf{x}, \mathbf{y})$ within $\epsilon^{-1.5}$ iterations. If we directly apply their methods to the hyperfunction $\varphi(\mathbf{x})$ for the NC-SC bilevel optimization, the inexact CRN method will require third-order derivatives of the lower-level function $g(\mathbf{x}, \mathbf{y})$. In addition, their methods always query the second-order oracle at every outer level iterations, while we provide a computational efficient variant that only computes the second-order oracle at the snapshot points. We show that even for the NS-SC minimax problem, our method demonstrates an improved computational complexity compared to theirs.

Qiu et al. (2026) proposed a second-order lazy bilevel method for stochastic NC-SC bilevel optimization. However, their method only explores the second-order oracle at the lower-level function $g$, thus cannot demonstrate the advantage of second-order method compared to the first-order methods in the deterministic setting.

# C. Useful Lemmas

**Lemma C.1** (Lemma 2, Wang & Li (2020)). *Running Algorithm 1 on $\ell_h$-smooth and $\mu_h$-strongly-convex objective function $h(\cdot)$ with parameters $\eta = 1/\ell_h$ and $\theta = \frac{\sqrt{\kappa_h}-1}{\sqrt{\kappa_h}+1}$ produces the output $\boldsymbol{y}_K$ satisfying $\|\boldsymbol{y}_K - \boldsymbol{y}^*\|^2 \leq (\kappa_h + 1)\left(1 - \frac{1}{\sqrt{\kappa_h}}\right)^K \|\boldsymbol{y}_0 - \boldsymbol{y}^*\|^2$, where $\boldsymbol{y}^* = \arg\min_y h(\boldsymbol{y})$ and $\kappa_h = \ell_h/\mu_h$.*

**Lemma C.2** (Lemma 3.2, Kwon et al. (2023b)). *Under Assumption 2.1, for $\lambda \geq 2\ell/\mu$, $\mathcal{L}_\lambda(\boldsymbol{x}, \cdot)$ is $(\lambda\mu/2)$-strongly convex.*

It is clear that $\boldsymbol{y}^*(\boldsymbol{x})$ is $\ell/\mu$-Lipschitz. And we can also show a similar result for $\boldsymbol{y}_\lambda^*(\boldsymbol{x})$.

**Lemma C.3** (Lemma B.6, Chen et al. (2025b)). *Under Assumption 2.1, for $\lambda \geq 2\ell/\mu$, it holds that $\boldsymbol{y}_\lambda^*(\boldsymbol{x})$ is $(4\ell/\mu)$-Lipschitz.*

**Lemma C.4** (Nesterov & Polyak (2006)). *Suppose Assumption 2.1 holds, according to Proposition 2.5, we have the following inequalities for the Hessian Lipschitz continuity:*

$$\left\|\nabla \mathcal{L}_\lambda^*(\boldsymbol{x}') - \nabla \mathcal{L}_\lambda^*(\boldsymbol{x}) - \nabla^2 \mathcal{L}_\lambda^*(\boldsymbol{x})(\boldsymbol{x}' - \boldsymbol{x})\right\| \leq \frac{\bar{\rho}}{2}\|\boldsymbol{x}' - \boldsymbol{x}\|^2, \tag{17}$$

$$\left|\mathcal{L}_\lambda^*(\boldsymbol{x}') - \mathcal{L}_\lambda^*(\boldsymbol{x}) - \langle\nabla \mathcal{L}_\lambda^*(\boldsymbol{x}), \boldsymbol{x}' - \boldsymbol{x}\rangle - \frac{1}{2}\left\langle\nabla^2 \mathcal{L}_\lambda^*(\boldsymbol{x})(\boldsymbol{x}' - \boldsymbol{x}), \boldsymbol{x}' - \boldsymbol{x}\right\rangle\right| \leq \frac{\bar{\rho}}{6}\|\boldsymbol{x}' - \boldsymbol{x}\|^3. \tag{18}$$

**Lemma C.5** (Nesterov & Polyak (2006)). *For any $M' \geq 0$, we denote $\boldsymbol{g}$ is the gradient of the objective function and $\boldsymbol{H}$ is the Hessian of the objective function, then the solution $\boldsymbol{s}^*$ of the following cubic regularized quadratic problem*

$$\boldsymbol{s}^* = \underset{\boldsymbol{x} \in \mathbb{R}^{d_x}}{\arg\min}\left(\boldsymbol{g}^\top \boldsymbol{s} + \frac{1}{2}\boldsymbol{s}^\top \boldsymbol{H}\boldsymbol{s} + \frac{M'}{6}\|\boldsymbol{s}\|^3\right)$$

*satisfies*

$$\mathbf{g} + \mathbf{H}s^* + \frac{M'}{2}\|s^*\|\,s^* = \mathbf{0}, \tag{19}$$

$$\mathbf{H} + \frac{M'}{2}\|s^*\|\,\mathbf{I} \succeq \mathbf{0}, \tag{20}$$

$$\mathbf{g}^\top s^* + \frac{1}{2}\left(s^*\right)^\top \mathbf{H}s^* + \frac{M'}{6}\|s^*\|^3 \leq -\frac{M'}{12}\|s^*\|^3. \tag{21}$$

**Lemma C.6** ((Doikov et al., 2023), Lemma B.1). *For any sequence of positive numbers $\{r_t\}_{t\geq 1}$, it holds for any $m \geq 1$:*

$$\sum_{t=1}^{m-1}\left(\sum_{i=1}^{t} r_i\right)^3 \leq \frac{m^3}{3}\sum_{t=1}^{m-1} r_t^3. \tag{22}$$

## D. The Proof of Section 3

### D.1. The Proof of Lemma 3.1

*Proof.* We first need to derive an upper bound for the following equations:

$$\left\|\nabla_{xy}^2 g\left(\boldsymbol{x},\boldsymbol{y}^*(\boldsymbol{x})\right)\left[\nabla_{yy}^2 g\left(\boldsymbol{x},\boldsymbol{y}^*(\boldsymbol{x})\right)\right]^{-1} - \nabla_{xy}^2 g\left(\boldsymbol{x},\boldsymbol{w}\right)\left[\nabla_{yy}^2 g\left(\boldsymbol{x},\boldsymbol{w}\right)\right]^{-1}\right\|$$

$$\text{and} \quad \left\|\nabla_{xy}^2 \mathcal{L}_\lambda\left(\boldsymbol{x},\boldsymbol{y}_\lambda^*(\boldsymbol{x})\right)\left[\nabla_{yy}^2 \mathcal{L}_\lambda\left(\boldsymbol{x},\boldsymbol{y}_\lambda^*(\boldsymbol{x})\right)\right]^{-1} - \nabla_{xy}^2 \mathcal{L}_\lambda\left(\boldsymbol{x},\boldsymbol{y}\right)\left[\nabla_{yy}^2 \mathcal{L}_\lambda\left(\boldsymbol{x},\boldsymbol{y}\right)\right]^{-1}\right\|.$$

Using the matrix identity $A^{-1} - B^{-1} = A^{-1}(B - A)B^{-1}$, we have

$$\left\|\left[\nabla_{yy}^2 g\left(\boldsymbol{x},\boldsymbol{y}^*(\boldsymbol{x})\right)\right]^{-1} - \left[\nabla_{yy}^2 g\left(\boldsymbol{x},\boldsymbol{w}\right)\right]^{-1}\right\|$$

$$\leq \left\|\left[\nabla_{yy}^2 g\left(\boldsymbol{x},\boldsymbol{y}^*(\boldsymbol{x})\right)\right]^{-1}\right\|\left\|\nabla_{yy}^2 g\left(\boldsymbol{x},\boldsymbol{w}\right) - \nabla_{yy}^2 g\left(\boldsymbol{x},\boldsymbol{y}^*(\boldsymbol{x})\right)\right\|\left\|\left[\nabla_{yy}^2 g\left(\boldsymbol{x},\boldsymbol{w}\right)\right]^{-1}\right\|$$

$$\leq \frac{\rho}{\mu^2}\|\boldsymbol{w} - \boldsymbol{y}^*(\boldsymbol{x})\|,$$

and we further have

$$\left\|\nabla_{xy}^2 g\left(\boldsymbol{x},\boldsymbol{y}^*(\boldsymbol{x})\right)\left[\nabla_{yy}^2 g\left(\boldsymbol{x},\boldsymbol{y}^*(\boldsymbol{x})\right)\right]^{-1} - \nabla_{xy}^2 g\left(\boldsymbol{x},\boldsymbol{w}\right)\left[\nabla_{yy}^2 g\left(\boldsymbol{x},\boldsymbol{w}\right)\right]^{-1}\right\|$$

$$\leq \left\|\nabla_{xy}^2 g\left(\boldsymbol{x},\boldsymbol{y}^*(\boldsymbol{x})\right) - \nabla_{xy}^2 g\left(\boldsymbol{x},\boldsymbol{w}\right)\right\|\left\|\left[\nabla_{yy}^2 g\left(\boldsymbol{x},\boldsymbol{y}^*(\boldsymbol{x})\right)\right]^{-1}\right\|$$

$$\quad + \left\|\nabla_{xy}^2 g\left(\boldsymbol{x},\boldsymbol{w}\right)\right\|\left\|\left[\nabla_{yy}^2 g\left(\boldsymbol{x},\boldsymbol{y}^*(\boldsymbol{x})\right)\right]^{-1} - \left[\nabla_{yy}^2 g\left(\boldsymbol{x},\boldsymbol{w}\right)\right]^{-1}\right\|$$

$$\leq \left(\frac{\rho}{\mu} + \frac{\ell\rho}{\mu^2}\right)\|\boldsymbol{w} - \boldsymbol{y}^*(\boldsymbol{x})\|.$$

Similarly, using the matrix identity $A^{-1} - B^{-1} = A^{-1}(B - A)B^{-1}$, we have

$$\left\|\left[\nabla_{yy}^2 \mathcal{L}_\lambda\left(\boldsymbol{x},\boldsymbol{y}_\lambda^*(\boldsymbol{x})\right)\right]^{-1} - \left[\nabla_{yy}^2 \mathcal{L}_\lambda\left(\boldsymbol{x},\boldsymbol{y}\right)\right]^{-1}\right\|$$

$$\leq \left\|\left[\nabla_{yy}^2 \mathcal{L}_\lambda\left(\boldsymbol{x},\boldsymbol{y}_\lambda^*(\boldsymbol{x})\right)\right]^{-1}\right\|\left\|\nabla_{yy}^2 \mathcal{L}_\lambda\left(\boldsymbol{x},\boldsymbol{y}\right) - \nabla_{yy}^2 \mathcal{L}_\lambda\left(\boldsymbol{x},\boldsymbol{y}_\lambda^*(\boldsymbol{x})\right)\right\|\left\|\left[\nabla_{yy}^2 \mathcal{L}_\lambda\left(\boldsymbol{x},\boldsymbol{y}\right)\right]^{-1}\right\|$$

$$\leq \frac{4\left(\rho + \lambda\rho\right)}{\lambda^2\mu^2}\|\boldsymbol{y} - \boldsymbol{y}_\lambda^*(\boldsymbol{x})\|,$$

and we further have

$$\left\| \nabla_{xy}^2 \mathcal{L}_\lambda \left(\boldsymbol{x}, \boldsymbol{y}_\lambda^*(\boldsymbol{x})\right) \left[\nabla_{yy}^2 \mathcal{L}_\lambda \left(\boldsymbol{x}, \boldsymbol{y}_\lambda^*(\boldsymbol{x})\right)\right]^{-1} - \nabla_{xy}^2 \mathcal{L}_\lambda \left(\boldsymbol{x}, \boldsymbol{y}\right) \left[\nabla_{yy}^2 \mathcal{L}_\lambda \left(\boldsymbol{x}, \boldsymbol{y}\right)\right]^{-1} \right\|$$

$$\leq \left\| \nabla_{xy}^2 \mathcal{L}_\lambda \left(\boldsymbol{x}, \boldsymbol{y}_\lambda^*(\boldsymbol{x})\right) - \nabla_{xy}^2 \mathcal{L}_\lambda \left(\boldsymbol{x}, \boldsymbol{y}\right) \right\| \left\| \left[\nabla_{yy}^2 \mathcal{L}_\lambda \left(\boldsymbol{x}, \boldsymbol{y}_\lambda^*(\boldsymbol{x})\right)\right]^{-1} \right\|$$

$$+ \left\| \nabla_{xy}^2 \mathcal{L}_\lambda \left(\boldsymbol{x}, \boldsymbol{y}\right) \right\| \left\| \left[\nabla_{yy}^2 \mathcal{L}_\lambda \left(\boldsymbol{x}, \boldsymbol{y}_\lambda^*(\boldsymbol{x})\right)\right]^{-1} - \left[\nabla_{yy}^2 \mathcal{L}_\lambda \left(\boldsymbol{x}, \boldsymbol{y}\right)\right]^{-1} \right\|$$

$$\leq \left( \frac{2\left(\rho + \lambda\rho\right)}{\lambda\mu} + \frac{4\left(\rho + \lambda\rho\right)\left(\ell + \lambda\ell\right)}{\lambda^2\mu^2} \right) \left\| \boldsymbol{y} - \boldsymbol{y}_\lambda^*(\boldsymbol{x}) \right\|.$$

According to

$$\nabla \mathcal{L}_\lambda^*(\boldsymbol{x}) = \nabla_x f\left(\boldsymbol{x}, \boldsymbol{y}_\lambda^*(\boldsymbol{x})\right) + \lambda\left(\nabla_x g\left(\boldsymbol{x}, \boldsymbol{y}_\lambda^*(\boldsymbol{x})\right) - \nabla_x g\left(\boldsymbol{x}, \boldsymbol{y}^*(\boldsymbol{x})\right)\right),$$

and

$$\mathbf{g}(\boldsymbol{x}; \boldsymbol{y}, \boldsymbol{w}) = \nabla_x f\left(\boldsymbol{x}, \boldsymbol{y}\right) + \lambda\left(\nabla_x g\left(\boldsymbol{x}, \boldsymbol{y}\right) - \nabla_x g\left(\boldsymbol{x}, \boldsymbol{w}\right)\right).$$

Then we have

$$\left\| \nabla \mathcal{L}_\lambda^*(\boldsymbol{x}) - \mathbf{g}(\boldsymbol{x}; \boldsymbol{y}, \boldsymbol{w}) \right\|$$

$$\leq \left\| \nabla_x f\left(\boldsymbol{x}, \boldsymbol{y}\right) - \nabla_x f\left(\boldsymbol{x}, \boldsymbol{y}_\lambda^*(\boldsymbol{x})\right) \right\| + \lambda \left\| \nabla_x g\left(\boldsymbol{x}, \boldsymbol{y}\right) - \nabla_x g\left(\boldsymbol{x}, \boldsymbol{y}_\lambda^*(\boldsymbol{x})\right) \right\|$$

$$+ \lambda \left\| \nabla_x g\left(\boldsymbol{x}, \boldsymbol{w}\right) - \nabla_x g\left(\boldsymbol{x}, \boldsymbol{y}^*(\boldsymbol{x})\right) \right\|$$

$$\leq 2\lambda\ell \left\| \boldsymbol{y} - \boldsymbol{y}_\lambda^*(\boldsymbol{x}) \right\| + \lambda\ell \left\| \boldsymbol{w} - \boldsymbol{y}^*(\boldsymbol{x}) \right\|.$$

Note that

$$\nabla^2 \mathcal{L}_\lambda^*(\boldsymbol{x}) = \nabla_{xx}^2 f\left(\boldsymbol{x}, \boldsymbol{y}_\lambda^*(\boldsymbol{x})\right) - \nabla_{xy}^2 \mathcal{L}_\lambda \left(\boldsymbol{x}, \boldsymbol{y}_\lambda^*(\boldsymbol{x})\right) \left[\nabla_{yy}^2 \mathcal{L}_\lambda \left(\boldsymbol{x}, \boldsymbol{y}_\lambda^*(\boldsymbol{x})\right)\right]^{-1} \nabla_{yx}^2 \mathcal{L}_\lambda \left(\boldsymbol{x}, \boldsymbol{y}_\lambda^*(\boldsymbol{x})\right)$$

$$+ \lambda \left( \nabla_{xx}^2 g\left(\boldsymbol{x}, \boldsymbol{y}_\lambda^*(\boldsymbol{x})\right) - \nabla_{xx}^2 g\left(\boldsymbol{x}, \boldsymbol{y}^*(\boldsymbol{x})\right) + \nabla_{xy}^2 g\left(\boldsymbol{x}, \boldsymbol{y}^*(\boldsymbol{x})\right) \left[\nabla_{yy}^2 g\left(\boldsymbol{x}, \boldsymbol{y}^*(\boldsymbol{x})\right)\right]^{-1} \nabla_{yx}^2 g\left(\boldsymbol{x}, \boldsymbol{y}^*(\boldsymbol{x})\right) \right),$$

and

$$\mathbf{H}(\boldsymbol{x}; \boldsymbol{y}, \boldsymbol{w}) := \nabla_{xx}^2 f\left(\boldsymbol{x}, \boldsymbol{y}\right) - \nabla_{xy}^2 \mathcal{L}_\lambda \left(\boldsymbol{x}, \boldsymbol{y}\right) \left[\nabla_{yy}^2 \mathcal{L}_\lambda \left(\boldsymbol{x}, \boldsymbol{y}\right)\right]^{-1} \nabla_{yx}^2 \mathcal{L}_\lambda \left(\boldsymbol{x}, \boldsymbol{y}\right)$$

$$+ \lambda \left( \nabla_{xx}^2 g\left(\boldsymbol{x}, \boldsymbol{y}\right) - \nabla_{xx}^2 g\left(\boldsymbol{x}, \boldsymbol{w}\right) + \nabla_{xy}^2 g\left(\boldsymbol{x}, \boldsymbol{w}\right) \left[\nabla_{yy}^2 g\left(\boldsymbol{x}, \boldsymbol{w}\right)\right]^{-1} \nabla_{yx}^2 g\left(\boldsymbol{x}, \boldsymbol{w}\right) \right),$$

We can obtain the following inequalities:

$$\left\| \nabla_{xx}^2 f\left(\boldsymbol{x}, \boldsymbol{y}_\lambda^*(\boldsymbol{x})\right) - \nabla_{xx}^2 f\left(\boldsymbol{x}, \boldsymbol{y}\right) \right\| \leq \rho \left\| \boldsymbol{y} - \boldsymbol{y}_\lambda^*(\boldsymbol{x}) \right\|,$$

$$\lambda \left\| \nabla_{xx}^2 g\left(\boldsymbol{x}, \boldsymbol{y}_\lambda^*(\boldsymbol{x})\right) - \nabla_{xx}^2 g\left(\boldsymbol{x}, \boldsymbol{y}\right) \right\| \leq \lambda\rho \left\| \boldsymbol{y} - \boldsymbol{y}_\lambda^*(\boldsymbol{x}) \right\|,$$

$$\lambda \left\| \nabla_{xx}^2 g\left(\boldsymbol{x}, \boldsymbol{y}^*(\boldsymbol{x})\right) - \nabla_{xx}^2 g\left(\boldsymbol{x}, \boldsymbol{w}\right) \right\| \leq \lambda\rho \left\| \boldsymbol{w} - \boldsymbol{y}^*(\boldsymbol{x}) \right\|,$$

and

$$\left\| \nabla_{xy}^2 g\left(\boldsymbol{x}, \boldsymbol{y}^*(\boldsymbol{x})\right) \left[\nabla_{yy}^2 g\left(\boldsymbol{x}, \boldsymbol{y}^*(\boldsymbol{x})\right)\right]^{-1} \nabla_{yx}^2 g\left(\boldsymbol{x}, \boldsymbol{y}^*(\boldsymbol{x})\right) - \nabla_{xy}^2 g\left(\boldsymbol{x}, \boldsymbol{w}\right) \left[\nabla_{yy}^2 g\left(\boldsymbol{x}, \boldsymbol{w}\right)\right]^{-1} \nabla_{yx}^2 g\left(\boldsymbol{x}, \boldsymbol{w}\right) \right\|$$

$$\leq \left\| \nabla_{xy}^2 g\left(\boldsymbol{x}, \boldsymbol{y}^*(\boldsymbol{x})\right) \left[\nabla_{yy}^2 g\left(\boldsymbol{x}, \boldsymbol{y}^*(\boldsymbol{x})\right)\right]^{-1} \right\| \left\| \nabla_{yx}^2 g\left(\boldsymbol{x}, \boldsymbol{y}^*(\boldsymbol{x})\right) - \nabla_{yx}^2 g\left(\boldsymbol{x}, \boldsymbol{w}\right) \right\|$$

$$+ \left\| \nabla_{yx}^2 g\left(\boldsymbol{x}, \boldsymbol{w}\right) \right\| \left\| \nabla_{xy}^2 g\left(\boldsymbol{x}, \boldsymbol{y}^*(\boldsymbol{x})\right) \left[\nabla_{yy}^2 g\left(\boldsymbol{x}, \boldsymbol{y}^*(\boldsymbol{x})\right)\right]^{-1} - \nabla_{xy}^2 g\left(\boldsymbol{x}, \boldsymbol{w}\right) \left[\nabla_{yy}^2 g\left(\boldsymbol{x}, \boldsymbol{w}\right)\right]^{-1} \right\|$$

$$\leq \left( \frac{2\ell\rho}{\mu} + \frac{\ell^2\rho}{\mu^2} \right) \left\| \boldsymbol{w} - \boldsymbol{y}^*(\boldsymbol{x}) \right\|,$$

and

$$\left\| \nabla_{xy}^2 \mathcal{L}_\lambda \left(\boldsymbol{x}, \boldsymbol{y}_\lambda^*(\boldsymbol{x})\right) \left[\nabla_{yy}^2 \mathcal{L}_\lambda \left(\boldsymbol{x}, \boldsymbol{y}_\lambda^*(\boldsymbol{x})\right)\right]^{-1} \nabla_{yx}^2 \mathcal{L}_\lambda \left(\boldsymbol{x}, \boldsymbol{y}_\lambda^*(\boldsymbol{x})\right) - \nabla_{xy}^2 \mathcal{L}_\lambda \left(\boldsymbol{x}, \boldsymbol{y}\right) \left[\nabla_{yy}^2 \mathcal{L}_\lambda \left(\boldsymbol{x}, \boldsymbol{y}\right)\right]^{-1} \nabla_{yx}^2 \mathcal{L}_\lambda \left(\boldsymbol{x}, \boldsymbol{y}\right) \right\|$$

$$\leq \left\| \nabla_{xy}^2 \mathcal{L}_\lambda \left(\boldsymbol{x}, \boldsymbol{y}_\lambda^*(\boldsymbol{x})\right) \left[\nabla_{yy}^2 \mathcal{L}_\lambda \left(\boldsymbol{x}, \boldsymbol{y}_\lambda^*(\boldsymbol{x})\right)\right]^{-1} \right\| \left\| \nabla_{yx}^2 \mathcal{L}_\lambda \left(\boldsymbol{x}, \boldsymbol{y}_\lambda^*(\boldsymbol{x})\right) - \nabla_{yx}^2 \mathcal{L}_\lambda \left(\boldsymbol{x}, \boldsymbol{y}\right) \right\|$$

$$+ \left\| \nabla_{yx}^2 \mathcal{L}_\lambda \left(\boldsymbol{x}, \boldsymbol{y}\right) \right\| \left\| \nabla_{xy}^2 \mathcal{L}_\lambda \left(\boldsymbol{x}, \boldsymbol{y}_\lambda^*(\boldsymbol{x})\right) \left[\nabla_{yy}^2 \mathcal{L}_\lambda \left(\boldsymbol{x}, \boldsymbol{y}_\lambda^*(\boldsymbol{x})\right)\right]^{-1} - \nabla_{xy}^2 \mathcal{L}_\lambda \left(\boldsymbol{x}, \boldsymbol{y}\right) \left[\nabla_{yy}^2 \mathcal{L}_\lambda \left(\boldsymbol{x}, \boldsymbol{y}\right)\right]^{-1} \right\|$$

$$\leq \left[ \frac{2(\ell + \lambda\ell)\left(\rho + \lambda\rho\right)}{\lambda\mu} + \left(\ell + \lambda\ell\right)\left( \frac{2\left(\rho + \lambda\rho\right)}{\lambda\mu} + \frac{4\left(\rho + \lambda\rho\right)\left(\ell + \lambda\ell\right)}{\lambda^2\mu^2} \right) \right] \left\| \boldsymbol{y} - \boldsymbol{y}_\lambda^*(\boldsymbol{x}) \right\|.$$

Combining the above inequations, we have

$$\left\|\nabla^2 \mathcal{L}_\lambda^* (\boldsymbol{x}) - \mathbf{H}(\boldsymbol{x}; \boldsymbol{y}, \boldsymbol{w})\right\| \le C_1 \left\|\boldsymbol{w} - \boldsymbol{y}^* (\boldsymbol{x})\right\| + C_2 \left\|\boldsymbol{y} - \boldsymbol{y}_\lambda^* (\boldsymbol{x})\right\|,$$

where $C_1 = \lambda\rho + \frac{2\ell\rho}{\mu} + \frac{\ell^2\rho}{\mu^2}$, $C_2 = \rho + \lambda\rho + (\ell + \lambda\ell)\left(\frac{4(\rho+\lambda\rho)}{\lambda\mu} + \frac{4(\rho+\lambda\rho)(\ell+\lambda\ell)}{\lambda^2\mu^2}\right)$. $\qquad\square$

## D.2. The Proof of Lemma 3.3

*Proof.* We first use induction to show that

$$\left\|\boldsymbol{y}_t - \boldsymbol{y}_\lambda^* (\boldsymbol{x}_t)\right\| \le \tilde{\epsilon}, \left\|\boldsymbol{w}_t - \boldsymbol{y}^* (\boldsymbol{x}_t)\right\| \le \tilde{\epsilon} \tag{23}$$

holds for any $t \ge 0$. For $t = 0$, Lemma C.1 directly implies

$$\left\|\boldsymbol{y}_0 - \boldsymbol{y}_\lambda^* (\boldsymbol{x}_0)\right\|$$
$$\le \sqrt{\kappa_2 + 1}\left(1 - \frac{1}{\sqrt{\kappa_2}}\right)^{K_0^2/2}\left\|\boldsymbol{y}_{-1} - \boldsymbol{y}_\lambda^* (\boldsymbol{x}_0)\right\|$$
$$= \sqrt{\kappa_2 + 1}\left(1 - \frac{1}{\sqrt{\kappa_2}}\right)^{K_0^2/2}\left\|\boldsymbol{y}_\lambda^* (\boldsymbol{x}_0)\right\|$$
$$\le \tilde{\epsilon},$$

$$\left\|\boldsymbol{w}_0 - \boldsymbol{y}^* (\boldsymbol{x}_0)\right\|$$
$$\le \sqrt{\kappa_1 + 1}\left(1 - \frac{1}{\sqrt{\kappa_1}}\right)^{K_0^1/2}\left\|\boldsymbol{w}_{-1} - \boldsymbol{y}^* (\boldsymbol{x}_0)\right\|$$
$$= \sqrt{\kappa_1 + 1}\left(1 - \frac{1}{\sqrt{\kappa_1}}\right)^{K_0^1/2}\left\|\boldsymbol{y}^* (\boldsymbol{x}_0)\right\|$$
$$\le \tilde{\epsilon}.$$

The above two blocks of inequalities are justified as follows: the first inequality is based on Lemma C.1; the second equation uses the initialization of $\boldsymbol{y}_{-1}$ and $\boldsymbol{w}_{-1}$ ; the last step use the definition of $K_0^1$, $K_0^2$ and $\tilde{\epsilon}$.

Suppose it holds that $\left\|\boldsymbol{w}_{t-1} - \boldsymbol{y}^* (\boldsymbol{x}_{t-1})\right\| \le \tilde{\epsilon}$ and $\left\|\boldsymbol{y}_{t-1} - \boldsymbol{y}_\lambda^* (\boldsymbol{x}_{t-1})\right\| \le \tilde{\epsilon}$ for any $t = t' - 1$, then we have

$$\left\|\boldsymbol{w}_{t'} - \boldsymbol{y}^* (\boldsymbol{x}_{t'})\right\|$$
$$\le \sqrt{\kappa_1 + 1}\left(1 - \frac{1}{\sqrt{\kappa_1}}\right)^{K_{t'}^1/2}\left\|\boldsymbol{w}_{t'-1} - \boldsymbol{y}^* (\boldsymbol{x}_{t'})\right\|$$
$$\le \sqrt{\kappa_1 + 1}\left(1 - \frac{1}{\sqrt{\kappa_1}}\right)^{K_{t'}^1/2}\left(\left\|\boldsymbol{w}_{t'-1} - \boldsymbol{y}^* (\boldsymbol{x}_{t'-1})\right\| + \left\|\boldsymbol{y}^* (\boldsymbol{x}_{t'-1}) - \boldsymbol{y}^* (\boldsymbol{x}_{t'})\right\|\right)$$
$$\le \sqrt{\kappa_1 + 1}\left(1 - \frac{1}{\sqrt{\kappa_1}}\right)^{K_{t'}^1/2}\left(\tilde{\epsilon} + \kappa \left\|\boldsymbol{x}_{t'-1} - \boldsymbol{x}_{t'}\right\|\right)$$
$$= \sqrt{\kappa_1 + 1}\left(1 - \frac{1}{\sqrt{\kappa_1}}\right)^{K_{t'}^1/2}\left(\tilde{\epsilon} + \kappa \left\|\boldsymbol{s}_{t'-1}^*\right\|\right) \le \tilde{\epsilon},$$

$$\left\|\boldsymbol{y}_{t'} - \boldsymbol{y}_\lambda^* (\boldsymbol{x}_{t'})\right\|$$
$$\le \sqrt{\kappa_2 + 1}\left(1 - \frac{1}{\sqrt{\kappa_2}}\right)^{K_{t'}^2/2}\left\|\boldsymbol{y}_{t'-1} - \boldsymbol{y}_\lambda^* (\boldsymbol{x}_{t'})\right\|$$

$$\leq \sqrt{\kappa_2 + 1} \left(1 - \frac{1}{\sqrt{\kappa_2}}\right)^{K_{t'}^2/2} \left(\|\boldsymbol{y}_{t'-1} - \boldsymbol{y}^*(\boldsymbol{x}_{t'-1})\| + \|\boldsymbol{y}_\lambda^*(\boldsymbol{x}_{t'-1}) - \boldsymbol{y}_\lambda^*(\boldsymbol{x}_{t'})\|\right)$$

$$\leq \sqrt{\kappa_2 + 1} \left(1 - \frac{1}{\sqrt{\kappa_2}}\right)^{K_{t'}^2/2} \left(\tilde{\epsilon} + 4\kappa \|\boldsymbol{x}_{t'-1} - \boldsymbol{x}_{t'}\|\right)$$

$$= \sqrt{\kappa_2 + 1} \left(1 - \frac{1}{\sqrt{\kappa_2}}\right)^{K_{t'}^2/2} \left(\tilde{\epsilon} + 4\kappa \|\boldsymbol{s}_{t'-1}^*\|\right) \leq \tilde{\epsilon}.$$

The above two blocks of inequalities are justified as follows: the first inequality is based on Lemma C.1; the second one uses triangle inequality; the third one is based on the hypothesis of induction and Proposition 2.3 and Lemma C.3; the last step uses the definition of $K_t^1$, $K_t^2$ and $\tilde{\epsilon}$.

Combining inequality (23) with Lemma 3.1, we obtain

$$\|\nabla \mathcal{L}_\lambda^*(\boldsymbol{x}_t) - \mathbf{g}(\boldsymbol{x}_t; \boldsymbol{y}_t, \boldsymbol{w}_t)\| \leq 2\lambda\ell \|\boldsymbol{y}_t - \boldsymbol{y}_\lambda^*(\boldsymbol{x}_t)\| + \lambda\ell \|\boldsymbol{w}_t - \boldsymbol{y}^*(\boldsymbol{x}_t)\| \leq C_g\epsilon,$$

$$\|\nabla^2 \mathcal{L}_\lambda^*(\boldsymbol{x}_t) - \mathbf{H}(\boldsymbol{x}_t; \boldsymbol{y}_t, \boldsymbol{w}_t)\| \leq C_1 \|\boldsymbol{w}_t - \boldsymbol{y}^*(\boldsymbol{x}_t)\| + C_2 \|\boldsymbol{y}_t - \boldsymbol{y}_\lambda^*(\boldsymbol{x}_t)\| \leq C_H\sqrt{M}\epsilon.$$

$\square$

### D.3. The Proof of Theorem 3.4

*Proof.* Let $M = \Omega(\bar{\rho})$, $T = \Theta\left((\varphi(\boldsymbol{x}_0) - \varphi^*)\sqrt{M}\epsilon^{-3/2}\right)$ and the setting of $\lambda$, then we can prove that the output $\hat{\boldsymbol{x}}$ of Algorithm 2 is an $\left((\mathcal{O}(\epsilon), \mathcal{O}(\kappa^{2.5}\bar{\ell}^{0.5}\epsilon^{0.5}))\right)$-SOSP of $\varphi(\cdot)$.

Since the algorithm 2 could find an $(\epsilon, \sqrt{M\epsilon})$-SOSP of $\mathcal{L}_\lambda^*(\boldsymbol{x})$ in Lemma 3.2, then we have

$$\|\nabla \mathcal{L}_\lambda^*(\boldsymbol{x})\| \leq \epsilon, \quad \nabla^2 \mathcal{L}_\lambda^*(\boldsymbol{x}) \succeq -\sqrt{M\epsilon}I.$$

According to Proposition 2.4, we have

$$\|\nabla \mathcal{L}_\lambda^*(\boldsymbol{x}) - \nabla\varphi(\boldsymbol{x})\| = \mathcal{O}\left(\frac{\bar{\ell}\kappa^3}{\lambda}\right), \quad \forall \boldsymbol{x} \in \mathbb{R}^{d_x},$$

$$\|\nabla^2 \mathcal{L}_\lambda^*(\boldsymbol{x}) - \nabla^2\varphi(\boldsymbol{x})\| = \mathcal{O}\left(\frac{\bar{\ell}\kappa^5}{\lambda}\right), \quad \forall \boldsymbol{x} \in \mathbb{R}^{d_x},$$

$$|\mathcal{L}_\lambda^*(\boldsymbol{x}) - \varphi(\boldsymbol{x})| = \mathcal{O}\left(\frac{\bar{\ell}\kappa^2}{\lambda}\right), \quad \forall \boldsymbol{x} \in \mathbb{R}^{d_x}.$$

With $\lambda \geq \bar{\ell}\kappa^3/\epsilon$, we have

$$\begin{aligned}
\|\nabla\varphi(\boldsymbol{x})\| &= \|\nabla\varphi(\boldsymbol{x}) - \nabla\mathcal{L}_\lambda^*(\boldsymbol{x}) + \nabla\mathcal{L}_\lambda^*(\boldsymbol{x})\| \\
&\leq \|\nabla\mathcal{L}_\lambda^*(\boldsymbol{x}) - \nabla\varphi(\boldsymbol{x})\| + \|\nabla\mathcal{L}_\lambda^*(\boldsymbol{x})\| \\
&\leq \mathcal{O}\left(\frac{\bar{\ell}\kappa^3}{\lambda}\right) + \epsilon \\
&\leq \mathcal{O}(\epsilon).
\end{aligned}$$

With $\lambda \geq \bar{\ell}\kappa^5/\sqrt{M\epsilon}$, we have

$$\nabla^2\varphi(\boldsymbol{x}) \succeq \nabla^2\mathcal{L}_\lambda^*(\boldsymbol{x}) - \mathcal{O}\left(\frac{\bar{\ell}\kappa^5}{\lambda}\right) I \succeq -\sqrt{M\epsilon}I - \mathcal{O}\left(\frac{\bar{\ell}\kappa^5}{\lambda}\right) I \succeq -\mathcal{O}(\sqrt{M\epsilon})I.$$

With $\lambda \geq \bar{\ell}\kappa^2/\Delta$, we have

$$\mathcal{L}_\lambda^*(\boldsymbol{x}_0) - \min\mathcal{L}_\lambda^*(\boldsymbol{x}) = \mathcal{L}_\lambda^*(\boldsymbol{x}_0) - \min\mathcal{L}_\lambda^*(\boldsymbol{x}) + \varphi(\boldsymbol{x}_0) - \varphi^* - \varphi(\boldsymbol{x}_0) + \varphi^*$$

$$= \Delta + 2\mathcal{O}\left(\frac{\bar{\ell}\kappa^2}{\lambda}\right)$$

$$= \mathcal{O}(\Delta).$$

We now proceed to establish the first-order oracle complexity.

$$\sum_{t=0}^{T-1}(K_t^1 + K_t^2)$$

$$\leq 4\sqrt{3\kappa}\left[\log\left(\frac{\sqrt{3\kappa+1}}{\tilde{\epsilon}}R\right) + \sum_{t=1}^{T}\log\left(\sqrt{3\kappa+1} + \frac{4\kappa\sqrt{3\kappa+1}}{\tilde{\epsilon}}\left\|s_{t-1}^*\right\|\right)\right] + 2T$$

$$= \frac{4\sqrt{3\kappa}}{3}\left[3\log\left(\frac{\sqrt{3\kappa+1}}{\tilde{\epsilon}}R\right) + \sum_{t=1}^{T}\log\left(\sqrt{3\kappa+1} + \frac{4\kappa\sqrt{3\kappa+1}}{\tilde{\epsilon}}\left\|s_{t-1}^*\right\|\right)^3\right] + 2T$$

$$\leq \frac{4\sqrt{3\kappa}}{3}\left[3\log\left(\frac{\sqrt{3\kappa+1}}{\tilde{\epsilon}}R\right) + \sum_{t=1}^{T}\log\left(8(3\kappa+1)^{1.5} + \frac{8(4\kappa)^3(3\kappa+1)^{1.5}}{\tilde{\epsilon}^3}\left\|s_{t-1}^*\right\|^3\right)\right] + 2T$$

$$= \frac{4\sqrt{3\kappa}}{3}\left[3\log\left(\frac{\sqrt{3\kappa+1}}{\tilde{\epsilon}}R\right) + \log\left(\prod_{t=1}^{T}\left(8(3\kappa+1)^{1.5} + \frac{8(4\kappa)^3(3\kappa+1)^{1.5}}{\tilde{\epsilon}^3}\left\|s_{t-1}^*\right\|^3\right)\right)\right] + 2T$$

$$\leq \frac{4\sqrt{3\kappa}}{3}\left[3\log\left(\frac{\sqrt{3\kappa+1}}{\tilde{\epsilon}}R\right) + \log\left(\frac{1}{T}\sum_{t=1}^{T}\left(8(3\kappa+1)^{1.5} + \frac{8(4\kappa)^3(3\kappa+1)^{1.5}}{\tilde{\epsilon}^3}\left\|s_{t-1}^*\right\|^3\right)\right)^T\right] + 2T$$

$$= \frac{4\sqrt{3\kappa}T}{3}\left[\frac{3}{T}\log\left(\frac{\sqrt{3\kappa+1}}{\tilde{\epsilon}}R\right) + \log\left(8(3\kappa+1)^{1.5} + \frac{8(4\kappa)^3(3\kappa+1)^{1.5}}{T\tilde{\epsilon}^3}\sum_{t=1}^{T}\left\|s_{t-1}^*\right\|^3\right)\right] + 2T,$$

where the first inequality is based on the fact $(a+b)^3 \leq 8\left(a^3 + b^3\right)$ for $a, b \geq 0$; the second inequality is based on AM–GM inequality.

Connecting the upper bound of $\sum_{t=1}^{T}\left\|s_{t-1}^*\right\|^3$ in the proof of Lemma 3.2:

$$\mathcal{L}_\lambda^*\left(x_0\right) - \min\mathcal{L}_\lambda^*(x) \geq \frac{M}{24}\sum_{t=0}^{T}\left\|s_t^*\right\|^3,$$

we have

$$\sum_{t=0}^{T-1}(K_t^1 + K_t^2)$$

$$\leq 2T + \frac{4\sqrt{3\kappa}T}{3}\left(\frac{3}{T}\log\left(\frac{\sqrt{3\kappa+1}}{\tilde{\epsilon}}R\right)\right) + \frac{4\sqrt{3\kappa}T}{3}\log\left(8(3\kappa+1)^{1.5} + \frac{192(4\kappa)^3(3\kappa+1)^{1.5}}{TM\tilde{\epsilon}^3}\Delta\right)$$

$$= \mathcal{O}\left(\sqrt{\bar{\ell}}\kappa^3\epsilon^{-1.5}\log(\bar{\ell}^{1.5}\kappa^{-3}\epsilon^{-4.5})\right) = \tilde{\mathcal{O}}\left(\sqrt{\bar{\ell}}\kappa^3\epsilon^{-1.5}\right).$$

The claim follows from the fact that we call gradient oracle for $\mathcal{O}\left(\sum_{t=0}^{T-1}(K_t^1 + K_t^2)\right)$ times and perform Hessian (inverse) and exact cubic sub-problem solver calls for $\mathcal{O}(T)$ times.

$\square$

# E. The Proof of Section 4

### E.1. The Proof of Lemma 4.2

*Proof.* We first explain the stopping condition of the Algorithm 4 with respect to $\epsilon$. When $x_{t+1}$ from Algorithm 4 is not an $(\epsilon, \sqrt{M\epsilon})$-SOSP of $\mathcal{L}_\lambda^*\left(x_{t+1}\right)$, we have $\left\|\nabla\mathcal{L}_\lambda^*\left(x_{t+1}\right)\right\| \geq \epsilon$ or $\xi\left(x_{t+1}\right) \geq \sqrt{M\epsilon}$.

We consider the gradient case, the equation (19) in Lemma C.5 and Lemma C.4 means

$$
\begin{aligned}
&\left\| \nabla \mathcal{L}_\lambda^* \left( \boldsymbol{x}_{t+1} \right) \right\| \\
&= \left\| \nabla \mathcal{L}_\lambda^* \left( \boldsymbol{x}_{t+1} \right) - \mathbf{g}(\boldsymbol{x}_t; \boldsymbol{y}_t, \boldsymbol{w}_t) - \mathbf{H}(\boldsymbol{x}_{\pi(t)}; \boldsymbol{y}_{\pi(t)}, \boldsymbol{w}_{\pi(t)}) \boldsymbol{s}_t^* - \frac{M}{2} \left\| \boldsymbol{s}_t^* \right\| \boldsymbol{s}_t^* \right\| \\
&\leq \left\| \nabla \mathcal{L}_\lambda^* \left( \boldsymbol{x}_{t+1} \right) - \nabla \mathcal{L}_\lambda^* \left( \boldsymbol{x}_t \right) - \nabla^2 \mathcal{L}_\lambda^* \left( \boldsymbol{x}_t \right) \boldsymbol{s}_t^* \right\| + \left\| \nabla \mathcal{L}_\lambda^* \left( \boldsymbol{x}_t \right) - \mathbf{g}(\boldsymbol{x}_t; \boldsymbol{y}_t, \boldsymbol{w}_t) \right\| + \frac{M}{2} \left\| \boldsymbol{s}_t^* \right\|^2 \\
&\quad + \left\| \nabla^2 \mathcal{L}_\lambda^* \left( \boldsymbol{x}_{\pi(t)} \right) \boldsymbol{s}_t^* - \mathbf{H}(\boldsymbol{x}_{\pi(t)}; \boldsymbol{y}_{\pi(t)}, \boldsymbol{w}_{\pi(t)}) \boldsymbol{s}_t^* \right\| + \left\| \nabla^2 \mathcal{L}_\lambda^* \left( \boldsymbol{x}_t \right) \boldsymbol{s}_t^* - \nabla^2 \mathcal{L}_\lambda^* \left( \boldsymbol{x}_{\pi(t)} \right) \boldsymbol{s}_t^* \right\| \\
&\leq \frac{\bar{\rho}}{2} \left\| \boldsymbol{s}_t^* \right\|^2 + \bar{C}_g \epsilon + \bar{C}_H \sqrt{M\epsilon} \left\| \boldsymbol{s}_t^* \right\| + \frac{M}{2} \left\| \boldsymbol{s}_t^* \right\|^2 + \bar{\rho} \left\| \boldsymbol{s}_t^* \right\| \left\| \boldsymbol{x}_{\pi(t)} - \boldsymbol{x}_t \right\| \\
&= \frac{\bar{\rho} + M}{2} \left\| \boldsymbol{s}_t^* \right\|^2 + \bar{C}_g \epsilon + \bar{C}_H \sqrt{M\epsilon} \left\| \boldsymbol{s}_t^* \right\| + \bar{\rho} \left\| \boldsymbol{s}_t^* \right\| \left\| \boldsymbol{x}_{\pi(t)} - \boldsymbol{x}_t \right\| \\
&\leq \frac{\bar{\rho} + M}{2} \left\| \boldsymbol{s}_t^* \right\|^2 + \bar{C}_g \epsilon + \frac{\bar{C}_H \left( \epsilon + M \left\| \boldsymbol{s}_t^* \right\|^2 \right)}{2} + \bar{\rho} \left\| \boldsymbol{s}_t^* \right\| \left\| \boldsymbol{x}_{\pi(t)} - \boldsymbol{x}_t \right\| \\
&= \frac{\left( 1 + \bar{C}_H \right) M + \bar{\rho}}{2} \left\| \boldsymbol{s}_t^* \right\|^2 + \left( \bar{C}_g + \frac{\bar{C}_H}{2} \right) \epsilon + \bar{\rho} \left\| \boldsymbol{s}_t^* \right\| \left\| \boldsymbol{x}_{\pi(t)} - \boldsymbol{x}_t \right\|.
\end{aligned}
\tag{24}
$$

Then we consider the Hessian case, the equation (20) in Lemma C.5 means:

$$
\begin{aligned}
&\nabla^2 \mathcal{L}_\lambda^* \left( \boldsymbol{x}_{t+1} \right) \\
&\succeq \mathbf{H}(\boldsymbol{x}_{\pi(t)}; \boldsymbol{y}_{\pi(t)}, \boldsymbol{w}_{\pi(t)}) - \left\| \mathbf{H}(\boldsymbol{x}_{\pi(t)}; \boldsymbol{y}_{\pi(t)}, \boldsymbol{w}_{\pi(t)}) - \nabla^2 \mathcal{L}_\lambda^* \left( \boldsymbol{x}_{t+1} \right) \right\| \boldsymbol{I} \\
&\succeq -\frac{M}{2} \left\| \boldsymbol{s}_t^* \right\| \boldsymbol{I} - \left\| \mathbf{H}(\boldsymbol{x}_{\pi(t)}; \boldsymbol{y}_{\pi(t)}, \boldsymbol{w}_{\pi(t)}) - \nabla^2 \mathcal{L}_\lambda^* \left( \boldsymbol{x}_{t+1} \right) \right\| \boldsymbol{I} \\
&\succeq -\frac{M}{2} \left\| \boldsymbol{s}_t^* \right\| \boldsymbol{I} - \left\| \mathbf{H}(\boldsymbol{x}_{\pi(t)}; \boldsymbol{y}_{\pi(t)}, \boldsymbol{w}_{\pi(t)}) - \nabla^2 \mathcal{L}_\lambda^* \left( \boldsymbol{x}_{\pi(t)} \right) \right\| \boldsymbol{I} - \left\| \nabla^2 \mathcal{L}_\lambda^* \left( \boldsymbol{x}_{\pi(t)} \right) - \nabla^2 \mathcal{L}_\lambda^* \left( \boldsymbol{x}_t \right) \right\| \boldsymbol{I} \\
&\quad - \left\| \nabla^2 \mathcal{L}_\lambda^* \left( \boldsymbol{x}_t \right) - \nabla^2 \mathcal{L}_\lambda^* \left( \boldsymbol{x}_{t+1} \right) \right\| \boldsymbol{I} \\
&\succeq -\frac{M}{2} \left\| \boldsymbol{s}_t^* \right\| \boldsymbol{I} - \bar{C}_H \sqrt{M\epsilon} \boldsymbol{I} - \bar{\rho} \| \boldsymbol{x}_{\pi(t)} - \boldsymbol{x}_t \| \boldsymbol{I} - \bar{\rho} \left\| \boldsymbol{s}_t^* \right\| \boldsymbol{I} \\
&\succeq -\frac{M + 2\bar{\rho}}{2} \left\| \boldsymbol{s}_t^* \right\| \boldsymbol{I} - \bar{C}_H \sqrt{M\epsilon} \boldsymbol{I} - \bar{\rho} \| \boldsymbol{x}_{\pi(t)} - \boldsymbol{x}_t \| \boldsymbol{I}.
\end{aligned}
\tag{25}
$$

If $\boldsymbol{x}_{t+1}$ is not an $(\epsilon, \sqrt{M\epsilon})$-SOSP, then

- if $\left\| \nabla \mathcal{L}_\lambda^* \left( \boldsymbol{x}_{t+1} \right) \right\| \geq \epsilon$, we have

$$
\epsilon \leq \frac{1}{\left( 1 - \bar{C}_g - \frac{\bar{C}_H}{2} \right)} \left( \frac{\left( 1 + \bar{C}_H \right) M + \bar{\rho}}{2} \left\| \boldsymbol{s}_t^* \right\|^2 + \bar{\rho} \left\| \boldsymbol{s}_t^* \right\| \left\| \boldsymbol{x}_{\pi(t)} - \boldsymbol{x}_t \right\| \right).
\tag{26}
$$

- if $\xi \left( \boldsymbol{x}_{t+1} \right) \geq \sqrt{M\epsilon}$, we have

$$
\epsilon \leq \frac{1}{M} \left( \frac{1}{1 - \bar{C}_H} \right)^2 \left( \frac{M + 2\bar{\rho}}{2} \left\| \boldsymbol{s}_t^* \right\| + \bar{\rho} \| \boldsymbol{x}_{\pi(t)} - \boldsymbol{x}_t \| \right)^2.
\tag{27}
$$

With $\bar{C}_g = 1/576$ and $\bar{C}_H = 1/288$, we can choose a upper bound as the stopping condition:

$$
\epsilon \leq \frac{1}{M} \left( \frac{288}{287} \right)^2 \left( \frac{M + 2\bar{\rho}}{\sqrt{2}} \left\| \boldsymbol{s}_t \right\| + \bar{\rho} \| \boldsymbol{x}_{\pi(t)} - \boldsymbol{x}_t \| \right)^2.
\tag{28}
$$

That means if

$$
\epsilon \geq \frac{1}{M} \left( \frac{288}{287} \right)^2 \left( \frac{M + 2\bar{\rho}}{\sqrt{2}} \left\| \boldsymbol{s}_t \right\| + \bar{\rho} \| \boldsymbol{x}_{\pi(t)} - \boldsymbol{x}_t \| \right)^2.
$$

then $\boldsymbol{x}_{t+1}$ from Algorithm 4 is an $(\epsilon, \sqrt{M\epsilon})$-SOSP of $\mathcal{L}_\lambda^*(\boldsymbol{x}_{t+1})$.

Next, we need to examine the difference $\mathcal{L}_\lambda^*(\boldsymbol{x}_t) - \mathcal{L}_\lambda^*(\boldsymbol{x}_{t+1})$.

For the sake of analysis, we need to take a larger upper bound on $\epsilon$:

$$
\begin{aligned}
\epsilon &\leq \frac{1}{M}(\frac{288}{287})^2 \left( \frac{M + 2\bar{\rho}}{\sqrt{2}} \|\boldsymbol{s}_t^*\| + \bar{\rho}\|\boldsymbol{x}_{\pi(t)} - \boldsymbol{x}_t\| \right)^2 \\
&= (\frac{288}{287})^2 \left( \frac{(M + 2\bar{\rho})^2}{2M} \|\boldsymbol{s}_t^*\|^2 + \frac{\sqrt{2}\,(M + 2\bar{\rho})\,\bar{\rho}}{M} \|\boldsymbol{s}_t^*\| \, \|\boldsymbol{x}_{\pi(t)} - \boldsymbol{x}_t\| + \frac{\bar{\rho}^2}{M} \|\boldsymbol{x}_{\pi(t)} - \boldsymbol{x}_t\|^2 \right) \\
&= (\frac{288}{287})^2 \left( \frac{M^2 + 4M\bar{\rho} + 4\bar{\rho}^2}{2M} \|\boldsymbol{s}_t^*\|^2 + \frac{\sqrt{2}\,(M + 2\bar{\rho})\,\bar{\rho}}{M} \|\boldsymbol{s}_t^*\| \, \|\boldsymbol{x}_{\pi(t)} - \boldsymbol{x}_t\| + \frac{\bar{\rho}^2}{M} \|\boldsymbol{x}_{\pi(t)} - \boldsymbol{x}_t\|^2 \right) \\
&\leq (\frac{288}{287})^2 \left( \left(\frac{1}{2}M + 4\bar{\rho}\right) \|\boldsymbol{s}_t^*\|^2 + 3\sqrt{2}\bar{\rho} \, \|\boldsymbol{s}_t^*\| \, \|\boldsymbol{x}_{\pi(t)} - \boldsymbol{x}_t\| + \frac{\bar{\rho}^2}{M} \|\boldsymbol{x}_{\pi(t)} - \boldsymbol{x}_t\|^2 \right).
\end{aligned}
\tag{29}
$$

Then, according to inequality (28) and $M \geq \bar{\rho}$, we will have

$$
\begin{aligned}
&\mathcal{L}_\lambda^*(\boldsymbol{x}_{t+1}) - \mathcal{L}_\lambda^*(\boldsymbol{x}_t) \\
&\leq \nabla \mathcal{L}_\lambda^*(\boldsymbol{x}_t)^\top \boldsymbol{s}_t^* + \frac{1}{2}(\boldsymbol{s}_t^*)^\top \nabla^2 \mathcal{L}_\lambda^*(\boldsymbol{x}_t)\boldsymbol{s}_t^* + \frac{\bar{\rho}}{6} \|\boldsymbol{s}_t^*\|^3 \\
&= \mathbf{g}(\boldsymbol{x}_t; \boldsymbol{y}_t, \boldsymbol{w}_t)^\top \boldsymbol{s}_t^* + (\nabla \mathcal{L}_\lambda^*(\boldsymbol{x}_t) - \mathbf{g}(\boldsymbol{x}_t; \boldsymbol{y}_t, \boldsymbol{w}_t))^\top \boldsymbol{s}_t^* + \frac{1}{2}(\boldsymbol{s}_t^*)^\top \nabla^2 \mathcal{L}_\lambda^*(\boldsymbol{x}_t)\boldsymbol{s}_t^* + \frac{\bar{\rho}}{6} \|\boldsymbol{s}_t^*\|^3 \\
&\leq - (\boldsymbol{s}_t^*)^\top \mathbf{H}(\boldsymbol{x}_{\pi(t)}; \boldsymbol{y}_{\pi(t)}, \boldsymbol{w}_{\pi(t)})\boldsymbol{s}_t^* - \frac{M}{2}\|\boldsymbol{s}_t^*\|^3 + \bar{C}_g\epsilon \|\boldsymbol{s}_t^*\| + \frac{1}{2}(\boldsymbol{s}_t^*)^\top \nabla^2 \mathcal{L}_\lambda^*(\boldsymbol{x}_t)\boldsymbol{s}_t^* + \frac{\bar{\rho}}{6} \|\boldsymbol{s}_t^*\|^3 \\
&\leq - \frac{1}{2}(\boldsymbol{s}_t^*)^\top \mathbf{H}(\boldsymbol{x}_{\pi(t)}; \boldsymbol{y}_{\pi(t)}, \boldsymbol{w}_{\pi(t)})\boldsymbol{s}_t^* - \frac{M}{4}\|\boldsymbol{s}_t^*\|^3 + \bar{C}_g\epsilon \|\boldsymbol{s}_t^*\| + \frac{1}{2}(\boldsymbol{s}_t^*)^\top \nabla^2 \mathcal{L}_\lambda^*(\boldsymbol{x}_t)\boldsymbol{s}_t^* + \frac{\bar{\rho}}{6} \|\boldsymbol{s}_t^*\|^3 \\
&\leq - \frac{M}{4}\|\boldsymbol{s}_t^*\|^3 + \bar{C}_g\epsilon \|\boldsymbol{s}_t^*\| + \frac{\bar{\rho}}{6} \|\boldsymbol{s}_t^*\|^3 \\
&\quad + \frac{1}{2}(\boldsymbol{s}_t^*)^\top \left( \nabla^2 \mathcal{L}_\lambda^*(\boldsymbol{x}_t) - \nabla^2 \mathcal{L}_\lambda^*(\boldsymbol{x}_{\pi(t)}) + \nabla^2 \mathcal{L}_\lambda^*(\boldsymbol{x}_{\pi(t)}) - \mathbf{H}(\boldsymbol{x}_{\pi(t)}; \boldsymbol{y}_{\pi(t)}, \boldsymbol{w}_{\pi(t)}) \right) \boldsymbol{s}_t^* \\
&\leq - \frac{M}{4}\|\boldsymbol{s}_t^*\|^3 + \bar{C}_g\epsilon \|\boldsymbol{s}_t^*\| + \frac{\bar{\rho}}{2}\|\boldsymbol{s}_t^*\|^2 \|\boldsymbol{x}_{\pi(t)} - \boldsymbol{x}_t\| + \frac{\bar{C}_H\sqrt{M\epsilon}}{2}\|\boldsymbol{s}_t^*\|^2 + \frac{\bar{\rho}}{6}\|\boldsymbol{s}_t^*\|^3 \\
&\leq - \frac{M}{4}\|\boldsymbol{s}_t^*\|^3 + \frac{\bar{\rho}}{2}\|\boldsymbol{s}_t^*\|^2 \|\boldsymbol{x}_{\pi(t)} - \boldsymbol{x}_t\| + \left(\bar{C}_g + \frac{\bar{C}_H}{4}\right)\epsilon \|\boldsymbol{s}_t^*\| + \frac{6\bar{C}_H M + 4\bar{\rho}}{24}\|\boldsymbol{s}_t^*\|^3,
\end{aligned}
\tag{30}
$$

where the first inequality comes from the equation (18) of Lemma C.4; the second inequality comes from the equation (19) of Lemma C.5.

We need to address the cross terms in the preceding expression to derive a larger upper bound for $\mathcal{L}_\lambda^*(\boldsymbol{x}_{t+1}) - \mathcal{L}_\lambda^*(\boldsymbol{x}_t)$.

By Young's inequality, we can obtain

$$
\frac{\bar{\rho}}{2}\|\boldsymbol{s}_t^*\|^2 \|\boldsymbol{x}_{\pi(t)} - \boldsymbol{x}_t\| = \left( \frac{M^{\frac{2}{3}}}{2 \cdot 32^{\frac{1}{3}}}\|\boldsymbol{s}_t^*\|^2 \right) \cdot \left( \frac{32^{\frac{1}{3}}\bar{\rho}}{M^{\frac{2}{3}}}\|\boldsymbol{x}_{\pi(t)} - \boldsymbol{x}_t\| \right) \leq \frac{M}{24}\|\boldsymbol{s}_t^*\|^3 + \frac{32\bar{\rho}^3}{3M^2}\|\boldsymbol{x}_{\pi(t)} - \boldsymbol{x}_t\|^3.
\tag{31}
$$

Then according to equation (29) and $M \geq \bar{\rho}$, we have

$$
\epsilon \|\boldsymbol{s}_t^*\| \leq (\frac{288}{287})^2 \left( \left(\frac{1}{2}M + 4\bar{\rho}\right) \|\boldsymbol{s}_t^*\|^3 + 3\sqrt{2}\bar{\rho} \|\boldsymbol{s}_t^*\|^2 \|\boldsymbol{x}_{\pi(t)} - \boldsymbol{x}_t\| + \frac{\bar{\rho}^2}{M}\|\boldsymbol{s}_t^*\| \, \|\boldsymbol{x}_{\pi(t)} - \boldsymbol{x}_t\|^2 \right).
\tag{32}
$$

Also by Young's inequality, we have

$$
\bar{\rho}\|\boldsymbol{s}_t^*\|^2 \|\boldsymbol{x}_{\pi(t)} - \boldsymbol{x}_t\| = \left( \frac{M^{\frac{2}{3}}}{24^{\frac{2}{3}}}\|\boldsymbol{s}_t^*\|^2 \right) \left( \frac{24^{\frac{2}{3}}\bar{\rho}}{M^{\frac{2}{3}}}\|\boldsymbol{x}_{\pi(t)} - \boldsymbol{x}_t\| \right) \leq \frac{M}{36}\|\boldsymbol{s}_t^*\|^3 + \frac{576\bar{\rho}^3}{3M^2}\|\boldsymbol{x}_{\pi(t)} - \boldsymbol{x}_t\|^3,
\tag{33}
$$

$$\frac{\bar{\rho}^2}{M}\left\|\boldsymbol{s}_t^*\right\|\left\|\boldsymbol{x}_{\pi(t)}-\boldsymbol{x}_t\right\|^2=\left(\frac{M^{\frac{1}{3}}}{36^{\frac{1}{3}}}\left\|\boldsymbol{s}_t^*\right\|\right)\left(\frac{36^{\frac{1}{3}}\bar{\rho}^2}{M^{\frac{4}{3}}}\left\|\boldsymbol{x}_{\pi(t)}-\boldsymbol{x}_t\right\|^2\right)\le\frac{M}{108}\left\|\boldsymbol{s}_t^*\right\|^3+\frac{4\bar{\rho}^3}{M^2}\left\|\boldsymbol{x}_{\pi(t)}-\boldsymbol{x}_t\right\|^3. \quad (34)$$

By connecting inequalities (33) and (34) to inequality (32), we get

$$\epsilon\left\|\boldsymbol{s}_t^*\right\|\le(\frac{288}{287})^2\left(\left((\frac{1}{2}+\frac{\sqrt{2}}{12}+\frac{1}{108})M+4\bar{\rho}\right)\left\|\boldsymbol{s}_t^*\right\|^3+\frac{\left(576\sqrt{2}+4\right)\bar{\rho}^3}{M^2}\left\|\boldsymbol{x}_{\pi(t)}-\boldsymbol{x}_t\right\|^3\right)$$
$$\le(\frac{16}{25}M+\frac{21}{5}\bar{\rho})\left\|\boldsymbol{s}_t^*\right\|^3+(\frac{288}{287})^2\frac{\left(576\sqrt{2}+4\right)\bar{\rho}^3}{M^2}\left\|\boldsymbol{x}_{\pi(t)}-\boldsymbol{x}_t\right\|^3. \quad (35)$$

By connecting inequalities (31) and (35) to inequality (30), we obtain

$$\mathcal{L}_\lambda^*\left(\boldsymbol{x}_{t+1}\right)-\mathcal{L}_\lambda^*\left(\boldsymbol{x}_t\right)$$
$$\overset{(31)}{\le}-\frac{M}{4}\left\|\boldsymbol{s}_t^*\right\|^3+\frac{M}{24}\left\|\boldsymbol{s}_t^*\right\|^3+\frac{32\bar{\rho}^3}{3M^2}\left\|\boldsymbol{x}_{\pi(t)}-\boldsymbol{x}_t\right\|^3+\frac{6\bar{C}_HM+4\bar{\rho}}{24}\left\|\boldsymbol{s}_t^*\right\|^3$$
$$+\left(\bar{C}_g+\frac{\bar{C}_H}{4}\right)\left((\frac{16}{25}M+\frac{21}{5}\bar{\rho})\left\|\boldsymbol{s}_t^*\right\|^3+(\frac{288}{287})^2\frac{\left(576\sqrt{2}+4\right)\bar{\rho}^3}{M^2}\left\|\boldsymbol{x}_{\pi(t)}-\boldsymbol{x}_t\right\|^3\right)$$
$$\le\frac{-\frac{37}{5}M+\frac{32}{5}\bar{\rho}}{36}\left\|\boldsymbol{s}_t^*\right\|^3+\frac{322\bar{\rho}^3}{25M^2}\left\|\boldsymbol{x}_{\pi(t)}-\boldsymbol{x}_t\right\|^3$$
$$\le-\frac{M}{36}\left\|\boldsymbol{s}_t^*\right\|^3+\frac{322\bar{\rho}^3}{25M^2}\left\|\boldsymbol{x}_{\pi(t)}-\boldsymbol{x}_t\right\|^3. \quad (36)$$

According to inequality (24) and (29) and $\bar{\rho}\le M$, we can get

$$\left\|\nabla\mathcal{L}_\lambda^*\left(\boldsymbol{x}_{t+1}\right)\right\|\le\frac{\left(1+\bar{C}_H\right)M+\bar{\rho}}{2}\left\|\boldsymbol{s}_t^*\right\|^2+\left(\bar{C}_g+\frac{\bar{C}_H}{2}\right)\epsilon+\bar{\rho}\left\|\boldsymbol{s}_t^*\right\|\left\|\boldsymbol{x}_{\pi(t)}-\boldsymbol{x}_t\right\|$$
$$\le\frac{\left(1+\bar{C}_H\right)M+\bar{\rho}}{2}\left\|\boldsymbol{s}_t^*\right\|^2+\bar{\rho}\left\|\boldsymbol{s}_t^*\right\|\left\|\boldsymbol{x}_{\pi(t)}-\boldsymbol{x}_t\right\|$$
$$+\frac{288}{287^2}\left(\left(\frac{1}{2}M+4\bar{\rho}\right)\left\|\boldsymbol{s}_t^*\right\|^2+3\sqrt{2}\bar{\rho}\left\|\boldsymbol{s}_t^*\right\|\left\|\boldsymbol{x}_{\pi(t)}-\boldsymbol{x}_t\right\|+\frac{\bar{\rho}^2}{M}\left\|\boldsymbol{x}_{\pi(t)}-\boldsymbol{x}_t\right\|^2\right)$$
$$\le\frac{25\left(M+\bar{\rho}\right)}{48}\left\|\boldsymbol{s}_t^*\right\|^2+\frac{49}{48}\bar{\rho}\left\|\boldsymbol{s}_t^*\right\|\left\|\boldsymbol{x}_{\pi(t)}-\boldsymbol{x}_t\right\|+\frac{1}{286}\frac{\bar{\rho}^2}{M}\left\|\boldsymbol{x}_{\pi(t)}-\boldsymbol{x}_t\right\|^2$$
$$\le\frac{25M}{24}\left\|\boldsymbol{s}_t^*\right\|^2+\frac{49}{48}\bar{\rho}\left\|\boldsymbol{s}_t^*\right\|\left\|\boldsymbol{x}_{\pi(t)}-\boldsymbol{x}_t\right\|+\frac{1}{286}\frac{\bar{\rho}^2}{M}\left\|\boldsymbol{x}_{\pi(t)}-\boldsymbol{x}_t\right\|^2.$$

Indeed, using the convexity of the function $t\mapsto t^{3/2}$ for $t\ge0$, that means $(a+b+c)^{\frac{3}{2}}\le3^{\frac{1}{2}}(a^{\frac{3}{2}}+b^{\frac{3}{2}}+c^{\frac{3}{2}})$, we obtain

$$\left\|\nabla\mathcal{L}_\lambda^*\left(\boldsymbol{x}_{t+1}\right)\right\|^{3/2}\le\left(\frac{25M}{24}\left\|\boldsymbol{s}_t^*\right\|^2+\frac{49}{48}\bar{\rho}\left\|\boldsymbol{s}_t^*\right\|\left\|\boldsymbol{x}_{\pi(t)}-\boldsymbol{x}_t\right\|+\frac{1}{286}\frac{\bar{\rho}^2}{M}\left\|\boldsymbol{x}_{\pi(t)}-\boldsymbol{x}_t\right\|^2\right)^{\frac{3}{2}}$$
$$\le\sqrt{3}\left(\frac{25M}{24}\right)^{\frac{3}{2}}\left\|\boldsymbol{s}_t^*\right\|^3+\sqrt{3}\left(\frac{49}{48}\right)^{\frac{3}{2}}\left(\bar{\rho}\left\|\boldsymbol{s}_t^*\right\|\left\|\boldsymbol{x}_{\pi(t)}-\boldsymbol{x}_t\right\|\right)^{\frac{3}{2}}$$
$$+\sqrt{3}\frac{1}{286^{\frac{3}{2}}}\frac{\bar{\rho}^3}{M^{\frac{3}{2}}}\left\|\boldsymbol{x}_{\pi(t)}-\boldsymbol{x}_t\right\|^3$$
$$\le\frac{39\sqrt{3}}{24}M^{\frac{3}{2}}\left\|\boldsymbol{s}_t^*\right\|^3+\frac{13\sqrt{3}}{24}\frac{\bar{\rho}^3}{M^{\frac{3}{2}}}\left\|\boldsymbol{x}_{\pi(t)}-\boldsymbol{x}_t\right\|^3,$$

where the bound $\left(\bar{\rho}\left\|\boldsymbol{s}_t^*\right\|\left\|\boldsymbol{x}_{\pi(t)}-\boldsymbol{x}_t\right\|\right)^{3/2}\le\frac{M^{3/2}}{2}\left\|\boldsymbol{s}_t^*\right\|^3+\frac{\bar{\rho}^3}{2M^{3/2}}\left\|\boldsymbol{x}_{\pi(t)}-\boldsymbol{x}_t\right\|^3$ is used to establish the third inequality.

Also, according to inequality (25) and (28) and $\bar{\rho} \leq M$, we can get

$$\xi\left(\mathbf{x}_{t+1}\right) \leq \frac{M + 2\bar{\rho}}{2} \|s_t^*\| + \bar{\rho}\|\boldsymbol{x}_{\pi(t)} - \boldsymbol{x}_t\| + \bar{C}_H \sqrt{M\epsilon}$$

$$\leq \left(\frac{\bar{C}_H}{\sqrt{2}\left(1 - \bar{C}_H\right)} + \frac{1}{2}\right)(M + 2\bar{\rho}) \|s_t^*\| + \frac{288}{287}\bar{\rho}\|\boldsymbol{x}_{\pi(t)} - \boldsymbol{x}_t\|$$

$$\leq \left(\frac{1}{287\sqrt{2}} + \frac{1}{2}\right)(3M) \|s_t^*\| + \frac{288}{287}\bar{\rho}\|\boldsymbol{x}_{\pi(t)} - \boldsymbol{x}_t\|.$$

Then, using convexity of the function $t \mapsto t^3$ for $t \geq 0$, that means $(a + b)^3 \leq 4(a^3 + b^3)$, we get

$$\xi\left(\boldsymbol{x}_{t+1}\right)^3 \leq \left(\left(\frac{1}{287\sqrt{2}} + \frac{1}{2}\right)(3M) \|s_t^*\| + \frac{288}{287}\bar{\rho}\|\boldsymbol{x}_{\pi(t)} - \boldsymbol{x}_t\|\right)^3$$

$$\leq 108 \left(\frac{1}{287\sqrt{2}} + \frac{1}{2}\right)^3 M^3 \|s_t^*\|^3 + 4(\frac{288}{287})^3\bar{\rho}^3\|\boldsymbol{x}_{\pi(t)} - \boldsymbol{x}_t\|^3.$$

Hence, rearranging the above equation, we obtain

$$\frac{1}{120\sqrt{3M}} \|\nabla \mathcal{L}_\lambda^*\left(\boldsymbol{x}_{t+1}\right)\|^{3/2} \leq \frac{M}{72} \|s_t^*\|^3 + \frac{\bar{\rho}^3}{216M^2}\|\boldsymbol{x}_{\pi(t)} - \boldsymbol{x}_t\|^3,$$

$$\frac{1}{987M^2}\xi\left(\boldsymbol{x}_{t+1}\right)^3 \leq \frac{M}{72} \|s_t^*\|^3 + \frac{\bar{\rho}^3}{144M^2}\|\boldsymbol{x}_{\pi(t)} - \boldsymbol{x}_t\|^3.$$

Finally, connecting with the inequality (36), we can obtain

$$\mathcal{L}_\lambda^*\left(\boldsymbol{x}_t\right) - \mathcal{L}_\lambda^*\left(\boldsymbol{x}_{t+1}\right) \geq \gamma(\boldsymbol{x}_{t+1}) + \frac{M}{72} \|s_t^*\|^3 - \frac{13\bar{\rho}^3}{M^2}\|\boldsymbol{x}_{\pi(t)} - \boldsymbol{x}_t\|^3.$$

$\square$

### E.2. The Proof of Theorem 4.3

*Proof.* Without loss of generality, we assume $T$ is a multiple of $m$, such that $m : T = mh$, therefore we can divide the method into $h$ stages, with the $i$-th stage ($1 \leq i \leq h$). And by the definition of $T$, we have

$$\|\nabla \mathcal{L}_\lambda^*\left(\boldsymbol{x}_t\right)\| \geq \epsilon \quad \text{or} \quad \xi\left(\boldsymbol{x}_t\right) \geq \sqrt{M\epsilon} \quad \text{for} \quad t = 0, \ldots, T - 1.$$

Consequently, by Lemma 4.2 we have

$$\mathcal{L}_\lambda^*(\boldsymbol{x}_t) - \mathcal{L}_\lambda^*(\boldsymbol{x}_{t+1}) \geq \gamma(\boldsymbol{x}_{t+1}) + \frac{M}{720} \|s_t^*\|^3 + \frac{9M}{720} \|s_t^*\|^3 - \frac{13\bar{\rho}^3}{M^2}\|\boldsymbol{x}_{\pi(t)} - \boldsymbol{x}_t\|^3, \text{ for } t = 0, \ldots, T - 1.$$

We first consider 1-th phase of the method, we have

$$\mathcal{L}_\lambda^*(\boldsymbol{x}_t) - \mathcal{L}_\lambda^*(\boldsymbol{x}_{t+1}) \geq \gamma(\boldsymbol{x}_{t+1}) + \frac{M}{720} \|s_t^*\|^3 + \frac{9M}{720} \|s_t^*\|^3 - \frac{13\bar{\rho}^3}{M^2}\|\boldsymbol{x}_0 - \boldsymbol{x}_t\|^3, \text{ for } t = 0, \ldots, m - 1.$$

Telescoping this bound for different $t$, and using triangle inequality for the last negative term,

$$\|\boldsymbol{x}_0 - \boldsymbol{x}_t\| \leq \sum_{i=0}^{t-1} \|\boldsymbol{x}_{i+1} - \boldsymbol{x}_i\|.$$

Then, we have

$$\mathcal{L}_\lambda^*(\boldsymbol{x}_0) - \mathcal{L}_\lambda^*(\boldsymbol{x}_m) \geq \sum_{t=0}^{m-1} \gamma(\boldsymbol{x}_{t+1}) + \frac{M}{720} \sum_{t=0}^{m-1} \|\boldsymbol{s}_t^*\|^3 + \frac{9M}{720} \sum_{t=1}^{m} r_t^3 - \frac{13\bar{\rho}^3}{M^2} \sum_{t=1}^{m} \left( \sum_{i=1}^{t} r_i \right)^3. \tag{37}$$

By using Lemma C.6 with $r_{t+1} := \|\boldsymbol{x}_{t+1} - \boldsymbol{x}_t\|$ for each $0 \leq t \leq m-1$, and let $M \geq 8(m+1)\bar{\rho}$, we have

$$\mathcal{L}_\lambda^*(\boldsymbol{x}_0) - \mathcal{L}_\lambda^*(\boldsymbol{x}_m) \geq \sum_{t=0}^{m-1} \gamma(\boldsymbol{x}_{t+1}).$$

Using the same analytical method as the first phase and the definition of $\gamma(\boldsymbol{x})$, for the $i$-th ($1 \leq i \leq t$) phase of the method with $M = \Omega(m\bar{\rho})$, we have

$$\mathcal{L}_\lambda^*\left(\boldsymbol{x}_{m(i-1)}\right) - \mathcal{L}_\lambda^*\left(\boldsymbol{x}_{mi}\right) \geq \frac{1}{987M^2} \sum_{t=0}^{m-1} \xi\left(\boldsymbol{x}_{t+1}\right)^3 \geq \frac{m}{987\sqrt{M}} \epsilon^{3/2},$$

or

$$\mathcal{L}_\lambda^*\left(\boldsymbol{x}_{m(i-1)}\right) - \mathcal{L}_\lambda^*\left(\boldsymbol{x}_{mi}\right) \geq \frac{1}{120\sqrt{3M}} \sum_{t=0}^{m-1} \|\nabla\mathcal{L}_\lambda^*\left(\boldsymbol{x}_{t+1}\right)\|^{3/2} \geq \frac{m}{120\sqrt{3M}} \epsilon^{3/2}.$$

Telescoping this bound for all phases, we obtain

$$\mathcal{L}_\lambda^*\left(\boldsymbol{x}_0\right) - \min \mathcal{L}_\lambda^*(\boldsymbol{x}) \geq \mathcal{L}_\lambda^*\left(\boldsymbol{x}_0\right) - \mathcal{L}_\lambda^*\left(\boldsymbol{x}_T\right) \geq \frac{T}{120\sqrt{3M}} \epsilon^{3/2}.$$

That means the output $\hat{\boldsymbol{x}}$ of Algorithm 4 is an $(\epsilon, \mathcal{O}(\sqrt{M\epsilon}))$-SOSP of $\mathcal{L}_\lambda^*(\boldsymbol{x})$. And using the same analysis in Theorem 3.4, we can prove the output $\hat{\boldsymbol{x}}$ of Algorithm 4 is also an $(\mathcal{O}(\epsilon), \mathcal{O}(\kappa^{2.5}\bar{\ell}^{0.5}m^{0.5}\epsilon^{0.5}))$-SOSP of $\varphi(\boldsymbol{x})$ with $T = \Theta\left((\varphi(\boldsymbol{x}_0) - \varphi^*)\sqrt{M}\epsilon^{-3/2}\right)$. It is worth noting that due to the Hessian being updated only every $m$ iterations, the second-order oracle complexities can be bounded by $\mathcal{O}(1 + \kappa^{2.5}\bar{\ell}^{0.5}m^{-0.5}\epsilon^{-1.5})$.

And from inequality (37), we can also have

$$\mathcal{L}_\lambda^*\left(\boldsymbol{x}_0\right) - \min \mathcal{L}_\lambda^*(\boldsymbol{x}) \geq \mathcal{L}_\lambda^*\left(\boldsymbol{x}_0\right) - \mathcal{L}_\lambda^*\left(\boldsymbol{x}_T\right) \geq \frac{M}{720} \sum_{t=0}^{T-1} \|\boldsymbol{s}_t^*\|^3. \tag{38}$$

Connecting the upper bound of $\sum_{t=0}^{T-1}(K_t^1 + K_t^2)$ and the upper bound of $\sum_{t=0}^{T-1} \|\boldsymbol{s}_t^*\|^3$ in equation (38), we have

$$\sum_{t=0}^{T-1}(K_t^1 + K_t^2)$$
$$\leq 2T + \frac{4\sqrt{3\kappa}T}{3}\left(\frac{3}{T}\log\left(\frac{\sqrt{3\kappa+1}}{\tilde{\epsilon}}R\right)\right) + \frac{4\sqrt{3\kappa}T}{3}\log\left(8(3\kappa+1)^{1.5} + \frac{5760(4\kappa)^3(3\kappa+1)^{1.5}}{TM\tilde{\epsilon}^3}\Delta\right)$$
$$= \mathcal{O}\left(\sqrt{m\bar{\ell}}\kappa^3\epsilon^{-1.5}\log(\bar{\ell}^{1.5}\kappa^{-3}m^{-1.5}\epsilon^{-4.5})\right) = \tilde{\mathcal{O}}\left(\sqrt{m\bar{\ell}}\kappa^3\epsilon^{-1.5}\right).$$

The claim follows from the fact that we call gradient oracle for $\mathcal{O}\left(\sum_{t=0}^{T-1}(K_t^1 + K_t^2)\right)$ times and perform Hessian (inverse) and exact cubic sub-problem solver calls for $\mathcal{O}(T)$ times. $\qquad\square$

### E.3. The Proof of Theorem 4.7

*Proof.* In the following proof, note that in minimax problems, the Hessian Lipschitz constant of $\varphi(\boldsymbol{x})$ is $\bar{\rho} = 4\sqrt{2}\kappa^3\rho$. LMCN and LFSBA follow the same approach in the proof of second-order complexity. Therefore, we only provide a brief explanation and present the necessary formulas.

We state the following facts without proof. Similar to Theorem 4.3, we have the following results:

Under Assumption 4.6, let $M \geq \bar{\rho}$ and $\bar{\rho} = 4\sqrt{2}\kappa^3\rho$ and suppose the following condition

$$\|\nabla\varphi^*(\boldsymbol{x}_t) - \mathbf{g}(\boldsymbol{x}_t; \boldsymbol{y}_t)\| \leq \bar{C}_g\epsilon \quad \text{and} \quad \|\nabla^2\varphi^*(\boldsymbol{x}_{\pi(t)}) - \mathbf{H}(\boldsymbol{x}_{\pi(t)}; \boldsymbol{y}_{\pi(t)})\| \leq \bar{C}_H\sqrt{M}\epsilon \tag{39}$$

hold with $\bar{C}_g := 1/576, \bar{C}_H := 1/288$ in Algorithm 5, then it holds that

$$\varphi^*(\boldsymbol{x}_t) - \varphi^*(\boldsymbol{x}_{t+1}) \geq \gamma(\boldsymbol{x}_{t+1}) + \frac{M}{72}\|\boldsymbol{x}_{t+1} - \boldsymbol{x}_t\|^3 - \frac{13\bar{\rho}^3}{M^2}\|\boldsymbol{x}_{\pi(t)} - \boldsymbol{x}_t\|^3, \tag{40}$$

where we denote $\gamma(\boldsymbol{x}) := \max\left\{\frac{1}{987M^2}\xi(\boldsymbol{x})^3, \frac{1}{120\sqrt{3M}}\|\nabla\varphi^*(\boldsymbol{x})\|^{3/2}\right\}$. The above result is the version of Lemma 4.2 for minimax problems and it can be proved using the same arguments.

Under the Assumption 4.6, let $\Delta := \varphi(\boldsymbol{x}_0) - \varphi^*$, $\tilde{\epsilon} = \min\left\{\bar{C}_g\epsilon/\ell, \bar{C}_H\sqrt{M}\epsilon/\rho\right\}$, $\bar{C}_g = 1/576$ and $\bar{C}_H = 1/288$, if we run Algorithm 5 with $M = \Omega(m\bar{\rho})$, $T = \Theta\left(\Delta\sqrt{M}\epsilon^{-3/2}\right)$, $\kappa_1 = \kappa$, $\ell_1 = \ell$,

$$K_t = \left\{\begin{array}{ll} \left\lceil 2\sqrt{\kappa}\log\left(\frac{\sqrt{\kappa+1}}{\tilde{\epsilon}}\|\boldsymbol{y}^*(\boldsymbol{x}_0)\|\right)\right\rceil & t = 0 \\ \left\lceil 2\sqrt{\kappa}\log\left(\frac{\sqrt{\kappa+1}}{\tilde{\epsilon}}(\tilde{\epsilon} + \kappa\|\boldsymbol{x}_t - \boldsymbol{x}_{t-1}\|)\right)\right\rceil & t \geq 1 \end{array}\right.,$$

and $\bar{\rho} = 4\sqrt{2}\kappa^3\rho$, then the output $\hat{\boldsymbol{x}}$ of Algorithm 5 is $\left(\epsilon, \kappa^{1.5}\sqrt{m\rho\epsilon}\right)$-SOSP of $\varphi(\boldsymbol{x})$. The first-order and second-order oracle complexities can be bounded by $\tilde{\mathcal{O}}\left(\kappa^2\sqrt{m\rho}\epsilon^{-1.5}\right)$ and $\mathcal{O}\left(\sqrt{\rho/m}\kappa^{1.5}\epsilon^{-1.5}\right)$, respectively. The above result is a simplified version of Theorem 4.3 for minimax problems. It can be proved using the same arguments as in Theorem 4.3, with the only difference being the substitution of the parameter $\bar{\rho}$ and minor modifications in the complexity analysis of the AGD subroutine. The overall proof strategy and technical details remain identical.

The following inequality appears in the proof of the preceding result and serves as an essential intermediate step, similar to Equation (38) in Theorem 4.3. It will be used in the subsequent analysis:

$$\varphi(\boldsymbol{x}_0) - \varphi^* \geq \varphi(\boldsymbol{x}_0) - \varphi(\boldsymbol{x}_T) \geq \frac{M}{720}\sum_{t=0}^{T-1}\|\boldsymbol{s}_t^*\|^3. \tag{41}$$

As the AGD part involves different settings, the previous proof does not directly apply; we therefore provide a new analysis of its complexity. We need to explain that the following conditions:

$$\|\nabla\varphi(\boldsymbol{x}_t) - \mathbf{g}(\boldsymbol{x}_t; \boldsymbol{y}_t)\| \leq \bar{C}_g\epsilon \quad \text{and} \quad \|\nabla^2\varphi(\boldsymbol{x}_t) - \mathbf{H}(\boldsymbol{x}_t; \boldsymbol{y}_t)\| \leq \bar{C}_H\sqrt{M}\epsilon \tag{42}$$

can be achieved by properly choosing the number of iterations $K_t$ in the AGD subroutine.

We first use induction to show that

$$\|\boldsymbol{y}_t - \boldsymbol{y}^*(\boldsymbol{x}_t)\| \leq \tilde{\epsilon} \tag{43}$$

holds for any $t \geq 0$. For $t = 0$, Lemma C.1 directly implies $\|\boldsymbol{y}_0 - \boldsymbol{y}^*(\boldsymbol{x}_0)\| \leq \tilde{\epsilon}$. Suppose it holds that $\|\boldsymbol{y}_{t-1} - \boldsymbol{y}^*(\boldsymbol{x}_{t-1})\| \leq \tilde{\epsilon}$ for any $t = t' - 1$, then we have

$$\|\boldsymbol{y}_{t'} - \boldsymbol{y}^*(\boldsymbol{x}_{t'})\|$$
$$\leq \sqrt{\kappa+1}\left(1 - \frac{1}{\sqrt{\kappa}}\right)^{K_{t'}/2}\|\boldsymbol{y}_{t'-1} - \boldsymbol{y}^*(\boldsymbol{x}_{t'})\|$$
$$\leq \sqrt{\kappa+1}\left(1 - \frac{1}{\sqrt{\kappa}}\right)^{K_{t'}/2}\left(\|\boldsymbol{y}_{t'-1} - \boldsymbol{y}^*(\boldsymbol{x}_{t'-1})\| + \|\boldsymbol{y}^*(\boldsymbol{x}_{t'-1}) - \boldsymbol{y}^*(\boldsymbol{x}_{t'})\|\right)$$
$$\leq \sqrt{\kappa+1}\left(1 - \frac{1}{\sqrt{\kappa}}\right)^{K_{t'}/2}\left(\tilde{\epsilon} + \kappa\|\boldsymbol{x}_{t'-1} - \boldsymbol{x}_{t'}\|\right)$$

$$=\sqrt{\kappa+1}\left(1-\frac{1}{\sqrt{\kappa}}\right)^{K_{t'}/2}\left(\tilde{\epsilon}+\kappa\left\|\boldsymbol{s}_{t'-1}^*\right\|\right)\leq\tilde{\epsilon},$$

where the first inequality is based on Lemma C.1; the second one use triangle inequality; the third one is based on induction hypothesis and Proposition 2.3; the last step use the definitions of $K_t$ and $\tilde{\epsilon}$.

Combining inequality (43) with Lemma C.1, Assumption 4.6, we obtain

$$\begin{aligned}&\|\nabla\varphi\left(\boldsymbol{x}_t\right)-\mathbf{g}(\boldsymbol{x}_t;\boldsymbol{y}_t)\|\\&=\|\nabla_x f\left(\boldsymbol{x}_t,\boldsymbol{y}_t\right)-\nabla_x f\left(\boldsymbol{x}_t,\boldsymbol{y}^*\left(\boldsymbol{x}_t\right)\right)\|\\&\leq\ell\left\|\boldsymbol{y}_t-\boldsymbol{y}^*\left(\boldsymbol{x}_t\right)\right\|\leq\bar{C}_g\epsilon\end{aligned}$$

and

$$\begin{aligned}&\left\|\nabla^2\varphi\left(\boldsymbol{x}_t\right)-\mathbf{H}(\boldsymbol{x}_t;\boldsymbol{y}_t)\right\|\\&=\left\|\nabla^2 f\left(\boldsymbol{x}_t,\boldsymbol{y}^*\left(\boldsymbol{x}_t\right)\right)-\nabla^2 f\left(\boldsymbol{x}_t,\boldsymbol{y}_t\right)\right\|\\&\leq\rho\left\|\boldsymbol{y}_t-\boldsymbol{y}^*\left(\boldsymbol{x}_t\right)\right\|\\&\leq\bar{C}_H\sqrt{M}\epsilon.\end{aligned}$$

The total gradient calls from AGD in Algorithm 5 satisfy

$$\begin{aligned}&\sum_{t=0}^{T-1}K_t\\&\leq 2\sqrt{\kappa}\left[\log\left(\frac{\sqrt{\kappa+1}}{\tilde{\epsilon}}\|y^*\left(\boldsymbol{x}_0\right)\|\right)+\sum_{t=1}^{T}\log\left(\sqrt{\kappa+1}+\frac{\kappa\sqrt{\kappa+1}}{\tilde{\epsilon}}\left\|\boldsymbol{s}_{t-1}^*\right\|\right)\right]+T\\&=\frac{2\sqrt{\kappa}}{3}\left[3\log\left(\frac{\sqrt{\kappa+1}}{\tilde{\epsilon}}\|y^*\left(\boldsymbol{x}_0\right)\|\right)+\sum_{t=1}^{T}\log\left(\sqrt{\kappa+1}+\frac{\kappa\sqrt{\kappa+1}}{\tilde{\epsilon}}\left\|\boldsymbol{s}_{t-1}^*\right\|\right)^3\right]+T\\&\leq\frac{2\sqrt{\kappa}}{3}\left[3\log\left(\frac{\sqrt{\kappa+1}}{\tilde{\epsilon}}\|y^*\left(\boldsymbol{x}_0\right)\|\right)+\sum_{t=1}^{T}\log\left(8(\kappa+1)^{1.5}+\frac{8\kappa^3(\kappa+1)^{1.5}}{\tilde{\epsilon}^3}\left\|\boldsymbol{s}_{t-1}^*\right\|^3\right)\right]+T\\&=\frac{2\sqrt{\kappa}}{3}\left[3\log\left(\frac{\sqrt{\kappa+1}}{\tilde{\epsilon}}\|y^*\left(\boldsymbol{x}_0\right)\|\right)+\log\left(\prod_{t=1}^{T}\left(8(\kappa+1)^{1.5}+\frac{8\kappa^3(\kappa+1)^{1.5}}{\tilde{\epsilon}^3}\left\|\boldsymbol{s}_{t-1}^*\right\|_2^3\right)\right)\right]+T\\&\leq\frac{2\sqrt{\kappa}}{3}\left[3\log\left(\frac{\sqrt{\kappa+1}}{\tilde{\epsilon}}\|y^*\left(\boldsymbol{x}_0\right)\|\right)+\log\left(\frac{1}{T}\sum_{t=1}^{T}\left(8(\kappa+1)^{1.5}+\frac{8\kappa^3(\kappa+1)^{1.5}}{\tilde{\epsilon}^3}\left\|\boldsymbol{s}_{t-1}^*\right\|^3\right)\right)^T\right]+T\\&=\frac{2\sqrt{\kappa}T}{3}\left[\frac{3}{T}\log\left(\frac{\sqrt{\kappa+1}}{\tilde{\epsilon}}\|y^*\left(\boldsymbol{x}_0\right)\|\right)+\log\left(8(\kappa+1)^{1.5}+\frac{8\kappa^3(\kappa+1)^{1.5}}{T\tilde{\epsilon}^3}\sum_{t=1}^{T}\left\|\boldsymbol{s}_{t-1}^*\right\|^3\right)\right]+T,\end{aligned}$$

where the first inequality is based on the fact $(a+b)^3\leq 8\left(a^3+b^3\right)$ for $a,b\geq 0$; the second inequality is based on AM–GM inequality.

Here we introduce $\epsilon'=2^{-2.5}\epsilon$ to eliminate the constant term $4\sqrt{2}$ in $M$. Connecting the upper bound of $\sum_{t=0}^{T-1}K_t$ and inequality (41), we have

$$\begin{aligned}&\sum_{t=0}^{T-1}K_t\\&\leq T+\frac{2\sqrt{\kappa}T}{3}\left(\frac{3}{T}\log\left(\frac{\sqrt{\kappa+1}}{\tilde{\epsilon}}\|\boldsymbol{y}^*\left(\boldsymbol{x}_0\right)\|\right)\right)+\frac{2\sqrt{\kappa}T}{3}\log\left(8(3\kappa+1)^{1.5}+\frac{5760\kappa^3(\kappa+1)^{1.5}}{TM\tilde{\epsilon}'^3}\Delta\right)\end{aligned}$$

$$= \tilde{\mathcal{O}}\left(\sqrt{\kappa M}\epsilon^{-1.5}\right) = \tilde{\mathcal{O}}\left(\kappa^2\sqrt{m\rho}\epsilon^{-1.5}\right).$$

$\square$

# F. The Details of Inexact Version of FSBA

In this section, we present the details of IFSBA method introduced in Section 3.3. It is worth emphasizing that IFSBA never explicitly constructs the Hessian ; all Hessian-related operations are carried out via Hessian–vector products, thereby avoiding any second-order oracle calls, matrix factorizations or inversions(Chen et al., 2022), as well as SVD for the projections(Huang, 2024).

## F.1. Construction of Matrix Chebyshev Polynomials Approximation

We first present the details of constructing $\mathbf{C}_{1,t}$ and $\mathbf{C}_{2,t}$ such that

$$\mathbf{C}_{1,t} \approx \left[\nabla^2_{yy} g\left(\boldsymbol{x},\boldsymbol{w}\right)\right]^{-1}, \quad \mathbf{C}_{2,t} \approx \left[\nabla^2_{yy}\mathcal{L}_\lambda\left(\boldsymbol{x},\boldsymbol{y}\right)\right]^{-1}.$$

The following lemma presents the upper bound of approximating the matrix inverse by Chebyshev polynomials.

**Lemma F.1** (Section 9.6.1 Axelsson (1996))**.** *Suppose symmetric matrix* $\mathbf{X} \in \mathbb{R}^{d\times d}$ *satisfies* $\mu'\mathbf{I} \preceq \mathbf{X} \preceq \ell'\mathbf{I}$ *with* $0 < \mu' \le \ell' < 1$*, then we have*

$$\left\|\mathbf{X}^{-1} - \left(\frac{c_0}{2}\mathbf{I} + \sum_{k=1}^{K'} c_k \mathbf{T}_k(\mathbf{Z}')\right)\right\| \le \frac{\sqrt{\ell'/\mu'} - 1}{\sqrt{\ell'\mu'}}\left(1 - \frac{2}{\sqrt{\ell'/\mu'} + 1}\right)^{K'},$$

*where* $\mathbf{Z}' = \frac{2}{\ell'-\mu'}\left(\mathbf{X} - \frac{\ell'+\mu'}{2}\mathbf{I}\right)$, $c_k = \frac{2}{\sqrt{\ell'\mu'}}\left(\frac{\sqrt{\mu'/\ell'}-1}{\sqrt{\mu'/\ell'}+1}\right)^k$ *for* $k = 0, 1, \dots, K'$*, and* $\mathbf{T}_k(\cdot)$ *are matrix Chebyshev polynomials defined by* $\mathbf{T}_0(\mathbf{Z}') := \mathbf{I}$, $\mathbf{T}_1(\mathbf{Z}') = \mathbf{Z}'$*, and* $\mathbf{T}_k(\mathbf{Z}') := 2\mathbf{Z}'\mathbf{T}_{k-1}(\mathbf{Z}') - \mathbf{T}_{k-2}(\mathbf{Z}')$ *for* $k \ge 2$*.*

Since $\mu\mathbf{I} \preceq \nabla^2_{yy} g(\boldsymbol{x},\boldsymbol{w}) \preceq \ell\mathbf{I}$ and $\frac{\lambda\mu}{2}\mathbf{I} \preceq \nabla^2_{yy}\mathcal{L}_\lambda(\boldsymbol{x},\boldsymbol{y}) \preceq (1+\lambda)\ell\mathbf{I}$, we constructed $\mathbf{C}_{1,t}$ and $\mathbf{C}_{2,t}$ according to

$$\mathbf{C}_{1,t} = \frac{c_{1,0}}{4\ell}\mathbf{I} + \frac{1}{2\ell}\sum_{k=1}^{K_1'} c_{1,k}\mathbf{T}_k(\mathbf{Z}_{1,t}) \quad \text{and} \quad \mathbf{C}_{2,t} = \frac{c_{2,0}}{4(\lambda+1)\ell}\mathbf{I} + \frac{1}{2(\lambda+1)\ell}\sum_{k=1}^{K_2'} c_{2,k}\mathbf{T}_k(\mathbf{Z}_{2,t}), \tag{44}$$

where

$$\mathbf{Z}_{1,t} = \frac{4\ell}{\ell-\mu}\left(\frac{1}{2\ell}\nabla^2_{yy}g(\boldsymbol{x}_t,\boldsymbol{y}_t) - \frac{\ell+\mu}{4\ell}\mathbf{I}\right), \quad \mathbf{Z}_{2,t} = \frac{2}{2(\lambda+1)\ell - \lambda\mu}\left(2\nabla^2_{yy}\mathcal{L}_\lambda(\boldsymbol{x}_t,\boldsymbol{y}_t) - ((\lambda+1)\ell + \frac{\lambda\mu}{2})\mathbf{I}\right),$$

and $\{c_{1,k}\}, \{c_{2,k}\}$ computed by

$$c_{1,k} = \frac{2}{\sqrt{\ell\mu}}\left(\frac{\sqrt{\mu/\ell}-1}{\sqrt{\mu/\ell}+1}\right)^k \quad \text{and} \quad c_{2,k} = \frac{2}{\sqrt{(1+\lambda)\ell\lambda\mu/2}}\left(\frac{\sqrt{\frac{\lambda\mu}{2(1+\lambda)\ell}}-1}{\sqrt{\frac{\lambda\mu}{2(1+\lambda)\ell}}+1}\right)^k.$$

Then, we are able to bound the difference between $\mathbf{C}(\boldsymbol{x}_t;\boldsymbol{y}_t,\boldsymbol{w}_t)$ and $\nabla^2\mathcal{L}_\lambda^*(\boldsymbol{x}_t)$ by combining the statements of Lemma 3.1 and Lemma F.1.

**Lemma F.2.** *Using the notation of Algorithm 3, under Assumption 2.1, we have*

$$\left\|\nabla^2\mathcal{L}_\lambda^*\left(\boldsymbol{x_t}\right) - \mathbf{C}(\boldsymbol{x}_t;\boldsymbol{y}_t,\boldsymbol{w}_t)\right\|$$
$$\le C_1\left\|\boldsymbol{w}_t - \boldsymbol{y}^*\left(\boldsymbol{x}_t\right)\right\| + C_2\left\|\boldsymbol{y}_t - \boldsymbol{y}_\lambda^*\left(\boldsymbol{x}_t\right)\right\| + \kappa\ell\left(1 - \frac{2}{\sqrt{\kappa}+1}\right)^{K_1'} + 6(\lambda+1)\kappa\ell\left(1 - \frac{2}{\sqrt{3\kappa}+1}\right)^{K_2'}.$$

### F.2. Gradient-Based Subproblem Solver

In this section, we formally present the subroutines Cubic-Solver and Final-Cubic-Solver (line 7 and line 10 in Algorithm 3) to solve the following cubic-regularized problem

$$s_t \approx \arg\min_{s \in \mathbb{R}^{d_x}} m_t(s) := g_t^\top s + \frac{1}{2} s^\top C_t s + \frac{M}{6} \|s\|^3. \tag{45}$$

We introduce the Cubic-Solver and Final-Cubic-Solver in Algorithm 6 and Algorithm 7, respectively. Cubic-Solver constructs gradient-based update to approximately solve (45) with desired accuracy in high probability. When $\Delta_t \geq -\frac{\epsilon^3}{128M}$, we run Final-Cubic-Solver to guarantee that the output $x_{t+1}$ is an $(\epsilon, \mathcal{O}(\sqrt{\epsilon}))$ SOSP of $\mathcal{L}_\lambda^*(\cdot)$.

---

**Algorithm 6** Cubic-Solver$(\mathbf{g}, \mathbf{H}, \sigma, \mathcal{K}(\epsilon, \delta'))$

---

1: **Input:** $\mathbf{g}, \mathbf{H}, \sigma, \mathcal{K}(\epsilon, \delta')$

2: **if** $\|\mathbf{g}\| \geq L^2/M$ **then**

3: $\quad R_C = -\frac{\mathbf{g}^\top \mathbf{H} \mathbf{g}}{M\|\mathbf{g}\|^2} + \sqrt{(\frac{\mathbf{g}^\top \mathbf{H} \mathbf{g}}{M\|\mathbf{g}\|^2})^2 + \frac{2\|\mathbf{g}\|}{M}}$

4: $\quad \hat{s} = -R_C \cdot \mathbf{g}/\|\mathbf{g}\|$

5: **else**

6: $\quad s_0 = 0, \ \eta = 1/(20L)$

7: $\quad \tilde{\mathbf{g}} = \mathbf{g} + \sigma \zeta$, where $\zeta \sim \text{Uniform}(\mathcal{S}^{d-1})$

8: $\quad$ **for** $k = 0, 1, \cdots, \mathcal{K}(\epsilon, \delta') - 1$ **do**

9: $\quad\quad s_{k+1} = s_k - \eta(\tilde{\mathbf{g}} + \mathbf{H} s_k + \frac{M}{2}\|s_k\| s_k)$

10: $\quad$ **end for**

11: $\quad \hat{s} = s_{\mathcal{K}(\epsilon, \delta')}$

12: **end if**

13: **Output:** $\hat{s}$ and $\Delta = \mathbf{g}^\top \hat{s} + \frac{1}{2}\hat{s}^\top \mathbf{H} \hat{s} + \frac{M}{6}\|\hat{s}\|^3$

---

---

**Algorithm 7** Final-Cubic-Solver

---

1: **Input:** $\mathbf{g}, \mathbf{H}, \epsilon$

2: $s_0 = 0, \ \mathbf{g}_0 = \mathbf{g}, \ \eta = 1/(20L)$

3: **for** $t = 0, 1, \cdots$ **do**

4: $\quad$ **if** $\|\mathbf{g}_t\| \leq \epsilon/2$ **then**

5: $\quad\quad$ **break**

6: $\quad$ **end if**

7: $\quad s_{t+1} = s_t - \eta \mathbf{g}_t$

8: $\quad \mathbf{g}_{t+1} = \mathbf{g} + \mathbf{H} s_{t+1} + \frac{M}{2}\|s_{t+1}\| s_{t+1}$

9: **end for**

10: **Output:** $\hat{s} = s_t$

---

### F.3. The Convergence Analysis

We provide the convergence analysis for IFSBA (Algorithm 3), following the same assumptions and notations as those used in section 3. We suppose $\epsilon \leq \frac{L^2}{M}$, otherwise, the second-order condition $\nabla^2 \mathcal{L}_\lambda^*(x_t) \succeq -\sqrt{M\epsilon}\mathbf{I}$ always holds and we only need to use gradient methods to find first-order stationary point.

The following lemma indicates that once $\mathbf{g}(x_t; y_t, w_t)$ and $\mathbf{C}(x_t; y_t, w_t)$ approximate $\nabla \mathcal{L}_\lambda^*(x_t)$ and $\nabla^2 \mathcal{L}_\lambda^*(x_t)$ well and Cubic-Solver iterates with sufficient steps, then IFSBA enjoys a similar iteration complexity as FSBA with high probability.

**Lemma F.3** (Theorem 3, Luo et al. (2022))**.** *Under Assumption 2.1, if we run Algorithm 3 with $\delta' = \delta/T$, $T =$*

$\left\lceil 626 \left( \mathcal{L}_\lambda^*(\boldsymbol{x}_0) - \min_{\boldsymbol{x}} \mathcal{L}_\lambda^*(\boldsymbol{x}) \right) \sqrt{M} \, \epsilon^{-1.5} \right\rceil$, *and suppose the iterations* $K_t^1, K_t^2$ *of AGD and the order* $K_1', K_2'$ *of Chebyshev-Polynomials in* (44) *are sufficiently large such that the following condition*

$$\|\nabla \mathcal{L}_\lambda^*(\boldsymbol{x}_t) - \mathbf{g}(\boldsymbol{x}_t; \boldsymbol{y}_t, \boldsymbol{w}_t)\| \leq \tilde{C}_g \epsilon, \quad \|\nabla^2 \mathcal{L}_\lambda^*(\boldsymbol{x}_t) - \mathbf{C}(\boldsymbol{x}_t; \boldsymbol{y}_t, \boldsymbol{w}_t)\| \leq \tilde{C}_H \sqrt{M} \epsilon, \tag{46}$$

*hold with* $\tilde{C}_g = 1/240$, $\tilde{C}_H = 1/200$, *and the hyperparameters of Cubic-Solver (Algorithm 6) satisfies that*

$$\eta = \frac{1}{20L}, \sigma = \frac{C_\sigma M^2 \sqrt{\epsilon^3/M^3}}{4608(4L + \sqrt{M\epsilon})}, \mathcal{K}(\epsilon, \delta') = \frac{19200L}{C_\sigma \sqrt{M\epsilon}} \left( 6\log(3 + \frac{9\sqrt{d_x}}{\delta'}) + 18\log(\frac{6L}{\sqrt{M\epsilon}}) + 14\log(\frac{48(L + \tilde{C}_H \sqrt{M\epsilon})}{C_\sigma \sqrt{M\epsilon}} + \frac{24}{C_\sigma})) \right)$$

*for some* $C_\sigma > 0$, *then the condition* $\Delta_t \geq -\frac{1}{128} \frac{\epsilon^3}{M}$ *must hold within no more than* $T = \mathcal{O}\left( \kappa^{2.5} \sqrt{\bar{\ell}} \epsilon^{-1.5} \right)$ *iterations; and the output* $\hat{\boldsymbol{x}}$ *is an* $(\epsilon, \mathcal{O}(\kappa^{2.5} \sqrt{\bar{\ell} \epsilon}))$*-SOSP of* $\mathcal{L}_\lambda^*(\cdot)$ *with probability at least* $1 - \delta$.

We provide the following lemma to satisfy the condition (46).

**Lemma F.4.** *Under Assumption 2.1, let* $\epsilon_H > 0$, $C_2 = \mathcal{O}(\lambda \bar{\ell} \kappa^2)$, $\tilde{\epsilon} = \min\left\{ \frac{\tilde{C}_g \epsilon}{2\lambda \ell}, \frac{\min\{\tilde{C}_H \sqrt{M\epsilon}, \epsilon_H L\}}{4C_2} \right\}$, $R = \max(\|\boldsymbol{y}^*(\boldsymbol{x}_0)\|, \|\boldsymbol{y}_\lambda^*(\boldsymbol{x}_0)\|)$, *and* $\Delta = \varphi(\boldsymbol{x}_0) - \varphi^*$. *if we run Algorithm 3 with the same settings as in Lemma F.3 and* $\lambda = \max\left\{ \bar{\ell} \kappa^2/\Delta, \bar{\ell} \kappa^3/\epsilon, \bar{\ell} \kappa^5/\sqrt{M\epsilon} \right\}$, $\kappa_1 = \kappa$, $\ell_1 = \ell$, $\kappa_2 = 3\kappa$, $\ell_2 = (1 + \lambda)\ell$, *the order* $K_1', K_2'$ *of Chebyshev-Polynomials in* (44) *is*

$$K_1' = K_2' = \frac{\sqrt{3\kappa} + 1}{2} \log \left( \frac{24(\lambda + 1)\kappa\ell}{\min\{\tilde{C}_H \sqrt{M\epsilon}, \epsilon_H L\}} \right)$$

*and the number of iterations of AGD subroutine as*

$$K_t^1 = K_t^2 = \begin{cases} \left\lceil 2\sqrt{3\kappa} \log\left( \frac{\sqrt{3\kappa}+1}{\tilde{\epsilon}} R \right) \right\rceil & t = 0 \\ \left\lceil 2\sqrt{3\kappa} \log\left( \frac{\sqrt{3\kappa}+1}{\tilde{\epsilon}} \left( \tilde{\epsilon} + 4\kappa \|\boldsymbol{x}_t - \boldsymbol{x}_{t-1}\| \right) \right) \right\rceil & t \geq 1 \end{cases},$$

*then the condition* (46) *in Lemma F.3 is satisfied.*

*Note that the value of* $K_1' = K_2' = O(\sqrt{\kappa})$ *corresponds to the number of Hessian-vector product calls per iteration of the cubic subproblem solver (Algorithms 6 and 7). Combining Lemma F.3, Lemma F.4, and the value of* $\mathcal{K}(\epsilon, \delta')$, *we obtain the main result for Algorithm 3 as follows.*

**Theorem F.5.** *Under Assumption 2.1, run Algorithm 3 under the same setting as in Lemma F.3 and F.4, let* $M = \Omega(\bar{\rho})$, $T = \Theta((\varphi(\boldsymbol{x}_0) - \varphi^*)\sqrt{M}\epsilon^{-3/2})$, *then* $\hat{\boldsymbol{x}}$ *is an* $((\mathcal{O}(\epsilon), \mathcal{O}(\kappa^{2.5} \bar{\ell}^{0.5} \epsilon^{0.5}))$*-SOSP of* $\varphi(\cdot)$ *with probability at least* $1 - \delta$. *In addition, the total number of* $K_t^1, K_t^2$ *can be bounded by* $\sum_{t=0}^{T-1}(K_t^1 + K_t^2) \leq \tilde{\mathcal{O}}(\kappa^3 \bar{\ell}^{0.5} \epsilon^{-1.5})$. *The complexities of the gradient calls and Hessian-vector product calls can be bounded by* $\tilde{\mathcal{O}}(\kappa^3 \bar{\ell}^{0.5} \epsilon^{-1.5})$ *and* $\mathcal{O}(\kappa^{3.5} \bar{\ell} \epsilon^{-2})$, *respectively.*

## F.4. The Proof of Lemma F.2

*Proof.* Recalling that $\mu \mathbf{I} \preceq \nabla_{yy}^2 g(\boldsymbol{x}_t, \boldsymbol{y}_t) \preceq \ell \mathbf{I}$, we estimate the inverse of the Hessian with respect to $\boldsymbol{y}$ as

$$\left( \frac{1}{2\ell} \nabla_{yy}^2 g(\boldsymbol{x}_t, \boldsymbol{y}_t) \right)^{-1} \approx \frac{c_{1,0}}{2} \mathbf{I} + \sum_{k=1}^{K_1'} c_{1,k} \mathbf{T}_k(\mathbf{Z}_{1,t}).$$

Lemma F.1 implies

$$\left\| \left( \frac{1}{2\ell} \nabla_{yy}^2 g(\boldsymbol{x}_t, \boldsymbol{y}_t) \right)^{-1} - \left( \frac{c_{1,0}}{2} \mathbf{I} + \sum_{k=1}^{K_1'} c_{1,k} \mathbf{T}_k(\mathbf{Z}_{1,t}) \right) \right\| \leq 2(\kappa - \sqrt{\kappa}) \left( 1 - \frac{2}{\sqrt{\kappa} + 1} \right)^{K_1'}.$$

Hence, we have

$$\left\| \nabla_{xy}^2 g(\boldsymbol{x}_t, \boldsymbol{y}_t) \left( \nabla_{yy}^2 g(\boldsymbol{x}_t, \boldsymbol{y}_t) \right)^{-1} \nabla_{yx}^2 g(\boldsymbol{x}_t, \boldsymbol{y}_t) - \nabla_{xy}^2 g(\boldsymbol{x}_t, \boldsymbol{y}_t) \mathbf{C}_{1,t} \nabla_{yx}^2 g(\boldsymbol{x}_t, \boldsymbol{y}_t) \right\|$$

$$
\leq \left\| \nabla_{xy}^2 g(\boldsymbol{x}_t, \boldsymbol{y}_t) \right\|^2 \left\| (\nabla_{yy}^2 g(\boldsymbol{x}_t, \boldsymbol{y}_t))^{-1} - \left( \frac{c_{1,0}}{4\ell} \mathbf{I} + \frac{1}{2\ell} \sum_{k=1}^{K_1'} c_{1,k} \mathbf{T}_k(\mathbf{Z}_{1,t}) \right) \right\|
$$

$$
\leq \ell(\kappa - \sqrt{\kappa}) \left( 1 - \frac{2}{\sqrt{\kappa} + 1} \right)^{K_1'}
$$

$$
\leq \kappa \ell \left( 1 - \frac{2}{\sqrt{\kappa} + 1} \right)^{K_1'}.
$$

Similarily, we have $\frac{\lambda\mu}{2}\mathbf{I} \preceq \nabla_{yy}^2 \mathcal{L}_\lambda (\boldsymbol{x}, \boldsymbol{y}) \preceq (1 + \lambda)\ell\mathbf{I}$. We estimate the inverse of the Hessian with respect to $\boldsymbol{y}$ as

$$
\left[ \frac{1}{2(\lambda + 1)\ell} \nabla_{yy}^2 \mathcal{L}_\lambda (\boldsymbol{x}_t, \boldsymbol{y}_t) \right]^{-1} \approx \frac{c_{2,0}}{2} \mathbf{I} + \sum_{k=1}^{K_2'} c_{2,k} \mathbf{T}_k(\mathbf{Z}_{2,t}).
$$

Lemma F.1 implies

$$
\left\| \left[ \frac{1}{2(\lambda + 1)\ell} \nabla_{yy}^2 \mathcal{L}_\lambda (\boldsymbol{x}_t, \boldsymbol{y}_t) \right]^{-1} - \left( \frac{c_{2,0}}{2} \mathbf{I} + \sum_{k=1}^{K_2'} c_{2,k} \mathbf{T}_k(\mathbf{Z}_{2,t}) \right) \right\| \leq 2(3\kappa - \sqrt{3\kappa}) \left( 1 - \frac{2}{\sqrt{3\kappa} + 1} \right)^{K_2'}.
$$

Hence, we have

$$
\left\| \nabla_{xy}^2 \mathcal{L}_\lambda (\boldsymbol{x}_t, \boldsymbol{y}_t) \left[ \nabla_{yy}^2 \mathcal{L}_\lambda (\boldsymbol{x}_t, \boldsymbol{y}_t) \right]^{-1} \nabla_{yx}^2 \mathcal{L}_\lambda (\boldsymbol{x}_t, \boldsymbol{y}_t) - \nabla_{xy}^2 \mathcal{L}_\lambda (\boldsymbol{x}_t, \boldsymbol{y}_t) \mathbf{C}_{2,t} \nabla_{yx}^2 \mathcal{L}_\lambda (\boldsymbol{x}_t, \boldsymbol{y}_t) \right\|
$$

$$
\leq \left\| \nabla_{xy}^2 \mathcal{L}_\lambda (\boldsymbol{x}_t, \boldsymbol{y}_t) \right\|^2 \left\| (\nabla_{yy}^2 \mathcal{L}_\lambda (\boldsymbol{x}_t, \boldsymbol{y}_t))^{-1} - \left( \frac{c_{2,0}}{4(\lambda + 1)\ell} \mathbf{I} + \frac{1}{2(\lambda + 1)\ell} \sum_{k=1}^{K_2'} c_{2,k} \mathbf{T}_k(\mathbf{Z}_{2,t}) \right) \right\|
$$

$$
\leq 2(\lambda + 1)\ell(3\kappa - \sqrt{3\kappa}) \left( 1 - \frac{2}{\sqrt{3\kappa} + 1} \right)^{K_2'}
$$

$$
\leq 6(\lambda + 1)\kappa\ell \left( 1 - \frac{2}{\sqrt{3\kappa} + 1} \right)^{K_2'}.
$$

Then, we have

$$
\| \mathbf{H}(\boldsymbol{x}_t; \boldsymbol{y}_t, \boldsymbol{w}_t) - \mathbf{C}(\boldsymbol{x}_t; \boldsymbol{y}_t, \boldsymbol{w}_t) \| \leq \kappa\ell \left( 1 - \frac{2}{\sqrt{\kappa} + 1} \right)^{K_1'} + 6(\lambda + 1)\kappa\ell \left( 1 - \frac{2}{\sqrt{3\kappa} + 1} \right)^{K_2'}.
$$

According to $\left\| \nabla^2 \mathcal{L}_\lambda^* (\boldsymbol{x}_t) - \mathbf{H}(\boldsymbol{x}_t; \boldsymbol{y}_t, \boldsymbol{w}_t) \right\| \leq C_1 \| \boldsymbol{w}_t - \boldsymbol{y}^* (\boldsymbol{x}_t) \| + C_2 \| \boldsymbol{y}_t - \boldsymbol{y}_\lambda^* (\boldsymbol{x}_t) \|$ for $C_1 := \mathcal{O}\left( \lambda\bar{\ell} + \bar{\ell}\kappa^2 \right)$ and $C_2 := \mathcal{O}\left( \lambda\bar{\ell}\kappa^2 \right)$. We obtain $\boldsymbol{w}_t \approx \boldsymbol{y}^*(\boldsymbol{x}_t)$ and $\boldsymbol{y}_t \approx \boldsymbol{y}_\lambda^*(\boldsymbol{x}_t)$ by AGD. Then we can bound the approximation error of $\mathbf{C}(\boldsymbol{x}_t; \boldsymbol{y}_t, \boldsymbol{w}_t)$ as follows:

$$
\left\| \nabla^2 \mathcal{L}_\lambda^* (\boldsymbol{x_t}) - \mathbf{C}(\boldsymbol{x}_t; \boldsymbol{y}_t, \boldsymbol{w}_t) \right\|
$$

$$
\leq \left\| \nabla^2 \mathcal{L}_\lambda^* (\boldsymbol{x_t}) - \mathbf{H}(\boldsymbol{x}_t; \boldsymbol{y}_t, \boldsymbol{w}_t) \right\| + \left\| \mathbf{H}(\boldsymbol{x}_t; \boldsymbol{y}_t, \boldsymbol{w}_t) - \mathbf{C}(\boldsymbol{x}_t; \boldsymbol{y}_t, \boldsymbol{w}_t) \right\|
$$

$$
\leq C_1 \| \boldsymbol{w}_t - \mathbf{y}^* (\boldsymbol{x}_t) \| + C_2 \| \boldsymbol{y}_t - \boldsymbol{y}_\lambda^* (\boldsymbol{x}_t) \| + \kappa\ell \left( 1 - \frac{2}{\sqrt{\kappa} + 1} \right)^{K_1'} + 6(\lambda + 1)\kappa\ell \left( 1 - \frac{2}{\sqrt{3\kappa} + 1} \right)^{K_2'}.
$$

$\square$

### F.5. The Proof of Lemma F.4

*Proof.* Since FSBA and IFSBA share the same procedure and analysis in the AGD part, the following result can be derived in the same manner as Lemma 3.3:

$$\|\boldsymbol{y}_t - \boldsymbol{y}_\lambda^*(\boldsymbol{x}_t)\| \le \tilde{\epsilon}, \|\boldsymbol{w}_t - \boldsymbol{y}^*(\boldsymbol{x}_t)\| \le \tilde{\epsilon} \tag{47}$$

holds for any $t \ge 0$. Combining inequality (47) with Lemma F.2, together with the definition of $K_1'$ and $K_2'$, we obtain

$$\|\nabla \mathcal{L}_\lambda^*(\boldsymbol{x}_t) - \mathbf{g}(\boldsymbol{x}_t; \boldsymbol{y}_t, \boldsymbol{w}_t)\| \le 2\lambda\ell \|\boldsymbol{y}_t - \boldsymbol{y}_\lambda^*(\boldsymbol{x}_t)\| + \lambda\ell \|\boldsymbol{w}_t - \boldsymbol{y}^*(\boldsymbol{x}_t)\| \le \tilde{C}_g\epsilon,$$

$$\|\nabla^2 \mathcal{L}_\lambda^*(\boldsymbol{x}_t) - \mathbf{C}(\boldsymbol{x}_t; \boldsymbol{y}_t, \boldsymbol{w}_t)\| \le C_1 \|\boldsymbol{w}_t - \boldsymbol{y}^*(\boldsymbol{x}_t)\| + C_2 \|\boldsymbol{y}_t - \boldsymbol{y}_\lambda^*(\boldsymbol{x}_t)\|$$

$$+ \kappa\ell \left(1 - \frac{2}{\sqrt{\kappa}+1}\right)^{K_1'} + 6(\lambda+1)\kappa\ell \left(1 - \frac{2}{\sqrt{3\kappa}+1}\right)^{K_2'}$$

$$\le \min\{\tilde{C}_H\sqrt{M\epsilon}, \epsilon_{\mathrm{H}}L\}.$$

From the above results, it can be observed that the condition (46) in Lemma F.3 is satisfied. $\qquad\square$

### F.6. The Proof of Theorem F.5

*Proof.* Let $M = \Omega(\bar{\rho})$, $T = \Theta\left((\varphi(\boldsymbol{x}_0) - \varphi^*)\sqrt{M}\epsilon^{-3/2}\right)$ and the setting of $\lambda$, then we can prove that the output $\hat{\boldsymbol{x}}$ of Algorithm 3 is an $((\mathcal{O}(\epsilon), \mathcal{O}(\kappa^{2.5}\bar{\ell}^{0.5}\epsilon^{0.5}))$-SOSP of $\varphi(\cdot)$.

Since the algorithm 3 could find an $(\epsilon, \sqrt{M\epsilon})$-SOSP of $\mathcal{L}_\lambda^*(\boldsymbol{x})$ in Lemma F.3, then we have

$$\|\nabla \mathcal{L}_\lambda^*(\boldsymbol{x})\| \le \epsilon, \quad \nabla^2 \mathcal{L}_\lambda^*(\boldsymbol{x}) \succeq -\sqrt{M\epsilon}I.$$

Following the analysis in Theorem 3.4 with the setting of $\lambda$ and Proposition 2.4 , we can show the following results:

$$\|\nabla \varphi(\boldsymbol{x})\| \le \mathcal{O}(\epsilon), \quad \nabla^2 \varphi(\boldsymbol{x}) \succeq -\mathcal{O}(\sqrt{M\epsilon})\mathbf{I}, \quad \mathcal{L}_\lambda^*(\boldsymbol{x}_0) - \min_{\boldsymbol{x}} \mathcal{L}_\lambda^*(\boldsymbol{x}) = \mathcal{O}(\Delta).$$

Since FSBA and IFSBA share the same structure in the AGD component, we can analogously to Lemma 3.3 establish the corresponding first-order oracle complexity:

$$\sum_{t=0}^{T-1}(K_t^1 + K_t^2)$$

$$= \frac{4\sqrt{3\kappa}T}{3}\left[\frac{3}{T}\log\left(\frac{\sqrt{3\kappa+1}}{\tilde{\epsilon}}R\right) + \log\left(8(3\kappa+1)^{1.5} + \frac{8(4\kappa)^3(3\kappa+1)^{1.5}}{T\tilde{\epsilon}^3}\sum_{t=1}^{T}\|\boldsymbol{s}_{t-1}\|^3\right)\right] + 2T.$$

Our Lemma F.3 corresponds to Theorem 3 in Luo et al. (2022). Under the same setting and within the proof of that theorem, the following lemma (Lemma 16 in in Luo et al. (2022)) was employed:

Under the setting of Lemma F.3, if it satisfies $\Delta_t \le -\frac{1}{128}\sqrt{\frac{\epsilon^3}{M}}$, then we have

$$\frac{M}{256}\|\boldsymbol{s}_t\|^3 \le \mathcal{L}_\lambda^*(\boldsymbol{x}_t) - \mathcal{L}_\lambda^*(\boldsymbol{x}_t + \boldsymbol{s}_t) - \frac{1}{626}\sqrt{\frac{\epsilon^3}{M}}, \tag{48}$$

with probability at least $1 - \delta'$.

Based on the above lemma, we conclude the total number of AGD calls is at most

$$\sum_{t=0}^{T-1}(K_t^1 + K_t^2)$$

$$\le \frac{2\sqrt{3\kappa}T}{3}\left(\frac{3}{T}\log\left(\frac{\sqrt{3\kappa+1}}{\tilde{\epsilon}}R\right)\right) + \frac{2\sqrt{3\kappa}T}{3}\log\left(8(3\kappa+1)^{1.5} + \frac{2048(4\kappa)^3(3\kappa+1)^{1.5}}{TM\tilde{\epsilon}^3}\Delta\right) + 2T$$

$$= \tilde{\mathcal{O}}\left(\sqrt{\kappa M}\epsilon^{-1.5}\right) = \tilde{O}\left(\kappa^3\sqrt{\bar{\ell}}\,\epsilon^{-1.5}\right).$$

The total number of Hessian-vector calls from Algorithm 3 is at most

$$T \cdot \mathcal{K}(\epsilon, \delta') \cdot (K_1' + K_2') \leq \tilde{O}\left(\kappa^{2.5}\sqrt{\bar{\ell}}\epsilon^{-1.5}\right) \cdot \tilde{O}\left(\frac{L}{\sqrt{M\epsilon}}\right) \cdot \tilde{O}(\sqrt{\kappa})$$

$$\leq \tilde{O}\left(\kappa^{2.5}\sqrt{\bar{\ell}}\epsilon^{-1.5}\right) \cdot \tilde{O}\left(\sqrt{\frac{\bar{\ell}\kappa}{\epsilon}}\right) \cdot \tilde{O}(\sqrt{\kappa})$$

$$\leq \tilde{O}\left(\bar{\ell}\kappa^{3.5}\epsilon^{-2}\right)$$

Using Lemma 8 of Tripuraneni et al. (2018), we know the total number of Hessian-vector calls from Algorithm 7 is at most

$$\tilde{\mathcal{O}}(\sqrt{\kappa}) \cdot \mathcal{O}(\frac{L^2}{M\epsilon}) = \tilde{\mathcal{O}}\left(\frac{\bar{\ell}\kappa}{\epsilon}\right),$$

which is not the leading term in total complexity for small $\epsilon$. □

# G. Experiment Details

We present the additional experiment details in this section.

**Code Availability.** The code is available at https://github.com/silas-yang9/FSBA.

## G.1. Experimental Setup for Section 5

Our experiments are carried out on a server equipped with an Intel Xeon Platinum 8352V CPU @ 2.10GHz, featuring 16 vCPUs and 120GB of memory. The GPU used is an NVIDIA RTX 4090 (24GB VRAM). We implement the algorithms using PyTorch 2.5.1 and Python 3.12, with GPU acceleration supported by CUDA 12.4. The operating system is Ubuntu 22.04.

## G.2. Additional Experiments on Synthetic Minimax Optimization

We first consider the synthetic minimax problem with $f(\cdot, \cdot)$ defined as $f(\boldsymbol{x}, \boldsymbol{y}) := w(x_3) - 10y_1^2 + x_1y_1 - 5y_2^2 + x_2y_2$, where $\boldsymbol{x} = [x_1, x_2, x_3]^\top$, $\boldsymbol{y} = [y_1, y_2]^\top$, and $w(\cdot)$ is a multi-stage function defined to be

$$w(x) = \begin{cases} \sqrt{\epsilon}(x + (L+1)\sqrt{\epsilon})^2 - \frac{1}{3}(x + (L+1)\sqrt{\epsilon})^3 - \frac{1}{3}(3L+1)\epsilon^{3/2}, & x \leq -L\sqrt{\epsilon}; \\ \epsilon x + \frac{\epsilon^{3/2}}{3}, & -L\sqrt{\epsilon} < x \leq -\sqrt{\epsilon}; \\ -\sqrt{\epsilon}x^2 - \frac{x^3}{3}, & -\sqrt{\epsilon} < x \leq 0; \\ -\sqrt{\epsilon}x^2 + \frac{x^3}{3}, & 0 < x \leq \sqrt{\epsilon}; \\ -\epsilon x + \frac{\epsilon^{3/2}}{3}, & \sqrt{\epsilon} < x \leq L\sqrt{\epsilon}; \\ \sqrt{\epsilon}(x - (L+1)\sqrt{\epsilon})^2 + \frac{1}{3}(x - (L+1)\sqrt{\epsilon})^3 - \frac{1}{3}(3L+1)\epsilon^{3/2}, & L\sqrt{\epsilon} < x; \end{cases}.$$

The hyperparameters in all methods are tuned as follows. We perform a grid search to tune the learning rates for the AGD steps, GDA, and the outer loop of PRAGDA from $\{c \times 10^i : c \in \{1, 5\}, i \in \{1, 2, 3\}\}$. The momentum parameter is selected from $\{c \times 0.1 : c \in \{1, 2, 3, 4, 5, 6, 7, 8, 9\}\}$. For LMCN, the frequency of Hessian updates $m$ is chosen from $\{1, 2, 3, 4, 5, 6, 7, 8, 9, 10\}$. For LMCN, iMCN and MCN, we set the cubic regularized Newton constant $M = 5$.

The experiments are conducted with different initial points: $(\boldsymbol{x}_1, \boldsymbol{y}_1) = ([10^{-3}, 10^{-3}, 10^{-1}]^\top, [0, 0]^\top)$ and $(\boldsymbol{x}_2, \boldsymbol{y}_2) = ([10^{-3}, 10^{-3}, 1]^\top, [0, 0]^\top)$. We compare our LMCN algorithm with the following baseline algorithms: PRAGDA (Yang et al., 2023), MCN (Luo et al., 2022), iMCN (Luo et al., 2022) and classical GDA (Lin et al., 2020). The results are shown in Figure 4.

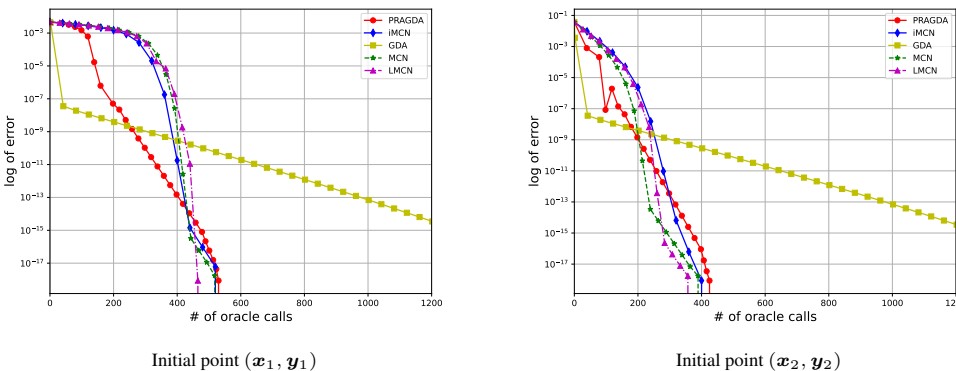

Figure 4. Comparison of LMCN and baseline algorithms in terms of oracle calls under different initial points $(\boldsymbol{x}_1, \boldsymbol{y}_1)$ and $(\boldsymbol{x}_2, \boldsymbol{y}_2)$.

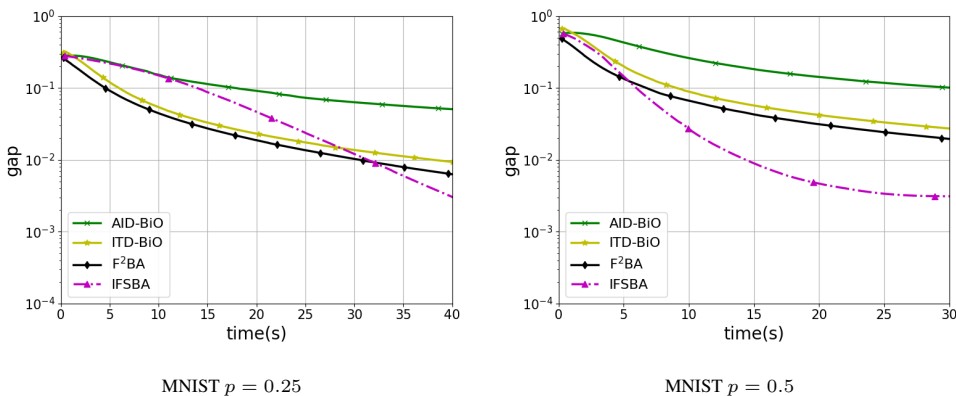

Figure 5. Comparison of various bilevel algorithms for data hypercleaning at different noise rate $p$ on "MNIST" datasets.

### G.3. Experiment Details in Section 5.1

We tune the inner-loop and outer-loop learning rate of all methods from $\{10^{-3}, 10^{-2}, 10^{-1}, 1, 10^1, 10^2, 10^3\}$, the iteration numbers of GD or AGD step from $\{5, 10, 30, 50\}$, and the iteration number of CG step from $\{5, 10, 30, 50\}$. For F$^2$BA and LFSBA, we additionally tune the multiplier $\lambda$ in $\{1, 10^1, 10^2, 10^3\}$. For LFSBA, we tune $M$ from $\{1, 10^1, 10^2, 10^3\}$ and $m$ from $\{1, 5, 10, 100\}$.

We also compare IFSBA method (Algorithm 3) with baseline methods, including ITD, AID with conjugate gradient, and F$^2$BA on "MNIST" datasets (LeCun et al., 2002). We report the results on $\mathcal{D}_{tr}$ with different noise rates $p = 25\%$ and $p = 50\%$ in Figure 5.

We follow the setting in Section 5.1 for choosing the hyperparameter. IFSBA requires additional tuning of the Cubic-Solver iterations and Matrix Chebyshev Polynomials, where the iteration steps are chosen from $\{1, 5, 10, 100\}$.

## H. Ablation Studies

### H.1. Ablation Studies on $m$

Our theoretical analysis suggests setting $m = \Theta\left(1 + \frac{d}{\sqrt{\kappa}}\right)$ to balance iteration complexity and per-iteration computational cost. To validate this choice, we conduct ablation studies on $m$ for both synthetic and data-cleaning problems. As shown in Figure 6, there is a clear trade-off in the choice of $m$. As suggested by Theorem 4.3, the choice of $m$ introduces a trade-off between iteration complexity and Hessian-related computation. Smaller $m$ improves the iteration complexity due to the $m^{1/2}$ factor, whereas larger $m$ reduces the cost of Hessian updates, as captured by the $m^{-1/2}$ factor in the second-order oracle term. In practice, moderately large values of $m$ achieve the best overall performance in terms of running time.

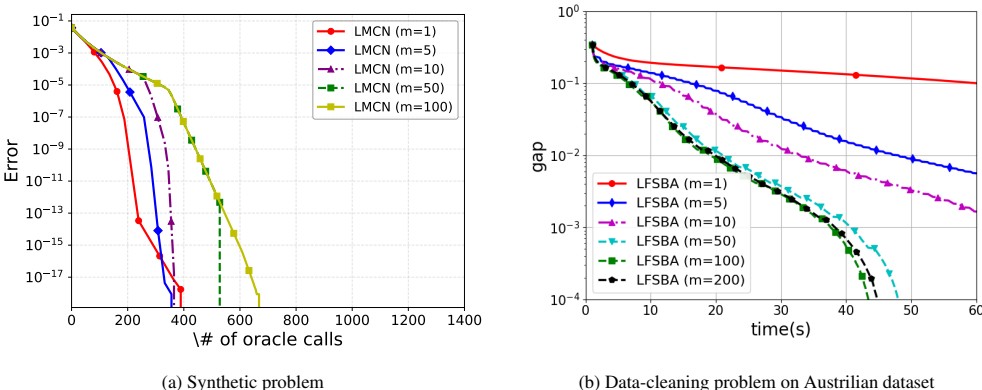

(a) Synthetic problem

(b) Data-cleaning problem on Austrilian dataset

*Figure 6.* Ablation study on the Hessian update frequency $m$.

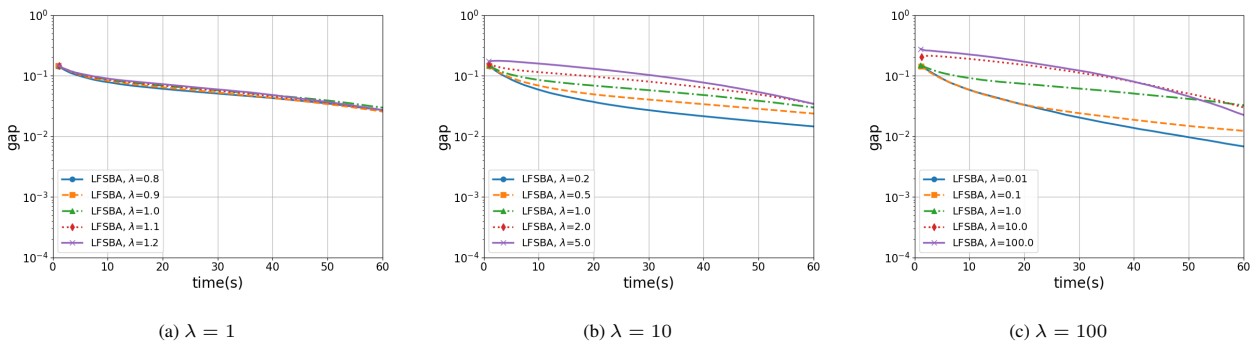

(a) $\lambda = 1$

(b) $\lambda = 10$

(c) $\lambda = 100$

*Figure 7.* Ablation study on the penalty multiplier $\lambda$ for the data-cleaning task.

## H.2. Ablation Studies on $\lambda$ and $M$

The exact values of theoretical constants, such as the smoothness parameter $\ell$, the Hessian Lipschitz parameter $\rho$, and the condition number $\kappa$, are highly problem-dependent. For example, in the data hypercleaning task, assuming the logit loss is bounded by $B = \max_{i,y} L(\langle \boldsymbol{a}_i, \boldsymbol{y} \rangle, \boldsymbol{y})$, the second-order derivatives of $g$ can be bounded as $\|\nabla^2_{xx} g(\boldsymbol{x}, \boldsymbol{y})\| \leq \frac{1}{|D_{\mathrm{tr}}|} \max_i |\sigma''(\boldsymbol{x}_i)| L_i(\boldsymbol{y}) = \mathcal{O}(B/|D_{\mathrm{tr}}|)$, $\|\nabla^2_{xy} g(\boldsymbol{x}, \boldsymbol{y})\| \leq \frac{1}{4|D_{\mathrm{tr}}|} \sqrt{\sum_{i \in D_{\mathrm{tr}}} \|\boldsymbol{a}_i\|^2}$, and $\|\nabla^2_{yy} g(\boldsymbol{x}, \boldsymbol{y})\| \leq \frac{1}{4|D_{\mathrm{tr}}|} \lambda_{\max}\left(\sum_{i \in D_{\mathrm{tr}}} \boldsymbol{a}_i \boldsymbol{a}_i^\top\right) + \mu$. These bounds provide estimates of the smoothness constant of $g$, and other problem-dependent constants can be bounded in a similar way. Nevertheless, computing tight global constants is often conservative and impractical. Therefore, in our implementation, we do not rely on strict worst-case bounds; instead, we treat the induced algorithmic parameters, including the cubic regularization parameter $M$ and the penalty multiplier $\lambda$, as tunable hyperparameters.

To examine the sensitivity of FSBA/LFSBA to these choices, we conduct additional ablation studies on the Australian dataset for the data hypercleaning task. Figures 7 and 8 report the results for different choices of $\lambda$ and $M$, respectively, while keeping the other hyperparameters fixed. We observe that FSBA/LFSBA consistently converges and achieves a small optimality gap even when these hyperparameters are varied over a wide range. Since $\lambda$ and $M$ are theoretically tied to problem-dependent constants, e.g., $M \geq c\rho$, this empirical stability demonstrates that our method is robust to the choice of these parameters in practice.

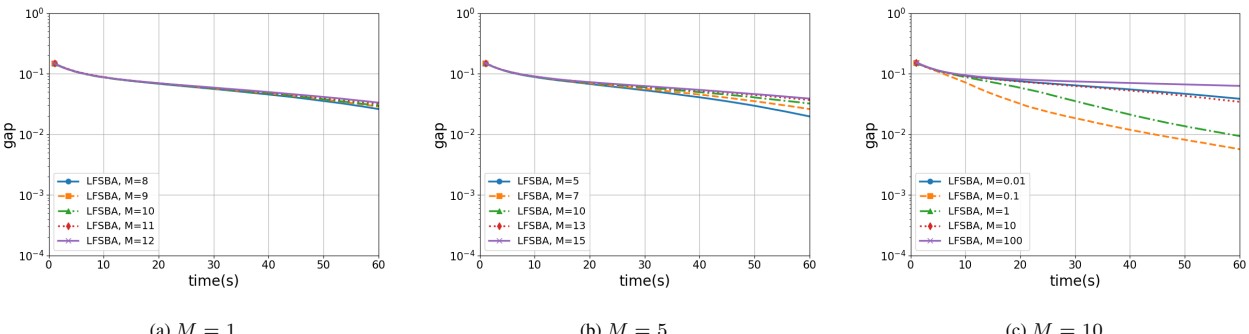

(a) $M = 1$        (b) $M = 5$        (c) $M = 10$

*Figure 8.* Ablation study on the cubic regularization parameter $M$ for the data-cleaning task.

