# OpenReview forum: "Second-Order Bilevel Optimization with Accelerated Convergence Rates"
_ICML.cc/2026/Conference — ICML 2026 regular_

### Official Review · Reviewer_t9gB · 2026-03-12

**Soundness:** 3
**Presentation:** 2
**Significance:** 3
**Originality:** 2
**Overall Recommendation:** 4
**Confidence:** 3

**Summary:**

The paper studies deterministic nonconvex-strongly-convex bilevel optimization and proposes FSBA, which applies cubic-regularized Newton to a bilevel proxy objective rather than directly to the hyper-objective. Its key technical ingredient is a bilevel-specific approximation of both the gradient and the Hessian of this proxy objective from approximate lower-level solutions, which enables an $\tilde O(\epsilon^{-1.5})$ complexity guarantee for an approximate second-order stationary point. The paper also introduces a lazy-Hessian variant LFSBA, an inexact variant IFSBA, extends the lazy idea to the nonconvex-strongly-concave minimax setting, and provides experiments on synthetic minimax, hyper-cleaning, and hyperparameter tuning.

**Compliance With Llm Reviewing Policy:**

Affirmed.

**Final Justification:**

The paper makes a technically solid contribution to second-order bilevel optimization, with a meaningful bilevel-specific approximation result and an improved convergence rate, and I found its overall significance sufficient for a borderline acceptance. The rebuttal was helpful and clarified most of my concerns, including the paper’s relation to Luo et al. (2022), but it did not materially change my overall assessment. Overall, I maintain my **weak accept** recommendation.

**Key Questions For Authors:**

1. The paper is closely related in spirit to Luo et al. (2022). Could the authors clarify more explicitly the main technical novelties of the present work relative to that paper, both in the method and in the analysis?

2. Could the authors clarify the theoretical guarantee of IFSBA, including its final complexity and how it should be compared theoretically with FSBA/LFSBA, given that it is analyzed under a different oracle model?

3. Could the authors provide more evidence or discussion on the practical benefit of LFSBA, especially how sensitive it is to the choice of $m$ and whether its gains persist beyond the oracle-cost model of Assumption 4.1?

**Limitations:**

yes

**Strengths And Weaknesses:**

The paper targets a meaningful theoretical gap: it derives an $\tilde O(\epsilon^{-1.5})$ rate, improving on the $\tilde O(\epsilon^{-1.75})$ rates reported for recent bilevel surrogate-based methods. The bilevel-specific estimator underlying Lemma 3.1 is also a meaningful contribution: even if the outer cubic-Newton template is familiar, this approximation step is what makes that framework applicable to the bilevel proxy objective. The paper also goes beyond the exact method by proposing LFSBA and transferring the lazy-Hessian idea back to the minimax setting, and the experiments are supportive overall.

The main weakness is that the paper is quite close in spirit to Luo et al. (2022): the high-level MCN recipe there already combines AGD for the inner problem, approximate first/second-order information, and a cubic-regularized Newton step, and the submission explicitly reuses a core convergence lemma from that line of work. This makes the contribution feel more like a substantive adaptation than a fully new second-order framework. The practical role of IFSBA is also less immediately clear than that of FSBA/LFSBA: while it is introduced in the main text as the scalable variant, its formal guarantee is deferred to the appendix and stated under a different oracle model based on Hessian-vector products, which makes its relation to the main results less transparent.

---

> ### Author Rebuttal · Authors · 2026-03-28
>
> We thank the reviewer for the positive evaluation and helpful feedback on our work.
>
> **To Q1 (Discussion with Luo et al. (2022))**
>
> In terms of the method:
> at high level, both this work and Luo et al. (2022) applies inexact cubic-Newton method. The difference is:
>
> 1. Luo et al. applies inexact CRN on $\Phi(x) = \max\_{y} f(x,y)$, which requires to find some $\hat{y} \approx y\^* (x)$. Our methods apply inexact CRN on the penalized function $L\_{\lambda}\^* (x) = \min \_{y}\max \_{w} (f(x,y)+  \lambda (g(x,y)-g(x,w))) $
> which requires to find $\hat{y}\approx y\^* \_{\lambda}(x), \hat{w} \approx y \^*(x)$.
>
>
> 2. Luo et al. updates the approximate Hessian $\nabla^2_{xx}f(x_t,y_t)\approx \nabla^2\Phi(x_t)$ at every iteration. In contrast, our LFSBA method reuses the approximate Hessian $H(x_t,y_t,w_t)$ from the snapshot point to trade off of the computational cost of the outer loop inexact cubic-steps and inner loop AGD steps.
>
> In terms of the analysis:
>
> 1. To construct sufficient approximation of  $\nabla\^2 L\_{\lambda}\^* (x)$ , we develop Lemma 3.1 to bound the error of $\|\|H(x,y,w) - \nabla\^2 L \_{\lambda} \^* (x)\|\|$ based on how well $y$ and $w$ approximate $y\^* \_{\lambda}(x)$ and $y \^*(x)$
>
> 2. We must control not only the error from the inexact gradient and Hessian, but also the error introduced by the lazy Hessian updates, which requires a new descent guarantee in Lemma 4.2. This analysis yields a sharper computational complexity than Luo et al. (2022) for minimax optimization.
>
> **To Q2 (Discussion on IFSBA)**
>
> The IFSBA can be viewed as an inexact version of FSBA. Because it relies exclusively on Hessian-Vector Products (HVPs) via Chebyshev polynomials, it entirely avoids explicit $d \times d$ matrix construction and inversion. We have provided its detailed complexity analysis in Appendix E.
>
> As shown in Theorem  E.5, ISFBA finds an $(\epsilon,\sqrt{\epsilon})$-SOSP within
> $\mathcal{O}(\kappa^3\bar{l}^{0.5}\epsilon^{-1.5})$ gradient calls and
> $\mathcal{O}(\kappa^{3.5}\bar{l}\epsilon^{-2})$ HVP calls.
> To compare this directly with our exact methods (where constructing the full Hessian requires $\mathcal{O}(d)$ HVPs):
>
> * FSBA requires
> $\mathcal{O}(\kappa^3\bar{l}^{0.5}\epsilon^{-1.5})$ gradient calls and
> $\mathcal{O}(\kappa^{2.5}d\bar{l}\epsilon^{-1.5})$ HVP calls.
>
> * For LFSBA, it takes
> $\mathcal{O}(\kappa^3\bar{l}^{0.5}m^{0.5}\epsilon^{-1.5})$ calls to the gradient and
> $\mathcal{O}(\kappa^{2.5}d \bar{l}^{0.5}m^{-0.5}\epsilon^{-1.5})$ calls to the Hessian-vector product.
>
> We will also provide the comparison in this computational model in revision.
>
> **To Q3**
>
> The benefits of LFSBA extend from theory to practical wall-clock time, persisting well beyond the oracle-cost model of Assumption 4.1.
>
> In terms of the gain beyond the oracle-cost model:
> In FSBA, executing the exact cubic-regularization step requires a matrix inversion/factorization, imposing a computational complexity of $\mathcal{O}(d^3)$ at every step. By contrast, as noted in Remark 4.5, LFSBA only computes the Hessian surrogate and its eigendecomposition at snapshot points. For all intermediate steps within the period $m$, the cubic-regularized Newton step drops to a computational cost of just $\tilde{\mathcal{O}}(d^2)$. This can also translate to the faster running times of LFSBA observed in our data hypercleaning experiments (Figure 2).
>
> In terms of the sensitivity to $m$
> Theoretically, our analysis suggests choosing $m = \Theta(1+d/\sqrt{\kappa})$ to optimally balance the iteration and per-step computational costs.
> Empirically, we observed that LFSBA consistently achieves shorter running times than FSBA provided $m$ is set to a moderate value (e.g., $m \in \{5, 10,100\}$).
> We will add an empirical ablation study on $m$ to the revised appendix. The following are the ablation studies on (m) for the synthetic problem and the data-cleaning problem.
>
> **Table: Oracle calls to reach log of error \(10^{-17}\) on the synthetic problem under different lazy update intervals \(m\).**
>
> | Method | Oracle calls |
> |---|---:|
> | LMCN \(m=1\)   | 364 |
> | LMCN \(m=5\) | **332** |
> | LMCN \(m=10\)  | 369 |
> | LMCN \(m=50\)  | 534 |
> | LMCN \(m=100\) | 644 |
>
> **Table: Gap values at different time points on the data-cleaning problem (\(p=50\%\)) with the australian dataset.**
>
> | Time (s) | LFSBA (\(m=1\)) | LFSBA (\(m=5\)) | LFSBA (\(m=10\)) | LFSBA (\(m=100\)) | LFSBA (\(m=200\)) |
> |---:|---:|---:|---:|---:|---:|
> | 10 | \(1.8\times 10^{-1}\) | \(1.4\times 10^{-1}\) | \(1.1\times 10^{-1}\) | **\(5.5\times 10^{-2}\)** | \(6.0\times 10^{-2}\) |
> | 20 | \(1.6\times 10^{-1}\) | \(8.0\times 10^{-2}\) | \(4.0\times 10^{-2}\) | **\(9.0\times 10^{-3}\)** | \(9.0\times 10^{-3}\) |
> | 30 | \(1.5\times 10^{-1}\) | \(3.5\times 10^{-2}\) | \(1.2\times 10^{-2}\) | **\(3.0\times 10^{-3}\)** | \(3.0\times 10^{-3}\) |
> | 40 | \(1.3\times 10^{-1}\) | \(1.7\times 10^{-2}\) | \(6.0\times 10^{-3}\) | **\(7.0\times 10^{-4}\)** | \(8.0\times 10^{-4}\) |

---

> > ### Author Rebuttal · Reviewer_t9gB · 2026-04-04
> >
> > Thank you for the detailed rebuttal. I would still consider the originality point only partially resolved, because while the rebuttal clarifies the differences, it does not materially change the fact that the central cubic-Newton machinery remains closely tied to Luo et al. (2022). That said, I maintain my weak accept recommendation.

---

> > > ### Author Response · Authors · 2026-04-04
> > >
> > > Thank you for your continued support and engaging discussion in our work.
> > >
> > > We understand the reviewer's perspective regarding the shared high-level cubic-Newton template. To address the question of originality, we wish to briefly highlight two fundamental, structural differences that distinguish our framework from Luo et al. (2022):
> > >
> > > 1. Luo et al. apply inexact CRN directly to the minimax hyperobjective $\Phi(x) = \max\_y f(x,y)$. If one were to directly translate their methodology to the bilevel hyperobjective $\varphi(x) = f(x, y\^* (x))$, computing the hyper-Hessian $\nabla\^2 \varphi(x)$ would inherently require the third-order derivatives of the lower-level function. Our methodological originality lies in shifting the CRN application away from the exact hyperobjective and onto the penalized objective function $\mathcal{L}\_ \lambda^*$. This design removes the need for third-order derivatives, solving a structural problem that minimax settings do not face.
> > >
> > > 2. Beyond the objective formulation, our LFSBA method introduces a new algorithmic design  via "lazy" Hessian updates.
> > > In standard inexact CRN (like the one that Luo et al. applies), the stopping criterion only requires the distance of the final update direction to be small enough such that $\|\|s\_t ^* \|\|=\mathcal{O}(\sqrt{\epsilon})$. However, because LFSBA reuses the Hessian from a snapshot point $\tilde{x}$, the approximation error compounds as the iterate $x_t$ drifts away from $\tilde{x}$. Consequently, LFSBA necessitates a novel stopping criterion (Algorithm 4, Line 10): combining the step size and the drift distance such that $(\|\|s_t^*\|\| + \|\|\tilde{x}-x_t\|\|)^2 = \mathcal{O}(\epsilon)$, which is very different from the prior inexact CRN methods.

---

### Official Review · Reviewer_bV4H · 2026-03-12

**Soundness:** 3
**Presentation:** 3
**Significance:** 2
**Originality:** 2
**Overall Recommendation:** 5
**Confidence:** 3

**Summary:**

The authors study a bilevel optimization problem, with standard smoothness assumptions on the lower and upper-level functions, and assume strong convexity on the lower level. The main idea of the paper is that the authors want to use second-order methods to obtain better rates or iteration complexities (compared to first-order methods), so that a clear motivation is available for the use of second-order methods. Noting that Hessian (or Hessian vector products) can be expensive, they also utilize lazy updates (where the Hessian is updated only periodically) in the spirit of Doikov et. al. This forms the second major algorithm proposed in this paper. Minmax problems are also considered. Finally, the work of the proposed algorithms is demonstrated on several examples.

**Compliance With Llm Reviewing Policy:**

Affirmed.

**Final Justification:**

The authors resolved my concerns, and I think the paper can be accepted.

**Key Questions For Authors:**

1. It seems that all the results only hold when exact knowledge of parameters like Lipschitz constants are known. This makes the algorithms unattractive to use. Can the authors explain this, and why they didn’t consider adaptive algorithms?

2. The equation at the bottom of page 3 and top of page 4 is confusing and should be reformatted

3. Algorithm statements and corresponding lemmas (e.g. Lemma 3.2): are there conditions on the input of the algorithms (e.g. $\ell_i, $K_t^I$, etc)? For which choices does Lemma 3.2 actually hold?

4. In Theorem 3.4, what does it mean “let $M=\Omega(\bar \rho)$” etc?

5. Why is Assumption 4.1 in an assumption environment? And in (9) do you mean number # 1st-order oracle calls, etc?

6. Theorem 4.7 is confusing. The result (which is in lines 374-378) only holds if the selection of parameters beginning in line 379 holds, right? Please state that upfront. Why is it written differently from the other theorems?

7. The figures are too small and barely readable, especially the legends.

**Limitations:**

yes

**Strengths And Weaknesses:**

The paper is mostly well written and seems to be technically sound. The topic itself is interesting, and the paper achieves new rates. Lazy Hessian evaluations are sensible within the context, and the authors quantify the complexity improvements w.r.t. the dimension of the problem. The results are not spectacular, and the originality is maybe a bit limited as it combines some different ideas from other papers together, but it does somewhat advance the field. Thus, the significance is not huge. A weakness of the paper is that the algorithms require exact knowledge of problem-specific constants like Lipschitz constants, condition numbers, etc., which means that it is not really realistic to apply their work except to toy problems. Also, the assumptions made seem to imply that one could write the problem as a smooth control problem with equality constraints, for which, e.g., SQP methods exist. Finally, the numerical experiments seem to be a bit academic.

---

> ### Author Rebuttal · Authors · 2026-03-28
>
> We thank the reviewer for their helpful comments.
>
> **To W1 (originality):**
>
> We respectfully emphasize the originality and significance of our paper from the following aspects:
>
> 1. While there exist literature on second-order oracles in bilevel optimization, our work is the first one to **demonstrate the advantage of using second-order oracles**.
> In addition, it is also novel to conduct second-order method on the penalized objective instead of the $\varphi(x)$ to avoid the third-order information.
>
> 2. We do not simply consider the second-oracles in improving the convergence rates, but also propose efficient variants including ISFBA and LSFBA to further reduce the computational cost. The LSFBA method can even improve the prior state-of-the-art second-order method for minimax optimization.
>
> 3. In terms of the analysis, we develop new Lemmas to bound the inexact gradient and Hessian (Lemma 3.1), as well as new analysis to control both the error from lazy Hessian and the inexact gradient and Hessian, which is more complicated than the standard non-convex or minimax settings.
>
>
>
> **To W2\&Q1 (problem parameter):**
>
> We appreciate the reviewer's perspective, but we emphasize that the problem parameters are unavoidable for the penalized methods for bilevel optimization.
> One is required to set the $\lambda =\Omega(l\kappa\^3\epsilon\^{-1})$ such that
> $\|\nabla L\_{\lambda}\^*(x) - \nabla \varphi(x)\| = \mathcal{\epsilon}$.
> This means that even setting the penalized parameter requires the knowledge of the problem parameter.
>
> In addition, we also notice some tuning-free or parameter-free methods for bilevel optimization, however they require computing the second-order information of $g$ and can only achieve the complexity no better than $\mathcal{O}(\epsilon^{-2})$ [A, B].
>
>
>
> **To W3 (Discussion on SQP):**
>
> The SQP method cannot achieve the $\epsilon^{-1.5}$ complexity find the second-order stationary point for equality-constraint optimization in general.
>
> We ultilize the problem structure of the non-convex strongly-convex bilevel optimization that the second-order stationary point of the primal function $\varphi$ can be achieved by finding the second-order stationary point of the lagrange function in our algorithm design to achieve the desired rate.
> The development of this bilevel-specific approach, rather than relying on standard SQP, is a core technical strength of this work.
>
>
> **To W4\&Q7 (experiments):**
> We thank the authors for their suggestion on our experiments. We will include new experiments including Meta learning based on [C] and adjust the figure size in our revision. Due to time constraints, in these additional experiments we focus on state-of-the-art bilevel optimization baselines that have been used in the meta-learning setting. We report the meta-learning results below.
>
> **Table: Test accuracy (%) on FC100 under matched running-time budgets in the 5-way 5-shot meta-learning setting.**
>
> | Method | 600s | 1000s | 1400s |
> |---|---:|---:|---:|
> | PZOBO | 48.8 | 45.5 | 47.5 |
> | qNBO  | 46.0 | 46.2 | 44.9 |
> | F2BA  | 48.4 | 50.9 | 47.0 |
> | IFSBA | **51.7** | **51.1** | **51.6** |
>
>
> **Table: Test accuracy (%) on miniImageNet under matched running-time budgets in the 5-way 5-shot meta-learning setting.**
>
> | Method | 600s | 900s | 1200s |
> |---|---:|---:|---:|
> | PZOBO | 46.0 | 48.0 | 49.6 |
> | qNBO  | 46.3 | 48.9 | 45.0 |
> | F2BA  | 47.9 | 49.0 | 52.0 |
> | IFSBA | **51.2** | **52.0** | **52.7** |
>
> **To Q2**
> We will reformat the equations in the revision.
>
> **To Q3**
> Lemma 3.2 does not specify the conditions on $K_t^{1}, K_t^{2}$ of the algorthms;
> rather, it establishes the conditions required for the inexact cubic-regularization step.
> If the approximated $g$ and $H$ satisfies (7), applying the descent steps (line 7-9 in Algorithm 2) gurantees finding the $(\epsilon,\sqrt{\epsilon})$ SOSP.
> We subsequently provide Lemma 3.3 to explicitly show how to satisfy these conditions by setting $K_t^1$ and $K_t^2$  large enough. We will clarify this progression in the text.
>
>
>
> **To Q4**
> It means that we should choose $M\geq c\rho$, where $c>0$ is a constant.
>
> **To Q5**
> Assumption 4.1 is intended as a computaional cost model to sepcify how first- and second-order oracle calls are translated into total computational cost, rather than a structural mathematical assumption on the problem itself.
> Regarding (9), you are correct that \# 1st-order oracle and \# 2nd-order oracle denote the numbers of first- and second-order oracle calls and (9) converts these counts into total computational cost under the cost model in Assumption 4.1.
>
> **To Q6**
> Yes, you are correct, we will reformulate this in the revision. The different presentation was mainly due to the space constraints.
>
> [A] Tuning-Free Bilevel Optimization: New Algorithms and Convergence Analysis. ICLR 25
>
> [B] Problem-Parameter-Free Decentralized Bilevel Optimization. NeurIPS 25.
>
> [C]QNBO: Quasi-Newton Meets Bilevel Optimization. ICLR 25.

---

> > ### Author Rebuttal · Reviewer_bV4H · 2026-04-02
> >
> > I would like to thank the authors for their detailed rebuttals. I'm willing to increase my score to 4. A follow-up question to Q1: in practice, how sensitive are the algorithms to the knowledge of these constants? It may be hard to estimate them for a problem at hand.

---

> > > ### Author Response · Authors · 2026-04-03
> > >
> > > We sincerely thank the reviewer for the continued engagement with our work and for the follow-up question.
> > >
> > > The exact values of parameters like Lipschitz constants are highly problem-dependent, and thus maybe hard to estimate. However, their upper bounds (for Lipschitz continuity) or lower bounds (for strong convexity) can be explicitly estimated based on the model and the datasets (as detailed in our response to Reviewer vZ28, Q3).
> > >
> > > In practice, rather than computing strict worst-case global bounds, we treat the dependent algorithmic parameters (such as the cubic penalty $M$ and the value-function penalty $\lambda$) as tunable hyperparameters. To explicitly test the sensitivity of our FSBA method to the lack of exact knowledge of these constants, we conducted an additional ablation study on the Australian dataset for the Data Hypercleaning task.
> > >
> > > First, we fixed all other hyperparameters and varied the cubic penalty parameter $M$:
> > > | Setting | Gap at 20s | Gap at 40s | Gap at 60s |
> > > | --- | --- | --- | --- |
> > > | $M=0.01$| $8.0\times10^{-2}$| $5.8\times10^{-2}$| $4.2\times10^{-2}$|
> > > | $M=0.1$| $3.3\times10^{-2}$| $1.2\times10^{-2}$| $6.0\times10^{-3}$|
> > > | $M=1$| $6.3\times10^{-2}$| $2.3\times10^{-2}$| $1.0\times10^{-2}$|
> > > | $M=10$| $7.2\times10^{-2}$| $5.0\times10^{-2}$| $3.4\times10^{-2}$|
> > > | $M=100$| $1.3\times10^{-1}$| $7.5\times10^{-2}$| $6.5\times10^{-2}$|
> > >
> > > Next, we fixed all other hyperparameters and varied the value-function penalty parameter $\lambda$:
> > > | Setting | Gap at 20s | Gap at 40s | Gap at 60s |
> > > | --- | --- | --- | --- |
> > > | $\lambda=0.01$ | $3.5\times10^{-2}$ | $1.4\times10^{-2}$ | $7.0\times10^{-3}$ |
> > > | $\lambda=0.1$ | $3.6\times10^{-2}$ | $1.9\times10^{-2}$ | $1.2\times10^{-2}$ |
> > > | $\lambda=1.0$ | $7.5\times10^{-2}$ | $5.2\times10^{-2}$ | $3.3\times10^{-2}$ |
> > > | $\lambda=10.0$ | $1.7\times10^{-1}$ | $8.5\times10^{-2}$ | $3.1\times10^{-2}$ |
> > > | $\lambda=100.0$ | $1.8\times10^{-1}$ | $8.5\times10^{-2}$ | $2.3\times10^{-2}$ |
> > >
> > > We find the FSBA method consistently converges and achieves a small optimality gap even when the hyperparameters are varied over a massive range ($10^{-2}$ to $10^2$). Because these hyperparameters are theoretically dictated by the problem constants (e.g., $M \ge c\rho$), this empirical stability confirms that our method is relatively robust to the problem parameters in practice.
> > >
> > > ---
> > > Dear Reviewer bV4H,
> > >
> > > Since it is near the end of author-reviewer discussion period, we respectfully ask whether our reply to your follow-up question has addressed your concern. In case you need any remaining clarifications, we would be more than happy to reply. If all your questions are addressed properly, we sincerely hope you consider to increase your score to support our work.
> > >
> > > Best regards,
> > >
> > > Authors

---

### Official Review · Reviewer_vZ28 · 2026-03-12

**Soundness:** 2
**Presentation:** 2
**Significance:** 2
**Originality:** 2
**Overall Recommendation:** 4
**Confidence:** 3

**Summary:**

This paper integrates cubic-regularized Newton updates and lazy Hessian strategies into the upper-level update for nonconvex-strongly-convex bilevel optimization, resulting in several second-order algorithms. Using second-order oracles, the proposed methods achieve an accelerated iteration complexity of $\tilde{\mathcal{O}}(\epsilon^{-1.5})$ for finding approximate second-order stationary points.

**Compliance With Llm Reviewing Policy:**

Affirmed.

**Key Questions For Authors:**

I have three questions.
* The first is about the cubic Newton update. What specific stopping condition is employed to determine when the sub-problem has been sufficiently solved?
* My second question is about the experiments. Are all assumptions required by the theory satisfied in the experimental settings? Please clarify this in the paper.
* Finally, what are the exact values or bounds for the Lipschitz parameters and other constants?

**Limitations:**

Not Applicable.

**Strengths And Weaknesses:**

**Strengths**

* The paper generalizes the results of Luo et al. (2022), which studied non-convex-strongly-concave minimax optimization and finding second-order stationary points, to the bilevel optimization setting.
* The method incorporates cubic Newton and lazy Hessian strategies directly into the upper-level update.
* While the methodological combination is relatively direct, the algorithm utilizes a second-order oracle to achieve an accelerated iteration complexity of $\tilde{\mathcal{O}}(\epsilon^{-1.5})$, which is consistent with known results for Newton-type methods in non-convex smooth optimization.

**Weaknesses**

* The convergence analysis requires strong assumptions, such as Lipschitz continuity of the third-order derivative.
* Many parameters in the analysis are difficult to verify in practice. This may be the reason why the authors tune the stepsizes as well as other hyperparameters in the experiments.
* In the numerical experiments, it would be better to demonstrate in more detail whether the assumptions required by the theory are satisfied.
* It would also be helpful to include remarks explaining the convergence results to provide more intuition behind the formulas. For instance, it is not clear how to estimate the number of oracle calls appearing in equation (9).

**References**

Luo, L., Li, Y. and Chen, C., 2022. Finding second-order stationary points in nonconvex-strongly-concave minimax optimization.

---

> ### Author Rebuttal · Authors · 2026-03-28
>
> We thank the reviewer for their postive and helpful feedbacks on our work.
>
> **To W1**
> The assumption regarding the Lipschitz continuity of the third-order derivative on the lower level function is necessary for finding the SOSP in bilevel optimization [A, B]. Otherwise, one cannot guarantee the Hessian Lipschitz continuity of $\varphi(x)$.
>
> **To W2 (discussion on the parameters)**
>
> We appreciate the reviewer's perspective, but we emphasize that the problem parameters are unavoidable for the penalized methods for bilevel optimization.
> One is required to set the $\lambda =\Omega(l\kappa\^3\epsilon\^{-1})$ such that
> $\|\nabla L\_{\lambda}\^*(x) - \nabla \varphi(x)\| = \mathcal{\epsilon}$.
> This means that even setting the penalized parameter requires the knowledge of the problem parameter.
>
> While we notice some tuning-free or parameter-free methods for bilevel optimization, they require computing the second-order information of $g$ and can only achieve a complexity no better than $\mathcal{O}(\epsilon^{-2})$ [C, D].
>
>
>
> **To W3\&Q2 (experimental setting)**
>
> Yes, the theoretical assumptions are satisfied in our experimental setting. We will explicitly include the following details in our revision:
>
> * **Synthetic minimax problem**: the quadratic term on $y$ guarantees the strong convexity on $g$. Since the base functions are infinitely differentiable, their derivatives are Lipschitz continuous in some bounded domain.
> * **Data Hypercleaning**: the lower level $g$ consists of training an $l_2$-regularized logistic regression, which is strongly convex. Both $f$ and $g$ are inifitely differentiable, thus their  derivatives are Lipschitz continuous in some bounded domain.
> *  **Hyperparameter tuning**: the regularization term of $g$ indicates that it is at least exp($x$)-strongly-convex. The Lipschitz continuous conditions follow the same rigorous guarantees as the Data Hypercleaning task.
>
>
> **To W4**
> Thanks for your suggestion, we will include remarks explaining the convergence results and intuition. Specifically regarding Equation (9), the intuition is that it estimates the total computational cost by multiplying the number of algorithm iterations (derived in Theorem 3.4) by the per-iteration oracle costs defined in Assumption 4.1.
>
> **To Q1**
> Let $m_t(s) = g_t^{\top}s + \frac{1}{2}s^{\top}H_t s + \frac{M}{6}\|\|s\|\|^3$. It is sufficient to set $\|\|\nabla m_t(s_t)\|\| \leq  \epsilon$ as stopping condition. Let $h(\cdot)$ to denote $L_{\lambda}^*(\cdot)$, then it holds that
> $$
> \|\|\nabla h(x_{t+1})\|\|\leq \|\|\nabla h(x_t) + \nabla^2 h(x_t) s_t\|\| + \frac{\rho}{2}\|\|s_t\|\|^2\\
> \leq \|\|g_t + H_ts_t\|\| + \frac{\rho}{2}\|\|s_t\|\|^2 + \|\|g_t-\nabla h(x_t)\|\| + \|\|(H_t-\nabla^2 h(x_t))s_t\|\| \\
> \leq (\|\|\nabla m_t(s_t)\|\|+\epsilon_g) + \frac{M+\rho}{2}\|\|s_t\|\|^2 + \epsilon_H\|\|s_t\|\|
> $$
> and
> $$
> h(x_{t+1})- h(x_t) \leq \nabla h(x_t)^{\top}s_t + \frac{1}{2} s_t^{\top}\nabla^2 h(x_t) s_t + \frac{L_2}{6}\|\|s_t\|\|^3\\
> \leq g_t^{\top}s_t + \frac{1}{2}s_t^{\top}H_ts_t  + \epsilon_g \|\|s_t\|\| + \epsilon_H \|\|s_t\|\|^2 + \frac{L_2}{6}\|\|s_t\|\|^3\\
> = s_t^{\top}(g_t+H_ts_t/2 + M\|s_t\|s_t/3) +  \epsilon_g \|\|s_t\|\| + \epsilon_H\| \|s_t\|\|^2 + \frac{L_2}{6}\|\|s_t\|\|^3 - \frac{M}{3}\|\|s_t\|\|^3\\
> \leq (\epsilon_g+ \|\|\nabla m_t(s_t)\|\|)\|\|s_t\|\| + \epsilon_H\|\|s_t\|\|^{3}- \frac{M}{6}\|\|s_t\|\|^3.
> $$
> Note that the exact CRN-step means $\nabla m_t (s_t)=0$, for the inexact CRN step, it is sufficient to set $\|\|\nabla m_t(s_t\| = \mathcal{O}(\epsilon_g) = \mathcal{O}(\epsilon)$ to guarantee the same convergence rate to find the first-order stationary point
>
>
> **To Q3**
> The exact values of the theoretical constants (such as the smoothness parameter $l$, the Hessian Lipschitz parameter $\rho$, and the condition number $\kappa$) are highly problem-dependent.
>
> For example, in the Data Hypercleaning task, assuming the logit loss is bounded by $B = \max_{i,y}L(\langle a_i,y\rangle,y)$, then second-order derivatives of $g$ can be bounded by
>
> $$
> \|\|\nabla\_{xx}^2 g(x,y)\|\|\_2 \le \frac{1}{|D\_{tr}|} \max\_i |\sigma\^{''} (x\_i )| L\_i (y) =\mathcal{O}(B/|D_{tr}|),
> $$
>
> $$
> \|\|\nabla\_{xy}\^2 g(x,y)\|\|\_2 \le \frac{1}{4|D_{tr}|} \sqrt{ \sum\_{ i\in|D\_{tr}|}\|a\_i \|\_2\^2 },
> $$
> and
> $$
> \|\|\nabla\_{yy}\^2 g(x,y)\|\|\_2 \le \|\| \frac{1}{4|D\_{tr}|} \sum\_{i=1}\^{D_{tr}} \sigma(x\_i) a\_i a\_i\^\top \|\|\_2 + \mu \le \frac{1}{4|D_{tr}|} \lambda_{max}\left( \sum_{i=1}^{D_{tr}} a_i a_i^\top \right) + \mu,
> $$
> which provides the smooth constant of $g$. For other parameters, we can bounded them in a similar way. We will detail these exact bounds in the appendix.
>
> [A] Efficiently Escaping Saddle Points in Bilevel Optimization. JMLR 2025.
>
> [B] Near-Optimal Nonconvex-Strongly-Convex Bilevel Optimization with Fully First-Order Oracles. JMLR 2025.
>
> [C] Tuning-Free Bilevel Optimization: New Algorithms and Convergence Analysis. ICLR 25
>
> [D] Problem-Parameter-Free Decentralized Bilevel Optimization. NeurIPS 25.

---

> > ### Author Rebuttal · Reviewer_vZ28 · 2026-04-03
> >
> > Thank you for the rebuttal. The responses clarified my questions. I will keep my week accept score but it should lean towards accept.

---

> > > ### Author Response · Authors · 2026-04-04
> > >
> > > We are glad that our responses have addressed your concerns. Thank you for your support to our work.

---

### Decision · Program_Chairs · 2026-04-30

**Decision:**

Accept (regular)

**Comment:**

This paper studies second-order methods for nonconvex-strongly-convex bilevel optimization, proposing FSBA (Fully Second-order Bilevel Approximation), which applies cubic-regularized Newton updates to a bilevel proxy objective. The paper also introduces a lazy variant (LFSBA) that reuses second-order information across iterations and an inexact variant (IFSBA) that avoids frequent Hessian computation. The main theoretical result is an accelerated iteration complexity of O-tilde(epsilon^{-1.5}) for finding approximate second-order stationary points, improving upon the O-tilde(epsilon^{-1.75}) rate of prior first-order bilevel methods. The approach extends to nonconvex-strongly-concave minimax optimization.

The three completed reviews are moderately positive, with scores of 4, 5, and 4. All reviewers acknowledge that the paper is technically sound and addresses a meaningful theoretical gap.